# Decoupled SGDA for Games with Intermittent Strategy Communication

**Ali Zindari** [* 1]  **Parham Yazdkhasti** [* 1]  **Anton Rodomanov** [1]  **Tatjana Chavdarova** [2]  **Sebastian U. Stich** [1]

## Abstract

We introduce *Decoupled SGDA*, a novel adaptation of Stochastic Gradient Descent Ascent (SGDA) tailored for multiplayer games with intermittent strategy communication. Unlike prior methods, Decoupled SGDA enables players to update strategies locally using outdated opponent strategies, significantly reducing communication overhead. For Strongly-Convex-Strongly-Concave (SCSC) games, it achieves near-optimal communication complexity comparable to the best-known GDA rates. For *weakly coupled* games where the interaction between players is lower relative to the non-interactive part of the game, Decoupled SGDA significantly reduces communication costs compared to standard SGDA. Additionally, *Decoupled SGDA* outperforms federated minimax approaches in noisy, imbalanced settings. These results establish *Decoupled SGDA* as a transformative approach for distributed optimization in resource-constrained environments.

## 1. Introduction

Several real-world problems in diverse areas, such as economics and computer science, can frequently be described as $N$-player differentiable games (Von Neumann & Morgenstern, 2007). While players may have competing objectives, the aim is to identify an equilibrium, a strategy where no player benefits from deviating unilaterally. Examples of such games in machine learning include Generative Adversarial Networks (GANs, Goodfellow et al., 2014), adversarial robustness (Madry et al., 2017; Shafahi et al.,

2019; Robey et al., 2023) and multi-agent reinforcement learning (e.g., Lowe et al., 2017; Li et al., 2019).

Several gradient-based methods have been proposed for solving minimax problems (Korpelevich, 1976; Popov, 1980; Balduzzi et al., 2018; Nouiehed et al., 2019; Chavdarova et al., 2020; Kovalev & Gasnikov, 2022). One of the most widely used is the gradient descent method. In the context of 2-player zero-sum minimax games, this approach is referred to as *Gradient Descent Ascent* (GDA), where the minimizing player ($\mathbf{u}$–player) takes descent steps and the maximizing player ($\mathbf{v}$–player) takes ascent steps.

In some situations, however, players may not have direct access to their opponents' exact strategies. The $\mathbf{u}$–player might only have a noisy estimate of $\mathbf{v}$ when updating its parameters, and vice versa. In extreme cases, players might operate with outdated strategies from their opponents, with limited opportunities to synchronize. We refer to this scenario as *games with intermittent strategy communication* (ISC-games). Here are a few illustrative examples:

- **Corporate competitors.** Companies frequently adjust their strategies based on individual objectives and the strategies of their competitors. For instance, *Netflix* may need to lower its prices if a competitor like *Max* reduces its subscription rates (Jagadeesan et al., 2022). Corporations may occasionally release (noisy) general information about their strategies, giving each company an imperfect understanding of its competitor's actions. Alternatively, companies might hire experts to estimate competitor strategies using publicly available data, although this process is expensive and infrequent.

- **$N$-agents with restricted communication.** In control theory, applications involving drones or robots are modeled with $N$-player games (see Spica et al., 2020; Laine et al., 2021; Zhou et al., 2021, and references therein). However, due to factors like long distances or limited battery life caused by weight constraints, communication between agents regarding learned strategies is costly and can only occur intermittently.

In summary, this paper focuses on the following questions.

- *Can players effectively learn and adapt locally in ISC-games when relying on noisy or outdated opponent strategies?*

---

*Equal contribution [1]CISPA Helmholtz Center for Information Security, Saarbrücken, Germany [2]Politecnico di Milano, Italy. Correspondence to: Ali Zindari <ali.zindari@cispa.de>, Parham Yazdkhasti <parham.yazdkhasti@cispa.de>, Anton Rodomanov , Tatjana Chavdarova <tatjana.chavdarova@polimi.it>, Sebastian U. Stich <stich@cispa.de>.

*Proceedings of the 42$^{nd}$ International Conference on Machine Learning*, Vancouver, Canada. PMLR 267, 2025. Copyright 2025 by the author(s).

- *How do the convergence rate and communication costs of the proposed optimization method compare to the baseline, and can communication costs be significantly reduced?*

To address the first question, we propose an extension of the gradient descent method where agents perform local updates while using outdated strategies from their opponents. In minimax problems, we term this approach *Decoupled SGDA*, and in $N$-player games, it is *Decoupled SGD*. The second question is examined by analyzing the convergence rate of Decoupled SGD(A) and identifying a specific class of problems, called *Weakly Coupled Games*, where communication speed-up (acceleration) can be achieved.

**Contributions.** Our contributions include:

- We introduce **Decoupled SGD(A)**, the first method designed for ISC-games, where each player performs local updates using outdated opponent strategies.
- We analyze its convergence in both the strongly-convex strongly-concave (SCSC) setting and in $N$-player games where each player's utility is strongly convex.
- We introduce a non-standard set of assumptions to measure the coupling of the players' objectives. This allows us to identify a specific regime, termed *Weakly Coupled Games*, where Decoupled SGD(A) demonstrates **communication acceleration** compared to the baseline GD(A), by removing the dependency on player conditioning.
- We show that Decoupled SGDA can even outperform the optimal first-order method for solving SCSC games in terms of communication rounds under a slightly stronger assumption than weakly coupled games.
- Our method achieves **robustness to noise imbalance**, a significant limitation of existing federated minimax methods.
- We study the convergence of Decoupled SGDA for quadratic minimax games with bilinear coupling between the players, providing additional in-depth insights into the algorithm's convergence behavior.
- Through numerical experiments, we validate the practical benefits of Decoupled SGDA in non-convex GAN training, federated learning with imbalanced noise, and in weakly coupled quadratic minimax games, showcasing its versatility.
- We propose **a novel heuristic (Ghost-SGDA)**, detailed in Appendix G, to further accelerate the convergence of Decoupled SGDA by leveraging predictive updates for opponent strategies. Our numerical results demonstrate its practical effectiveness, showing that Ghost-SGDA can achieve faster communication efficiency and convergence even in highly interactive games. This heuristic opens a promising new direction for distributed optimization, potentially going beyond the theoretical guarantees proven in this work.

To simplify the exposition, the main body of the paper focuses on the minimax setting, while the extension to $N$-player games is presented in Appendix C.

### 1.1. Related Works

Our work[1] draws from multiple lines of work, and herein, we review these and discuss the difference with federated learning. Appendix E gives additional discussion and lists works on decentralized optimization. The latter are further from our work in that there is no centralized communication, and nodes communicate with neighbors.

**Game optimization.** Nemirovski (2004); Nesterov (2007) achieve a rate of $\mathcal{O}(\frac{1}{T})$ for convex-concave minimax problems. For strongly-convex-strongly-concave games, (i) Thekumparampil et al. (2019) combine Nestrov's Accelerated Gradient and mirror-prox and achieve $\tilde{\mathcal{O}}(\frac{1}{T^2})$ rate of convergence, (ii) Wang & Li (2020) explore ideas from accelerated proximal point and achieve a linear rate, and (iii) Kovalev & Gasnikov (2022) propose a method with $\mathcal{O}(\sqrt{\kappa_u \kappa_v} \log \frac{1}{\epsilon})$ rate of convergence which matches the lower bounds (Zhang et al., 2022b; Ibrahim et al., 2020). Several works focus on accelerating the convergence of GDA (Lee et al., 2024; Zhang et al., 2022a). Quadratic games with bilinear coupling are studied in (Zhang et al., 2021). Nouiehed et al. (2019) propose a method that performs multiple first-order steps on only one of the parameters to solve minimax problems. Tsaknakis et al. (2021) study a generalized minimax problem with linear constraints coupling the decision variables. Tseng & Yun (2009) study coordinate gradient descent method for minimizing the sum of a smooth and separable convex function. Jain et al. (2018) and Yoon & Ryu (2021) present algorithms with accelerated $\mathcal{O}(1/k^2)$ rates for smooth minimax optimization and establish the optimality of this rate through a matching lower bound. In the context of multi-player games, several works have explored multi-agent reinforcement learning in a distributed setting, where agents update their policies without access to the policies of others (Lu et al., 2021; Sayin et al., 2021; Jiang & Lu, 2022). Independent and concurrent work by Yoon et al. (2025) also considers decoupled updates for equilibrium computation in $N$-player games. While we both address this setting (see Appendix C), their method does not achieve the same communication efficiency as ours. See Remark 4.6 for a detailed comparison.

**Federated learning.** Building on the foundational work of McMahan et al. (2017), numerous works have explored distributed minimization, or federated learning, across various settings (e.g., Stich, 2019a; Koloskova et al., 2020; Karimireddy et al., 2020; Woodworth et al., 2020a;b). In

---

[1] An early version of this work was presented at an ICML 2024 workshop (Zindari et al., 2024).

the context of minimax optimization, Deng & Mahdavi (2021); Sharma et al. (2022); Zhang et al. (2024) extended the so-called *Local SGD* (Stich, 2019a) to the minimax setting, achieving convergence rates for different classes of functions in both heterogeneous and homogeneous regimes. Although both Federated Minimax and Decoupled SGDA are designed to solve minimax optimization problems in a distributed fashion, their approaches to achieving this are fundamentally different. Refer to Section F.1 for more details.

## 2. Setting and preliminaries

In the main body of the paper, we focus on the two-player setting, as this allows us to clearly present the key ideas and insights of our approach. The extension to the more general $N$-player games is detailed in Appendix C. Specifically, we consider the following saddle-point problem over $\mathcal{X} = \mathcal{X}_u \times \mathcal{X}_v$, with $\mathcal{X}_u = \mathbb{R}^{d_u}, \mathcal{X}_v = \mathbb{R}^{d_v}$:

$$\min_{\mathbf{u}\in\mathcal{X}_u} \max_{\mathbf{v}\in\mathcal{X}_v} f(\mathbf{u}, \mathbf{v}), \qquad \text{(SP)}$$

where $f\colon \mathcal{X} \to \mathbb{R}$ is a differentiable function. Its solution is defined as a point $\mathbf{x}^\star \equiv (\mathbf{u}^\star, \mathbf{v}^\star) \in \mathcal{X}$ satisfying the following variational principle: $f(\mathbf{u}^\star, \mathbf{v}) \leqslant f(\mathbf{u}^\star, \mathbf{v}^\star) \leqslant f(\mathbf{u}, \mathbf{v}^\star)$ for all $(\mathbf{u}, \mathbf{v}) \in \mathcal{X}$. In ISC-games, players often have access only to the outdated strategies of their opponents. To address this in our analysis, it is useful to define the following operator $F_{\bar{\mathbf{x}}}(\mathbf{x})\colon \mathcal{X} \to \mathcal{X}$, which incorporates a reference point $\bar{\mathbf{x}} \equiv (\bar{\mathbf{u}}, \bar{\mathbf{v}}) \in \mathcal{X}$—typically the most recent synchronization point—to account for this delay:

$$F_{\bar{\mathbf{x}}}(\mathbf{x}) := (\nabla_u f(\mathbf{u}, \bar{\mathbf{v}}), -\nabla_v f(\bar{\mathbf{u}}, \mathbf{v})), \qquad \mathbf{x}, \bar{\mathbf{x}} \in \mathcal{X},$$
$$F(\mathbf{x}) := (\nabla_u f(\mathbf{u}, \mathbf{v}), -\nabla_v f(\mathbf{u}, \mathbf{v})), \qquad \mathbf{x} \in \mathcal{X}. \qquad (1)$$

In the special case of $\bar{\mathbf{x}} = \mathbf{x}$, we recover the definition of the commonly used operator $F(\mathbf{x}) = F_{\bar{\mathbf{x}}=\mathbf{x}}(\mathbf{x}) = (\nabla_u f(\mathbf{u}, \mathbf{v}), -\nabla_v f(\mathbf{u}, \mathbf{v}))$.

**Notation.** Bold lower and upper case denote vectors and matrices, respectively. We often denote the two players as $\mathbf{u} \in (\mathcal{X}_u = \mathbb{R}^{d_u})$ and $\mathbf{v} \in (\mathcal{X}_v = \mathbb{R}^{d_v})$. The product space $\mathcal{X} = \mathcal{X}_u \times \mathcal{X}_v = \mathbb{R}^d$ (with $d = d_u + d_v$) consists of vectors $\mathbf{x} = (\mathbf{u}, \mathbf{v}) \in \mathbb{R}^d$, where $\mathbf{u} \in \mathcal{X}_u$ and $\mathbf{v} \in \mathcal{X}_v$. For a differentiable function $f\colon \mathcal{X} \to \mathbb{R}$, we denote partial gradients at a point $\mathbf{x} = (\mathbf{u}, \mathbf{v}) \in \mathcal{X}$ w.r.t. the corresponding variables by $\nabla_u f(\mathbf{x})$ and $\nabla_v f(\mathbf{x})$, respectively, so that $\nabla f(\mathbf{x}) = (\nabla_u f(\mathbf{x}), \nabla_v f(\mathbf{x}))$. $\langle \cdot, \cdot \rangle$ denotes inner product. We assume that the spaces $\mathcal{X}_u$ and $\mathcal{X}_v$ are equipped with certain Euclidean norms, $\|\mathbf{u}\|_u := \langle \mathbf{P}_u \mathbf{u}, \mathbf{u} \rangle^{1/2}$ and $\|\mathbf{v}\|_v := \langle \mathbf{P}_v \mathbf{v}, \mathbf{v} \rangle^{1/2}$, respectively, where $\mathbf{P}_u$ and $\mathbf{P}_v$ are given symmetric positive definite matrices. The norm in the space $\mathcal{X}$ is then defined by $\|\mathbf{x}\| = (\alpha_u \|\mathbf{u}\|_u^2 + \alpha_v \|\mathbf{v}\|_v^2)^{1/2}$ where $\alpha_u, \alpha_v > 0$; thus,

$\|\mathbf{x}\| = \langle \mathbf{P}\mathbf{x}, \mathbf{x} \rangle^{1/2}$, where $\mathbf{P}$ is the block-diagonal matrix with blocks $\alpha_u \mathbf{P}_u$ and $\alpha_v \mathbf{P}_v$ ($\mathbf{P} = \mathrm{diag}(\alpha_u \mathbf{P}_u, \alpha_v \mathbf{P}_v)$). The parameters $\alpha_u, \alpha_v$ can be seen as scaling factors for players that can be optimized separately. One can easily assume $\mathbf{P}_u = \mathbf{P}_v = \mathbf{I}$, $\mathbf{P} = \mathbf{I}$ and $\alpha_u = \alpha_v = 1$ and recover the common euclidean norm $\|\mathbf{x}\| = \sqrt{\langle \mathbf{x}, \mathbf{x} \rangle}$. The corresponding dual norms are defined in the standard way: $\|\mathbf{g}_u\|_{u,*} := \max_{\|\mathbf{u}\|_u = 1} \langle \mathbf{g}_u, \mathbf{u} \rangle = \langle \mathbf{g}_u, \mathbf{P}_u^{-1}\mathbf{g}_u \rangle^{1/2}$ ($\mathbf{g}_u \in \mathcal{X}_u$), $\|\mathbf{g}_v\|_{v,*} := \max_{\|\mathbf{v}\|_v = 1} \langle \mathbf{g}_v, \mathbf{v} \rangle = \langle \mathbf{g}_v, \mathbf{P}_v^{-1}\mathbf{g}_v \rangle^{1/2}$ ($\mathbf{g}_v \in \mathcal{X}_v$), and $\|\mathbf{g}\|_* := \max_{\|\mathbf{x}\|=1} \langle \mathbf{g}, \mathbf{x} \rangle = (\frac{1}{\alpha_u}\|\mathbf{g}_u\|_{u,*}^2 + \frac{1}{\alpha_v}\|\mathbf{g}_v\|_{v,*}^2)^{1/2} = \langle \mathbf{g}, \mathbf{P}^{-1}\mathbf{g} \rangle^{1/2}$ ($\mathbf{g} \equiv (\mathbf{g}_u, \mathbf{g}_v) \in \mathcal{X}$).

Now we outline the necessary assumptions for establishing the convergence of our method.

**Assumption 2.1** (Strong monotonicity). Operators $F_{\bar{\mathbf{x}}}$ and $F$ from (1) are strongly monotone with parameters $\bar{\mu}, \mu > 0$, i.e., for all $\mathbf{x}, \bar{\mathbf{x}}, \mathbf{x}' \in \mathcal{X}$, the following inequalities hold:

$$\langle F_{\bar{\mathbf{x}}}(\mathbf{x}) - F_{\bar{\mathbf{x}}}(\mathbf{x}'), \mathbf{x} - \mathbf{x}' \rangle \geqslant \bar{\mu}\|\mathbf{x} - \mathbf{x}'\|^2,$$
$$\langle F(\mathbf{x}) - F(\mathbf{x}'), \mathbf{x} - \mathbf{x}' \rangle \geqslant \mu\|\mathbf{x} - \mathbf{x}'\|^2. \qquad (2)$$

We can show that $\bar{\mu} = \min\{\mu_u/\alpha_u, \mu_v/\alpha_v\}$ (Proof in Lemma B.5) where $\mu_u$ is the strong convexity parameter of $f$ in $\mathbf{u}$ and $\mu_v$ is the strong concavity parameter of $f$ in $\mathbf{v}$.

**Assumption 2.2** (Lipschitz smoothness). Operators $F_{\bar{\mathbf{x}}}$ and $F$ from (1) are Lipschitz with parameters $\bar{L}$ and $L$, i.e., for all $\mathbf{x}, \mathbf{x}', \bar{\mathbf{x}} \in \mathcal{X}$, the following inequalities hold:

$$\|F_{\bar{\mathbf{x}}}(\mathbf{x}) - F_{\bar{\mathbf{x}}}(\mathbf{x}')\|_* \leqslant \bar{L}\|\mathbf{x} - \mathbf{x}'\|,$$
$$\|F(\mathbf{x}) - F(\mathbf{x}')\|_* \leqslant L\|\mathbf{x} - \mathbf{x}'\|. \qquad (3)$$

While we assume both operators are smooth, Lemma B.2 shows that $\bar{L} \leqslant L$.

**Assumption 2.3.** The norm of the difference between operators $F_{\bar{\mathbf{x}}}$ from (1) is upper bounded with parameter $L_c$ for for all $\mathbf{x}, \bar{\mathbf{x}} \in \mathcal{X}$ as follows:

$$\|F_{\bar{\mathbf{x}}}(\mathbf{x}) - F(\mathbf{x})\|_* \leqslant L_c\|\mathbf{x} - \bar{\mathbf{x}}\|. \qquad (4)$$

It is possible to show $L_c = 1/\sqrt{\alpha_u \alpha_v}\max\{L_{uv}, L_{vu}\}$ (Proof in Lemma B.6) where $\|\nabla_u f(\mathbf{u}, \mathbf{v}) - \nabla_u f(\mathbf{u}, \mathbf{v}')\|_{u,*} \leqslant L_{uv}\|\mathbf{v} - \mathbf{v}'\|_v$ and $\|\nabla_v f(\mathbf{u}, \mathbf{v}) - \nabla_v f(\mathbf{u}', \mathbf{v})\|_{v,*} \leqslant L_{vu}\|\mathbf{u} - \mathbf{u}'\|_u$. Here, we take the derivative with respect to one variable while varying the other. We will demonstrate in Section 4 that this constant plays an important role in communication acceleration as it quantifies the interaction level of the game. In fact, $L_c$ can be much smaller than $L$ and can be even zero. We show it always holds that $L_c \leqslant L$ (Proof in Lemma B.2).

For the reader's convenience, we also present Tables 2 and 3 in the appendix, summarizing our notations.

**Assumption 2.4.** There exists finite constants $\bar{\sigma}^2$ such that for all $\mathbf{x}, \bar{\mathbf{x}} \in \mathcal{X}$:

$$\mathbb{E}_\xi \left[ \| G_{\bar{\mathbf{x}}}(\mathbf{x}, \xi) - F_{\bar{\mathbf{x}}}(\mathbf{x}) \|_*^2 \right] \leqslant \bar{\sigma}^2 . \tag{5}$$

where $G_{\bar{\mathbf{x}}}(\mathbf{x}, \xi)$ is an unbiased stochastic gradient oracle that each player has access to with the property $\mathbb{E}[G_{\bar{\mathbf{x}}}(\mathbf{x}, \xi)] = F_{\bar{\mathbf{x}}}(\mathbf{x})$.

As we assumed that the above inequality holds for all $\bar{\mathbf{x}} \in \mathcal{X}$, we also cover the common operator $F$ and we denote $G(\mathbf{x}, \xi) \equiv G_{\bar{\mathbf{x}}=\mathbf{x}}(\mathbf{x}, \xi)$.

## 3. Decoupled SGDA for two-player games

In this section we introduce Decoupled SGDA and explain its motivation.

**Common Approach: Stochastic Gradient Descent Ascent (SGDA).** A standard way for solving (SP) is as follows:

$$\mathbf{x}_{t+1} = \mathbf{x}_t - \gamma \mathbf{P}^{-1} G(\mathbf{x}_t, \xi),$$
$$G(\mathbf{x}, \xi) \equiv \begin{pmatrix} \nabla_u f(\mathbf{u}, \mathbf{v}; \xi) \\ -\nabla_v f(\mathbf{u}, \mathbf{v}; \xi) \end{pmatrix}, \tag{6}$$

for a given positive definite matrix $\mathbf{P}$.[2] However, in a distributed setting, the players need one *round of communication* to exchange their parameters $(\mathbf{u}_t, \mathbf{v}_t)$ in every step of the method. This is because SGDA requires the most recent parameters from each player to take an step. In many real-world scenarios, however, communicating at every step may not be feasible due to the high cost.

**Communication Efficient Strategy Exchange.** To alleviate this communication issue, earlier works proposed so-called *local update methods* that reduce the amount of communication by performing local parameter updates for each player separately. For these methods, it is common to assume that both players—the minimization player $\mathbf{u}$, and the maximization player $\mathbf{v}$—have access unbiased stochastic oracles $G_u(\mathbf{x}, \xi), G_v(\mathbf{x}, \xi) : \mathcal{X} \to \mathcal{X}$, with the property $\mathbb{E}_\xi[G_u(\mathbf{x}, \xi)] = F(\mathbf{x})$, $\mathbb{E}_\xi[G_v(\mathbf{x}, \xi)] = F(\mathbf{x})$ and following bound on the variance of the noise:

$$\begin{cases} \mathbb{E}_\xi[\|[G_u(\mathbf{x}, \xi)]_u - [F(\mathbf{x})]_u\|_{u,*}^2] \leqslant \sigma_{uu}^2, \\ \mathbb{E}_\xi[\|[G_u(\mathbf{x}, \xi)]_v - [F(\mathbf{x})]_v\|_{v,*}^2] \leqslant \sigma_{uv}^2, \end{cases}$$
$$\begin{cases} \mathbb{E}_\xi[\|[G_v(\mathbf{x}, \xi)]_v - [F(\mathbf{x})]_v\|_{v,*}^2] \leqslant \sigma_{vv}^2, \\ \mathbb{E}_\xi[\|[G_v(\mathbf{x}, \xi)]_u - [F(\mathbf{x})]_u\|_{u,*}^2] \leqslant \sigma_{vu}^2. \end{cases} \tag{7}$$

Here, we use the operator $[\cdot]_i$ to denote the coordinates corresponding to player $i \in \{u, v\}$. Both players could perform

---

[2]In optimization literature, the matrix $\mathbf{P}$ is known as the preconditioning matrix. For simplicity, we can assume $\mathbf{P} = \mathbf{I}$ without disrupting the flow of the paper.

$K \geqslant 1$ updates on a local copy of the parameters. After every communication round, local variables are initialized as $\mathbf{x}_t^v = \mathbf{x}_t^u = \mathbf{x}_t$, and updated as:

$$\mathbf{x}_{t+K}^u = \mathbf{x}_t^u - \gamma (\alpha_u \mathbf{P}_u)^{-1} \sum_{i=0}^{K-1} G_u(\mathbf{x}_{t+i}^u, \xi_{t+i}),$$
$$\mathbf{x}_{t+K}^v = \mathbf{x}_t^v - \gamma (\alpha_v \mathbf{P}_v)^{-1} \sum_{i=0}^{K-1} G_v(\mathbf{x}_{t+i}^v, \xi_{t+i}). \tag{8}$$

The local variables are then synchronized in a communication round, $\mathbf{x}_{t+K} := \frac{1}{2} \left( \mathbf{x}_{t+K}^u + \mathbf{x}_{t+K}^v \right)$. This is a standard approach in distributed optimization. However, this method does not apply to our setting, as we would need to assume that the stochastic noise of the oracles $\sigma_{uv}^2$ and $\sigma_{vu}^2$ are bounded.

**Decoupled SGDA: Communication Efficient with Reliable Information.** We are considering a setting where the two players may not have access to their opponent's strategies or gradients, and only assume that the private components of the gradients have bounded variance, see Assumption 2.4. This reflects real-world challenges where it is hard to share reliable (with bounded noise) information between the communication rounds. For this setting, we therefore propose that each player should only use the *reliable information*, that is $[G_u(\mathbf{x}, \xi)]_u$ for player $\mathbf{u}$, and $[G_v(\mathbf{x}, \xi)]_v$ for player $\mathbf{v}$, and wait for the communication round to get reliable information about the other players. We introduce the oracle $G_0(\mathbf{x}, \xi) \equiv G_{\bar{\mathbf{x}}=\mathbf{x}_0^r}(\mathbf{x}, \xi)$ for $i = \{u, v\}$ where $\mathbf{x}_0 = (\mathbf{u}_0, \mathbf{v}_0)$ refers to the parameters of each player at the beginning of the round. Now we can write the update rule of our method as:

$$\mathbf{x}_K^r = \mathbf{x}_0^r - \gamma \mathbf{P}^{-1} \sum_{t=0}^{K-1} G_0(\mathbf{x}_t^r, \xi_t),$$
$$G_0(\mathbf{x}_t^r) \equiv \begin{pmatrix} [G_u(\mathbf{u}_t^r, \mathbf{v}_0^r)]_u \\ [G_v(\mathbf{u}_0^r, \mathbf{v}_t^r)]_v \end{pmatrix} \equiv \begin{pmatrix} \nabla_u f(\mathbf{u}_t^r, \mathbf{v}_0^r) \\ -\nabla_v f(\mathbf{u}_0^r, \mathbf{v}_t^r) \end{pmatrix}. \tag{9}$$

Here, the index $t$ denotes the local update step in the current local update phase, and the superscript $r$ indexes the local phases. One communication round is needed for exchanging the updated parameters $(\mathbf{u}_K^r, \mathbf{v}_K^r)$ when passing to the next round. Here we allow the variances $\sigma_{uv}^2$ and $\sigma_{vu}^2$ to be arbitrarily large and we only need $\bar{\sigma}^2 \leqslant \sigma_{uu}^2 + \sigma_{vv}^2$ to be finite which is an advantage of our method compared to local update methods.

**Method.** We formalize our method in Algorithm 1. Decoupled SGDA has a round-wise update scheme allowing each player to share his parameters only once in a while. At the beginning of each round $r$, each player receives the most recent parameters of the other player. Then all players start taking $K$ local steps and updating **only** their own parameters

---

**Algorithm 1** Decoupled SGDA for two-player[a] games

---

1: **Input:** step size $\gamma$, initialization $\mathbf{x}_0 = (\mathbf{u}_0, \mathbf{v}_0)$, number of rounds $R$, number of local GD steps $K$, $\alpha_u$, $\alpha_v$, $\mathbf{P}$[b]
2: **for** $r \in \{1, \ldots, R\}$ **do**
3:     **for** $t \in \{1, \ldots, K\}$ **do**
4:         $\mathbf{u}_{t+1}^r \leftarrow \mathbf{u}_t^r - \gamma(\alpha_u \mathbf{P}_u)^{-1} \nabla_u f(\mathbf{u}_t^r, \mathbf{v}_0^r; \xi)$
5:         $\mathbf{v}_{t+1}^r \leftarrow \mathbf{v}_t^r + \gamma(\alpha_v \mathbf{P}_v)^{-1} \nabla_v f(\mathbf{u}_0^r, \mathbf{v}_t^r; \xi)$
6:     **end for**
7:     **Communicate** $(\mathbf{u}_K^r, \mathbf{v}_K^r)$ to each player
8: **end for**
9: **Output:** $\mathbf{x}_K^R = (\mathbf{u}_K^R, \mathbf{v}_K^R)$

---

[a]The extension to $N$-player games is displayed in Appendix C in Alg. 2.

[b]The constants $\alpha_u, \alpha_v, \mathbf{P}$ are determined by the vector norm that we specify. For simplicity, the reader can assume that $\alpha_u = \alpha_v = 1$ and $\mathbf{P}_u = \mathbf{P}_v = \mathbf{I}$. These terms are included for the sake of completeness, though they are not essential for the main results.

using the information they received at the beginning of the round from other players. Note that our method is a general framework and one can use any first-order method to take local steps and not just the GD steps as illustrated here. The extension to $N$-player games is displayed in Appendix C in Alg. 2.

**Intuition.** To provide some intuition on why Decoupled SGDA might work, consider that the objective of minimax games (SP) can be written as:

$$f(\mathbf{u}, \mathbf{v}) = g(\mathbf{u}) - h(\mathbf{v}) + r(\mathbf{u}, \mathbf{v}) \qquad (10)$$

where $g(\mathbf{u})$ and $h(\mathbf{v})$ represent the independent contributions of each player, and  captures the interaction between them. Note that in this formulation, $r(\mathbf{u}, \mathbf{v})$ cannot be decomposed in the same way as $f$, as it specifically captures the interdependent aspects of $\mathbf{u}$ and $\mathbf{v}$.

In the special case when $r(\mathbf{u}, \mathbf{v}) \equiv 0$, i.e., there is no interaction, the problem does not require any communication: the optimal solution can be found by minimizing $g$ and $h$ **separately**. A method like SGDA is, therefore, not a good choice in this setting, as it requires to communicate parameters $(\mathbf{u}, \mathbf{v})$ in every step of the method, although this is unnecessary. In contrast, when the coupling $r(\mathbf{u}, \mathbf{v})$ is significant, then optimizing $g$ and $h$ separately might not be a good strategy. Decoupled SGDA aims to find a balance between the two extremes. In the following, we will characterize some settings where Decoupled SGDA provably uses significantly fewer communication rounds than SGDA or other baselines (see also Table 1).

**Extensions of Decoupled SGDA (Appendix G).** It is clear that our method is a general framework, providing flexibility for various modifications and adaptations. For

instance, our method allows for any first-order update rule to be applied for the local steps like GDA, Extra Gradient (EG), and Optimistic Gradient Descent Ascent (OGDA). Note that in this work, we focused on GDA updates, leaving the analysis of other methods for future work. Moreover, in Section G, we present **Ghost-SGDA**, where each player aims to estimate the other player's parameters using the so-called **Ghost Sequence**, which leads to further acceleration in terms of the number of rounds.

## 4. Convergence Guarantees

We *could* analyze our method under the common smoothness assumptions in the literature (see Appendix B.2 for the details). However, these assumptions are often overly pessimistic in distributed settings, as they fail to account for the interaction level between players. Specifically, they treat games with high and low interaction identically, yielding the same convergence rates in both cases. In contrast, we introduce an important parameter $L_c$ (see Assumption 2.3) that quantifies the interaction level of the game. This allows us to achieve communication acceleration in games with low interaction through a novel proof technique (see Section B.1). To formalize this, we first introduce the notion of *Weakly Coupled Games / Regime* and then provide the convergence guarantee for our method.

Given a strongly-convex strongly-concave (SCSC) zero-sum minimax game $f(\mathbf{u}, \mathbf{v})$, we define the **coupling degree** parameter $\kappa_c$ for this game as follows:

$$\boxed{\kappa_c := \frac{L_c}{\bar{\mu}}} \qquad (11)$$

This variable measures the level of interaction in the game. A smaller value of $\kappa_c$ indicates less interaction. For any $f(\mathbf{u}, \mathbf{v})$, we say the game is **Weakly Coupled** if:

$$\kappa_c \leqslant \frac{1}{4}. \qquad (12)$$

We say the game is **Fully Decoupled** if $\kappa_c = 0$, which implies that $r(\mathbf{u}, \mathbf{v}) = 0$ (see Eq. (10) and Lemma B.3). These games are an extreme case of weakly coupled games. In weakly coupled games, each player's dynamics are mostly driven by their own pay-off function, with little influence from the other player.

**Theorem 4.1.** *For any $R \geqslant 1$ and any $K \geqslant \frac{1}{\gamma\mu} \log\left(\frac{4}{\kappa_c}\right)$, after running Decoupled SGDA for a total of $T = KR$ iterations on a function $f$, with the stepsize $\gamma \leqslant \frac{\bar{\mu}}{L^2}$ in the weakly coupled regime ($4\kappa_c \leqslant 1$), we get a rate of:*

$$\mathbb{E}\big[\|\mathbf{x}_K^R - \mathbf{x}^\star\|^2\big] \leqslant D^2 \exp\Big(-(1 - 4\kappa_c)R\Big) + \frac{8\kappa_c \bar{\sigma}^2 \gamma}{\mu(1 - 4\kappa_c)}.$$

*Table 1.* Comparison of Communication Complexity and Acceleration Condition for Different Methods. *Speed up* lists the conditions under which Decoupled SGDA achieves acceleration relative to the respective method. Note that for simplicity and in order to compare our results with other works, we consider $\mathbf{P}_u = \mathbf{P}_v = \mathbf{I}$, $\mathbf{P} = \mathbf{I}$ and $\alpha_u = \alpha_v = 1$.

| Method | Communication Complexity *(Fully Decoupled)* | Communication Complexity *(General Bound)* | Speed Up |
|---|---|---|---|
| GDA (Lee et al., 2024) | $(\kappa_u + \kappa_v) \log \frac{1}{\epsilon}$ | $(\kappa_u + \kappa_v + \kappa_{uv}^2) \log \frac{1}{\epsilon}$ | $\kappa_c \leqslant \frac{1}{4}$ (weakly coupled) |
| EG/OGDA (Mokhtari et al., 2020) | $(\kappa_u + \kappa_v) \log \frac{1}{\epsilon}$ | $(\kappa_u + \kappa_v) \log \frac{1}{\epsilon}$ | $\kappa_c \leqslant \frac{1}{2}\sqrt{1 - \frac{1}{\max\{\kappa_u \kappa_v\}}}$ |
| APPA (Lin et al., 2020) | $\sqrt{\kappa_u \kappa_v} \log^3 \frac{1}{\epsilon}$ | $\sqrt{\kappa_u \kappa_v} \log^3 \frac{1}{\epsilon}$ | $\kappa_c \leqslant \frac{1}{2}\sqrt{1 - \frac{1}{\sqrt{\kappa_u \kappa_v}}}$ |
| FOAM (Kovalev & Gasnikov, 2022) | $\sqrt{\kappa_u \kappa_v} \log \frac{1}{\epsilon}$ | $\sqrt{\kappa_u \kappa_v} \log \frac{1}{\epsilon}$ | $\kappa_c \leqslant \frac{1}{2}\sqrt{1 - \frac{1}{\sqrt{\kappa_u \kappa_v}}}$ |
| PEARL-SGD (Yoon et al., 2025) | $\kappa^2 \log \frac{1}{\epsilon}$ | $\kappa^2 \log \frac{1}{\epsilon}$ | $\kappa_c \leqslant \frac{1}{4}$ (weakly coupled) |
| **Decoupled SGDA (ours)** | **0** | $\min\left\{ \frac{1}{1-4\kappa_c} \log \frac{1}{\epsilon}, \ \kappa^2 \log \frac{1}{\epsilon} \right\}$ | - |

*Moreover, For any $R, K \geqslant 1$, after running Decoupled SGDA for a total of $T = KR$ iterations on a function $f$, with the stepsize $\gamma \leqslant \min\left\{ \frac{\mu}{L^2}, \frac{\mu}{KLL_c} \right\}$ in the non-weakly coupled regime, we get a rate of:*

$$\mathbb{E}\left[\|\mathbf{x}_K^R - \mathbf{x}^\star\|^2\right] \leqslant D^2 \exp\left(-\frac{\gamma\mu}{2} KR\right) + \frac{2\bar{\sigma}^2 \gamma}{\mu}.$$

*where $D = \|\mathbf{x}_0 - \mathbf{x}^\star\|$.*

**Corollary 4.2.** *Decoupled GDA with a stepsize of $\gamma = \frac{\bar{\mu}}{L^2}$ converges to the saddle point **without** any communication on fully decoupled games ($\kappa_c = 0$) if $K \to \infty$.*

**Fully Decoupled Games.** For the sake of comparison, we define the condition numbers[3]: $\kappa_u = \frac{L_u}{\mu_u}$, $\kappa_v = \frac{L_v}{\mu_v}$, and $\kappa_{uv} = \kappa_{vu} = \frac{L_c}{\sqrt{\mu_u \mu_v}}$. We also use $\kappa = \frac{L}{\mu}$. The most recent rate proposed for GDA (Lee et al., 2024) requires $\mathcal{O}\left((\kappa_u + \kappa_v) \log \frac{1}{\epsilon}\right)$ rounds of communication when the game is fully decoupled. A major drawback of GDA and several other common methods in this setting is that poor conditioning in one of the players (large $\kappa_u, \kappa_v$) significantly increases the number of communication rounds. In contrast, our method overcomes this issue by utilizing local steps, effectively eliminating the dependency on players' conditioning.

**Corollary 4.3.** *With the choice of $\gamma = \frac{\bar{\mu}}{RL^2}$ if the game is weakly coupled we get:*

$$\mathbb{E}\left[\|\mathbf{x}_K^R - \mathbf{x}^\star\|^2\right] \leqslant D^2 \exp\left(-(1-4\kappa_c)R\right) + \frac{8\bar{\sigma}^2 \bar{\mu} \kappa_c}{R\mu L^2 (1-4\kappa_c)}.$$

*Consequently, to reach $\mathbb{E}\left[\|\mathbf{x}_K^R - \mathbf{x}^\star\|^2\right] \leqslant \epsilon$, it suffices to perform $R = \max\left\{ \frac{1}{4-\kappa_c} \log\left(\frac{2D^2}{\epsilon}\right), \frac{16\bar{\mu}\kappa_c \bar{\sigma}^2}{\mu L^2 (1-4\kappa_c)\epsilon} \right\}$ rounds*

---

[3] $\|\nabla_u f(\mathbf{u}, \mathbf{v}) - \nabla_u f(\mathbf{u}', \mathbf{v})\| \leqslant L_u \|\mathbf{u} - \mathbf{u}'\|$ and $\|\nabla_v f(\mathbf{u}, \mathbf{v}) - \nabla_v f(\mathbf{u}, \mathbf{v}')\| \leqslant L_v \|\mathbf{v} - \mathbf{v}'\|$

*with $K = \frac{L^2}{\mu\bar{\mu}} \log(\frac{4}{\kappa_c})$. Moreover, with the choice of $\gamma = \min\left\{ \frac{\mu}{32KL^2}, \frac{1}{\mu KR} \log(\max\{2, \frac{\mu^2 D^2}{\bar{\sigma}^2} KR\}) \right\}$ if the game is not weakly coupled we get:*

$$\mathbb{E}\left[\|\mathbf{x}_K^R - \mathbf{x}^\star\|^2\right] \leqslant D^2 \exp\left(-\frac{\mu^2}{2L^2} R\right) + \frac{\bar{\sigma}^2}{\mu^2 KR}.$$

*Consequently, to reach $\mathbb{E}\left[\|\mathbf{x}_K^R - \mathbf{x}^\star\|^2\right] \leqslant \epsilon$, it suffices to perform $R = \frac{2L^2}{\mu^2} \log(\frac{D^2}{\epsilon})$ with $K = \frac{2\bar{\sigma}^2}{\mu^2 \epsilon}$.*

**Weakly and Non-Weakly Coupled Games.** The main property of our rate for weakly coupled games is the **absence** of $\kappa_u, \kappa_v$, or $\kappa$, which can be very large even if the player interaction is low. We are able to capture this effect due to differentiating between different smoothness parameters. In addition, mathematically identifying the regime in which we can benefit from low interaction and achieve communication acceleration is another important aspect of our work. This stands in contrast to most popular methods, whose communication complexity always depends on the quantities $\kappa_u, \kappa_v$ or $\kappa$ (see Table 1), which can be overly pessimistic, especially in the weakly coupled regime. Moreover, for non-weakly coupled games, our rate recovers the standard $\mathcal{O}(\kappa^2 \log(1/\epsilon))$ rate for GDA from (Zhang et al., 2022a; Azizian et al., 2020).

**Noise Term.** Our method does not depend on $\sigma_{uv}$ or $\sigma_{vu}$, allowing them to be arbitrarily large. In contrast, existing federated minimax methods assume these quantities are bounded, which may not hold in many real-world settings. In the weakly coupled regime, $\bar{\sigma}$ is multiplied by $\kappa_c$, a small quantity, reducing the effect of noise. In the non-weakly coupled regime, we can mitigate noise by taking more local steps. Herein, we state the communication complexity of our method and compare it with GDA as the baseline.

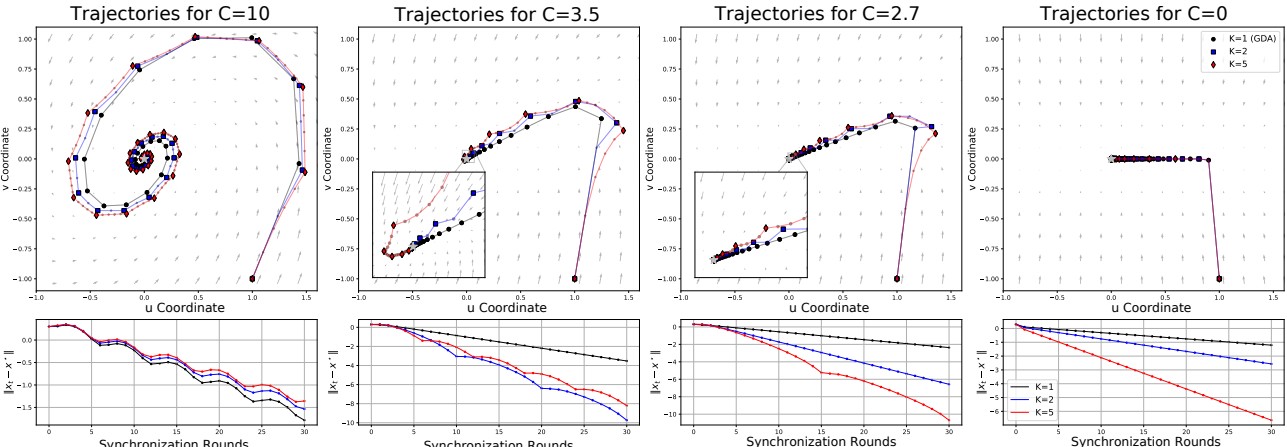

*Figure 1.* **Trajectories (top row) and distance to equilibrium over synchronization rounds (bottom row) of GDA ($K = 1$) and Decoupled SGDA with $K = \{2, 5\}$ on the** (13) **problem** ($d = 2$). **C** in (13) is a constant here—the larger, the stronger the interactive term. **Left-to-right:** decreasing the constant $c \in \{10, 3.5, 2, 7, 0\}$. The markers denote the local steps and star the solution. See § 5 for discussion.

**Corollary 4.4.** *For any $K \geqslant \frac{1}{\gamma\mu}\log\left(\frac{4}{\kappa_c}\right)$, after running Decoupled SGDA on a weakly coupled game, we have the following communication complexity in order to achieve $\epsilon$ accuracy in the noiseless setting:*

| Decoupled GDA | | GDA |
|:---:|:---:|:---:|
| $\mathcal{O}\left(\dfrac{1}{1-4\kappa_c}\log\dfrac{1}{\epsilon}\right)$ | *vs.* | $\mathcal{O}\left((\kappa_u + \kappa_v + \kappa_{uv}^2)\log\dfrac{1}{\epsilon}\right)$ |

*Moreover, Decoupled SGDA in the weakly coupled regime **always** has better communication complexity compared to the baseline GDA. In other words, $\frac{1}{1-4\kappa_c} \leqslant \kappa_u + \kappa_v + \kappa_{uv}^2$.*

Table 1 compares our method with other first-order methods in terms of **communication complexity** in both the fully decoupled and weakly coupled regimes. It is clear that in the fully decoupled regime, our method outperforms all other methods. Furthermore, it is expected to compare our method with GDA by considering it as the baseline because our method uses GD local updates (and not updates using EG or momentum). In Corollary 4.4, we stated that we always have a better complexity compared to GDA in the weakly coupled regime. However, we can show that under a slightly stronger assumption, our method achieves better communication complexity than the optimal first-order method for solving SCSC games.

**Corollary 4.5.** *For any SCSC zero-sum minimax game with coupling degree $\kappa_c \leqslant \frac{1}{2}\sqrt{1 - \frac{1}{\sqrt{\kappa_u\kappa_v}}}$, our method achieves a better communication complexity than FOAM which is the optimal first-order method for solving SCSC games. In another word, if $\frac{1}{1-4\kappa_c} \ll \sqrt{\kappa_u\kappa_v}$, our method achieves significant communication acceleration compared to FOAM.*

Corollary 4.5 shows that our method can even outperform the optimal first-order method in terms of communication

rounds. The assumption $\kappa_c \leqslant \frac{1}{2}\sqrt{1 - \frac{1}{\sqrt{\kappa_u\kappa_v}}}$ can recover the weakly coupled condition if $\max\{\kappa_u, \kappa_v\} \to \infty$. Although the rate $\mathcal{O}(\sqrt{\kappa_u\kappa_v}\log(1/\epsilon))$ is optimal and matches the lower bound from (Zhang et al., 2022b), large condition number of players ($\kappa_u, \kappa_v$) can increase the communication overhead significantly.

*Remark* 4.6. The recent work of Yoon et al. (2025) considers a similar method in the general $N$-player setting (we also address this case in Appendix C) and establishes a linear convergence rate of $\mathcal{O}\left(\left(\frac{\ell + L_{\max}\sqrt{\ell/\mu}}{\mu}\right)\log(1/\epsilon)\right)$, where $\ell$ denotes the star-cocoercivity constant and $L_{\max} = \max\{L_1, \ldots, L_N\}$ is the largest smoothness parameter among the players. For the class of $\mu$-strongly monotone and $L$-Lipschitz continuous operators considered in our work, it holds that $L_{\max}\sqrt{\ell/\mu} \leqslant \ell$ (see Yoon et al. 2025, Appendix D), which simplifies their rate to $\mathcal{O}\left(\frac{\ell}{\mu}\log(1/\epsilon)\right)$. In general, one has the bound $\ell \leqslant \frac{L^2}{\mu}$ (Facchinei & Pang, 2003), which implies a worst-case convergence rate of $\mathcal{O}(\kappa^2\log(1/\epsilon))$, offering no communication acceleration. While acceleration is theoretically possible when $\ell$ is small, the improvement is limited: since $L \leqslant \ell$, the best achievable rate is $\mathcal{O}(\kappa\log(1/\epsilon))$. In the fully decoupled setting, this matches the convergence rate of GDA (Lee et al., 2024), which, as discussed earlier, is suboptimal. By contrast, our method requires only a single round of communication in this regime (see Corollary 4.2), demonstrating a significant advantage. A key distinction between the two works is in how interaction between players is modeled. Our analysis introduces and leverages the coupling parameter $L_c$, which more directly captures the interaction structure and leads to sharper communication complexity bounds.

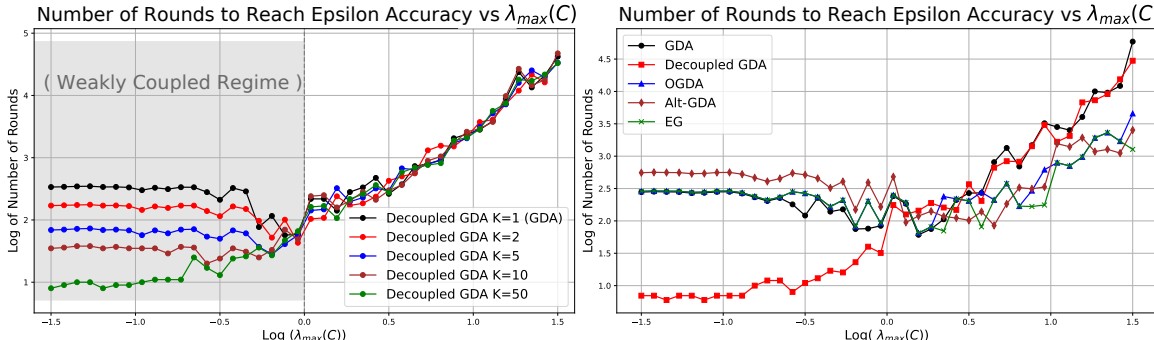

*Figure 2.* **Number of rounds (log-scale; lower is better) to reach epsilon accuracy for varying $\lambda_{\max}(\mathbf{C})$ in** (13). **Left:** Decoupled GDA with different $K$-values and GDA ($K = 1$). **Right:** comparison between GDA, Decoupled GDA, Optimistic GDA (Popov, 1980), Alternating GDA and Extragradient (Korpelevich, 1976).

## 5. Experiments

In this section, we evaluate the empirical performance of Decoupled GDA. For all experiments described in this section, we provide additional implementation details (and hyperparameters) in Appendix I.

### 5.1. Quadratic Games

Herein, we consider the following problem class:

$$\min_{\mathbf{u}} \max_{\mathbf{v}} \quad \frac{1}{2}\langle \mathbf{u}, \mathbf{A}\mathbf{u}\rangle - \frac{1}{2}\langle \mathbf{v}, \mathbf{B}\mathbf{v}\rangle + \langle \mathbf{u}, \mathbf{C}\mathbf{v}\rangle, \quad (13)$$

where $\mathbf{u}, \mathbf{v} \in \mathbb{R}^{\frac{d}{2}}$, and $\mathbf{A}, \mathbf{B}, \mathbf{C}$ are $\frac{d}{2} \times \frac{d}{2}$ positive definite matrices. We will use varying $\mathbf{C}$ to control the players' interaction.

Figure 1 illustrates the performances of Decoupled SGDA on the (13) for varying numbers of local steps $K$ and different intensities of the interactive term of (13). The results show that as the interactive term weakens, Decoupled SGDA converges more quickly than the GDA baseline ($K = 1$). Additionally, with a stronger interactive term, increasing the number of local steps $K$ leads to faster convergence for the same number of synchronization rounds. Figure 2 depicts the performances over a spectrum of payoff functions controlled by the constant matrix $\mathbf{C}$ in (13). In the Weakly Coupled Game regime, highlighted by shading, Decoupled SGDA outperforms the baseline GDA. In Figure 2 (right), we compare it with other optimization methods, demonstrating that Decoupled SGDA achieves similar results with significantly fewer communication rounds in the weakly coupled regime.

### 5.2. Communication Efficiency For Non-convex Functions

While our theoretical focus was on SCSC games, in this section, we explore if our insights extend to broader problem instances. We focus on a *Toy GAN* non-convex game as

follows:

$$\min_{\mathbf{u}} \max_{\mathbf{v}} \big\{ \mathbb{E}_{\phi \sim \mathcal{N}(0,\Sigma)}[\phi^T \mathbf{v}\phi] - \\ \mathbb{E}_{\phi \sim \mathcal{N}(0,1)}[(\mathbf{u}\phi)^T \mathbf{v}(\mathbf{u}\phi)] + \lambda_1 \|\mathbf{u}\|^2 - \lambda_2 \|\mathbf{v}\|^2 \big\}, \quad (14)$$

where $\mathbf{u} \in \mathbb{R}^{d_1}$, $\mathbf{v} \in \mathbb{R}^{d_2}$. Figure 3 shows the smallest gradient norm (lower is better) each algorithm can achieve for a fixed number of communication rounds, with varying values of $1/\lambda$. As $\lambda$ decreases, the regularization terms dominate, making the game less interactive (similar to the weakly coupled regime). When $\lambda$ increases, reducing interaction, Decoupled GDA achieves a much lower gradient norm with the same number of communication rounds. This demonstrates that Decoupled GDA efficiently solves non-convex problems in settings analogous to the weakly coupled regime by leveraging local updates to reduce communication. This experiment highlights the method's capabilities beyond SCSC games. The trajectory of Decoupled GDA iterations for this non-convex minimax problem can be found in Appendix H.1. **Decoupled SGDA with gradient approximation.** Herein, we compare *Decoupled SGDA* with Federated Minimax, aka (8). We study environments with gradient oracles with *unbalanced noise* unbalanced noise. Each player has access to a gradient oracle that provides low-variance noise for their own gradients but high-variance noise for the remaining players. In the quadratic game introduced earlier, each oracle adding zero-mean Gaussian noise to the gradient. The variance differs between gradients for a player's own parameters (diagonal variance) and those of others (off-diagonal variance). Equation 7 formalizes this. In both experiments, we kept the diagonal variance $(\sigma_{uu}^2, \sigma_{vv}^2)$ constant, while varying the off-diagonal variance $(\sigma_{vu}^2, \sigma_{uv}^2)$ in the second experiment. Figure 4 compares *Decoupled SGDA* and *Local SGDA*, the latter being the most commonly used method for federated minimax problems (Deng & Mahdavi, 2021). It depicts the smallest gradient norm each algorithm achieves within a fixed number of communication rounds across different scenarios.

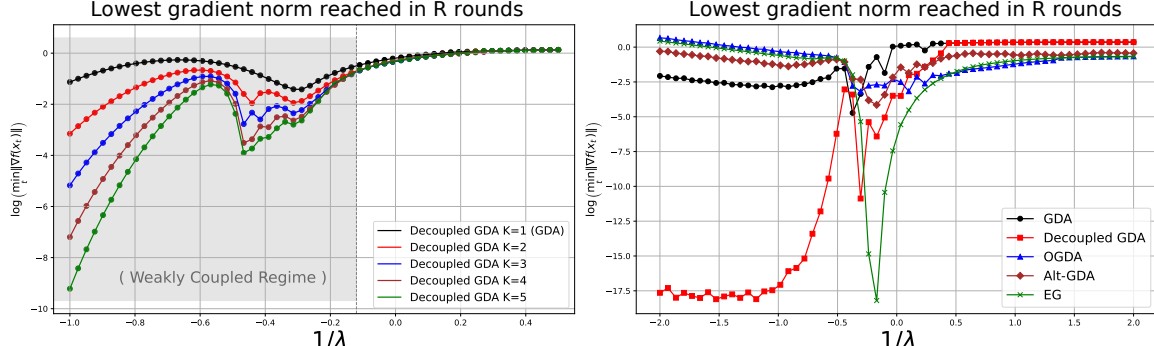

*Figure 3.* **Lowest gradient norm reached after a fixed number of communication rounds, for varying** $1/\lambda$ **in** (14). **Left:** Effect of $K$. **Right:** different optimization methods, GDA, Decoupled GDA, Optimistic GDA (Popov, 1980), ALT–alternating GDA and Extragradient (Korpelevich, 1976). See § 5 for discussion.

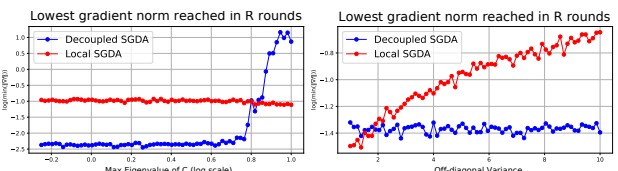

*Figure 4.* **Lowest gradient norm reached by Decoupled SGDA and Local SGDA for a fixed number of communication rounds using unbalanced noisy gradient oracles.** **Left:** *Decoupled SGDA* vs. *Federated Minimax* for varying values of $\|\mathbf{C}\|$ and fixed variance. **Left:** *Decoupled SGDA* vs. *Local SGDA* for varying levels of off-diagonal variance noise ($\sigma_{uv}$, $\sigma_{vu}$). See 5.

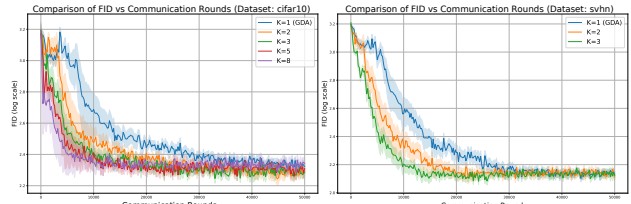

*Figure 5.* $y$-**axis: FID scores (log scale; lower is better) during GAN training, versus** $x$-**axis communication rounds.** **Left:** results on the *CIFAR-10* (Krizhevsky, 2009) dataset. **Right:** results on the *SVHN* (Netzer et al., 2011) dataset.

The left plot demonstrates how both methods perform in games with varying levels of interaction. When the interaction is weaker, Decoupled SGDA achieves significantly lower gradient norms with the same number of communication rounds. The right plot highlights the effect of noise variance, showing that while high noise negatively impacts Local SGDA, it has minimal to no effect on Decoupled SGDA. In the presence of imbalanced noise, the results suggest that switching from local SGDA to Decoupled SGDA is beneficial, even for highly interactive games.

### 5.3. Communication Efficiency in GAN Training.

Figure 5 compares Decoupled SGDA with baseline methods in terms of FID score over communication rounds. The results show that Decoupled SGDA converges faster and requires fewer communication rounds than standard GDA and its variants. This advantage is particularly evident on the CIFAR-10 and SVHN datasets, where increasing the number of local steps (K) leads to lower FID scores. These findings highlight the efficiency of our approach in reducing communication overhead while maintaining strong performance in complex, non-convex tasks such as GAN training.

## 6. Conclusion

We proposed *Decoupled SGDA*, an effective optimization method for games with intermittent strategy communication, particularly suited for settings with weak interaction between players or imbalanced noise levels. Our theoretical and empirical results show that Decoupled SGDA outperforms traditional methods like Local SGDA in terms of communication efficiency and robustness, and extends beyond strongly-convex strongly-concave (SCSC) games to non-convex settings. Its adaptability to varying interaction and noise makes it a valuable tool for federated and decentralized optimization.

Several future directions remain. One could explore varying $K$ across players or adapting other game optimization methods (e.g., Extra-Gradient) to broader settings without requiring strong convexity. Moreover, the decoupled updates naturally lend themselves to privacy-sensitive applications, as players do not need direct access to others' parameters, reducing risks from gradient sharing (see Zhu et al., 2019; Zhao et al., 2020; Wei et al., 2020, and references therein). Future work could investigate privacy-preserving extensions and further develop Ghost-SGDA to enhance performance in decentralized and privacy-constrained environments.

## Impact Statement

This work is purely theoretical with no direct societal impact. However, our findings can improve communication efficiency in decentralized systems, such as federated learning and multi-agent reinforcement learning, by reducing communication complexity.

## Acknowledgments

TC was supported by the FAIR (Future Artificial Intelligence Research) project, funded by the NextGenerationEU program within the PNRR-PE-AI scheme (M4C2, Investment 1.3, Line on Artificial Intelligence).

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

# Table of Contents

## A. Summary of Parameters

*Table 2.* Summary of Symbols for Two-Player Games

| Symbol | Definition | Mathematical Definition |
|--------|------------|------------------------|
| $L$ | Smoothness parameter for operator $F(\mathbf{x})$ | $\|F(\mathbf{x}) - F(\mathbf{x}')\|_* \leqslant L \|\mathbf{x} - \mathbf{x}'\|$ |
| $\bar{L}$ | Smoothness parameter for operator $F_{\bar{\mathbf{x}}}(\mathbf{x})$ | $\|F_{\bar{\mathbf{x}}}(\mathbf{x}) - F_{\bar{\mathbf{x}}}(\mathbf{x}')\|_* \leqslant \bar{L} \|\mathbf{x} - \mathbf{x}'\|$ |
| $L_c$ | Difference between $F_{\bar{\mathbf{x}}}(\mathbf{x})$ and $F(\mathbf{x})$ | $\|F_{\bar{\mathbf{x}}}(\mathbf{x}) - F(\mathbf{x})\|_* \leqslant L_c \|\bar{\mathbf{x}} - \mathbf{x}\|$ |
| $L_u$ | Smoothness parameter with respect to $\mathbf{u}$ | $\|\nabla_u f(\mathbf{u}, \mathbf{v}) - \nabla_u f(\mathbf{u}', \mathbf{v})\|_{u,*} \leqslant L_u \|\mathbf{u} - \mathbf{u}'\|_u$ |
| $L_v$ | Smoothness parameter with respect to $\mathbf{v}$ | $\|\nabla_v f(\mathbf{u}, \mathbf{v}) - \nabla_v f(\mathbf{u}, \mathbf{v}')\|_{v,*} \leqslant L_v \|\mathbf{v} - \mathbf{v}'\|_v$ |
| $L_{uv}$ | Interaction smoothness parameter | $\|\nabla_u f(\mathbf{u}, \mathbf{v}) - \nabla_u f(\mathbf{u}, \mathbf{v}')\|_{u,*} \leqslant L_{uv} \|\mathbf{v} - \mathbf{v}'\|_v$ |
| $L_{vu}$ | Interaction smoothness parameter for $v$ with respect to $u$ | $\|\nabla_v f(\mathbf{u}, \mathbf{v}) - \nabla_v f(\mathbf{u}', \mathbf{v})\|_{v,*} \leqslant L_{vu} \|\mathbf{u} - \mathbf{u}'\|_u$ |
| $\mu_u$ | Strong convexity parameter for $\mathbf{u}$ | $f(\mathbf{u}', \mathbf{v}) \geqslant f(\mathbf{u}, \mathbf{v}) + \langle \nabla_u f(\mathbf{u}, \mathbf{v}), \mathbf{u}' - \mathbf{u} \rangle + \frac{\mu_u}{2} \|\mathbf{u}' - \mathbf{u}\|_u^2$ |
| $\mu_v$ | Strong concavity parameter for $\mathbf{v}$ | $f(\mathbf{u}, \mathbf{v}') \leqslant f(\mathbf{u}, \mathbf{v}) + \langle \nabla_v f(\mathbf{u}, \mathbf{v}), \mathbf{v}' - \mathbf{v} \rangle - \frac{\mu_v}{2} \|\mathbf{v}' - \mathbf{v}\|_v^2$ |
| $\bar{\mu}$ | Strong monotonicity parameter for $F_{\bar{\mathbf{x}}}(\mathbf{x})$ | $\langle F_{\bar{\mathbf{x}}}(\mathbf{x}) - F_{\bar{\mathbf{x}}}(\mathbf{x}'), \mathbf{x} - \mathbf{x}' \rangle \geqslant \bar{\mu} \|\mathbf{x} - \mathbf{x}'\|^2$ |
| $\mu$ | Strong monotonicity parameter for $F(\mathbf{x})$ | $\langle F(\mathbf{x}) - F(\mathbf{x}'), \mathbf{x} - \mathbf{x}' \rangle \geqslant \mu \|\mathbf{x} - \mathbf{x}'\|^2$ |
| $\kappa_c$ | Coupling degree of the game | $\frac{L_c}{\bar{\mu}}$ |

*Table 3.* Summary of Symbols for $N$-Player Games

| Symbol | Definition | Mathematical Definition |
|--------|------------|------------------------|
| $\hat{L}_n$ | Upper bound for diagonal elements $L_{ii}$ | $\|\nabla_n f_n(\mathbf{x}) - \nabla_n f_n(\mathbf{x} + \mathbf{U}_n \mathbf{h}_n)\|_{n,*} \leqslant \hat{L}_n \|\mathbf{h}_n\|_n$ |
| $\bar{L}_n$ | Upper bound for off-diagonal elements $L_{ij}$ for $i \neq j$ | $\|\nabla_n f_n(\mathbf{x}) - \nabla_n f_n(\mathbf{x} + \sum_{i \neq n} \mathbf{U}_i \mathbf{h}_i)\|_{n,*} \leqslant \bar{L}_n \|\sum_{i \neq n} \mathbf{U}_i \mathbf{h}_i\|$ |
| $L_c$ | Difference between $F(\mathbf{x})$ and $F_{\bar{\mathbf{x}}}(\mathbf{x})$ | $\|F(\mathbf{x}) - F_{\bar{\mathbf{x}}}(\mathbf{x})\|_* \leqslant L_c \|\mathbf{x} - \bar{\mathbf{x}}\|$ |
| $\bar{L}$ | Smoothness parameter for operator $F_{\bar{\mathbf{x}}}(\mathbf{x})$ | $\|F_{\bar{\mathbf{x}}}(\mathbf{x}) - F_{\bar{\mathbf{x}}}(\mathbf{x}')\|_* \leqslant \bar{L} \|\mathbf{x} - \mathbf{x}'\|$ |
| $L$ | Smoothness parameter for operator $F(\mathbf{x})$ | $\|F(\mathbf{x}) - F(\mathbf{x}')\|_* \leqslant L \|\mathbf{x} - \mathbf{x}'\|$ |
| $\bar{\mu}$ | Strong monotonicity parameter for $F_{\bar{\mathbf{x}}}(\mathbf{x})$ | $\langle F_{\bar{\mathbf{x}}}(\mathbf{x}) - F_{\bar{\mathbf{x}}}(\mathbf{x}'), \mathbf{x} - \mathbf{x}' \rangle \geqslant \bar{\mu} \|\mathbf{x} - \mathbf{x}'\|^2$ |
| $\mu$ | Strong monotonicity parameter for $F(\mathbf{x})$ | $\langle F(\mathbf{x}) - F(\mathbf{x}'), \mathbf{x} - \mathbf{x}' \rangle \geqslant \mu \|\mathbf{x} - \mathbf{x}'\|^2$ |
| $\mu_n$ | Strong convexity parameter for $f_n(\mathbf{x})$ | $\langle \nabla_n f_n(\mathbf{x}) - \nabla_n f_n(\mathbf{x} + \mathbf{U}_n \mathbf{d}_n), \mathbf{x}^n - \mathbf{x}'^n \rangle \geqslant \mu_n \|\mathbf{x}'^n - \mathbf{x}^n\|_n^2$ |
| $\kappa_c$ | Coupling degree of the game | $\frac{L_c}{\bar{\mu}}$ |

## B. Missing Proofs for Section 4

**Lemma B.1.** *Let $\{r_t\}_{t>0}$ be a sequence of numbers satisfying:*

$$r_{t+1} \leqslant (1 - a\gamma)r_t + \gamma b$$

*for constants $a, \gamma, b > 0$ assuming $a\gamma < 1$. After unrolling the recursion $K$ times we get:*

$$r_K \leqslant (1 - a\gamma)^K + \frac{b}{a} \tag{15}$$

*Proof.*

$$r_K \leqslant (1 - a\gamma)^K + \sum_{i=0}^{K-1} (1 - a\gamma)^i \gamma b$$

$$\leqslant (1 - a\gamma)^K + \sum_{i=0}^{\infty} (1 - a\gamma)^i \gamma b \leqslant (1 - a\gamma)^K + \frac{b}{a}$$

$\square$

**Lemma B.2.** *For the parameters $L, \bar{L}$ and $L_c$ from Assumptions 2.2 and 2.3, we can say $L_c \leqslant L$ and $\bar{L} \leqslant L$.*

*Proof.* Recall that:

$$\|F(\mathbf{x}) - F(\mathbf{x}')\|_* \leqslant L\|\mathbf{x} - \mathbf{x}'\|$$
$$\|F_{\bar{\mathbf{x}}}(\mathbf{x}) - F_{\bar{\mathbf{x}}}(\mathbf{x}')\|_* \leqslant \bar{L}\|\mathbf{x} - \mathbf{x}'\|$$
$$\|F_{\bar{\mathbf{x}}}(\mathbf{x}) - F(\mathbf{x})\|_* \leqslant L_c\|\mathbf{x} - \bar{\mathbf{x}}\|$$

we start with the definition of the $\|F_{\bar{\mathbf{x}}}(\mathbf{x}) - F(\mathbf{x})\|_*$:

$$\|F_{\bar{\mathbf{x}}}(\mathbf{x}) - F(\mathbf{x})\|_*^2 = \frac{1}{\alpha_u}\|\nabla_u f(\mathbf{u}, \mathbf{v}) - \nabla_u f(\mathbf{u}, \bar{\mathbf{v}})\|_{u,*}^2 + \frac{1}{\alpha_v}\|\nabla_v f(\mathbf{u}, \mathbf{v}) - \nabla_v f(\bar{\mathbf{u}}, \mathbf{v})\|_{v,*}^2$$

$$\leqslant \frac{1}{\alpha_u}\|\nabla_u f(\mathbf{u}, \mathbf{v}) - \nabla_u f(\mathbf{u}, \bar{\mathbf{v}})\|_{u,*}^2 + \frac{1}{\alpha_v}\|\nabla_v f(\mathbf{u}, \mathbf{v}) - \nabla_v f(\mathbf{u}, \bar{\mathbf{v}})\|_{v,*}^2$$

$$+ \frac{1}{\alpha_v}\|\nabla_v f(\mathbf{u}, \mathbf{v}) - \nabla_v f(\bar{\mathbf{u}}, \mathbf{v})\|_{v,*}^2 + \frac{1}{\alpha_u}\|\nabla_u f(\mathbf{u}, \mathbf{v}) - \nabla_u f(\bar{\mathbf{u}}, \mathbf{v})\|_{u,*}^2$$

$$= \|F(\mathbf{u}, \mathbf{v}) - F(\mathbf{u}, \bar{\mathbf{v}})\|_*^2 + \|F(\mathbf{u}, \mathbf{v}) - F(\bar{\mathbf{u}}, \mathbf{v})\|_*^2$$

$$\leqslant L^2\alpha_v\|\mathbf{v} - \bar{\mathbf{v}}\|_v^2 + L^2\alpha_u\|\mathbf{u} - \bar{\mathbf{u}}\|_u^2 = L^2\|\mathbf{x} - \bar{\mathbf{x}}\|^2$$

which means that we can upper bound $\|F_{\bar{\mathbf{x}}}(\mathbf{x}) - F(\mathbf{x})\|_*^2$ by the constant $L$ at the worst case. However, the constant $L_c$ that we use can be much smaller. Next, for the the inequality $\|F_{\bar{\mathbf{x}}}(\mathbf{x}) - F_{\bar{\mathbf{x}}}(\mathbf{x}')\|_*$ we have:

$$\|F_{\bar{\mathbf{x}}}(\mathbf{x}) - F_{\bar{\mathbf{x}}}(\mathbf{x}')\|_* = \frac{1}{\alpha_u}\|\nabla_u f(\mathbf{u}, \bar{\mathbf{v}}) - \nabla_u f(\mathbf{u}', \bar{\mathbf{v}})\|_{u,*}^2 + \frac{1}{\alpha_v}\|\nabla_v f(\bar{\mathbf{u}}, \mathbf{v}) - \nabla_v f(\bar{\mathbf{u}}, \mathbf{v}')\|_{v,*}^2$$

$$\leqslant \frac{1}{\alpha_u}\|\nabla_u f(\mathbf{u}, \bar{\mathbf{v}}) - \nabla_u f(\mathbf{u}', \bar{\mathbf{v}})\|_{u,*}^2 + \frac{1}{\alpha_v}\|\nabla_v f(\mathbf{u}, \bar{\mathbf{v}}) - \nabla_v f(\mathbf{u}', \bar{\mathbf{v}})\|_{v,*}^2$$

$$+ \frac{1}{\alpha_v}\|\nabla_v f(\bar{\mathbf{u}}, \mathbf{v}) - \nabla_v f(\bar{\mathbf{u}}, \mathbf{v}')\|_{v,*}^2 + \frac{1}{\alpha_u}\|\nabla_u f(\bar{\mathbf{u}}, \mathbf{v}) - \nabla_u f(\bar{\mathbf{u}}, \mathbf{v}')\|_{u,*}^2$$

$$= \|F(\mathbf{u}, \bar{\mathbf{v}}) - F(\mathbf{u}', \bar{\mathbf{v}})\|_*^2 + \|F(\bar{\mathbf{u}}, \mathbf{v}) - F(\bar{\mathbf{u}}, \mathbf{v}')\|_*^2$$

$$\leqslant L^2\alpha_u\|\mathbf{u} - \mathbf{u}'\|_u^2 + L^2\alpha_v\|\mathbf{v} - \mathbf{v}'\|_v^2 = L^2\|\mathbf{x} - \mathbf{x}'\|$$

which means we can upper bound $\|F_{\bar{\mathbf{x}}}(\mathbf{x}) - F_{\bar{\mathbf{x}}}(\mathbf{x}')\|_*$ by the constant $L$ in the worst case. However, the constant $\bar{L}$ can be much smaller. $\square$

**Lemma B.3.** *Given a SCSC game $f(\mathbf{u}, \mathbf{v})$. If the parameter $L_c$ from Assumption 2.3 is zero, the game is fully decoupled and players do not interact.*

*Proof.* Recall that a game can be expressed as:

$$f(\mathbf{u}, \mathbf{v}) = g(\mathbf{u}) - h(\mathbf{v}) + r(\mathbf{u}, \mathbf{v}).$$

We only need to show that $L_c = 0$ implies $r(\mathbf{u}, \mathbf{v}) = 0$. This means that each player's payoff is affected only by their own strategy. Also, recall that $L_c = \frac{1}{\sqrt{\alpha_u \alpha_v}} \max\{L_{uv}, L_{vu}\}$, where $L_{uv}$ and $L_{vu}$ are defined as follows:

$$\|\nabla_u f(\mathbf{u}, \mathbf{v}) - \nabla_u f(\mathbf{u}, \mathbf{v}')\|_{u,*} \leqslant L_{uv}\|\mathbf{v} - \mathbf{v}'\|_v,$$
$$\|\nabla_v f(\mathbf{u}, \mathbf{v}) - \nabla_v f(\mathbf{u}', \mathbf{v})\|_{v,*} \leqslant L_{vu}\|\mathbf{u} - \mathbf{u}'\|_u.$$

Since it is clear that if $L_c = 0$, then $L_{uv} = L_{vu} = 0$, the right-hand side of the above inequalities must be zero. Now, we rewrite the left-hand side of these inequalities in terms of the functions $g, h$, and $r$, which must also be zero:

$$\|\nabla_u f(\mathbf{u}, \mathbf{v}) - \nabla_u f(\mathbf{u}, \mathbf{v}')\|_{u,*} = \|\nabla_u g(\mathbf{u}) + \nabla_u r(\mathbf{u}, \mathbf{v}) - \nabla_u g(\mathbf{u}) - \nabla_u r(\mathbf{u}, \mathbf{v}')\|_{u,*}$$
$$= \|\nabla_u r(\mathbf{u}, \mathbf{v}) - \nabla_u r(\mathbf{u}, \mathbf{v}')\|_{u,*}.$$

For the above expression to be zero, either $r(\mathbf{u}, \mathbf{v}) = 0$ or $\nabla_u r(\mathbf{u}, \mathbf{v}) = \nabla_u r(\mathbf{u}, \mathbf{v}')$ must hold. However, the latter is impossible, as it implies that $r(\mathbf{u}, \mathbf{v})$ depends only on $\mathbf{u}$, which contradicts our assumption (if a term depends **only** on $\mathbf{u}$, it is already captured in $g(\mathbf{u})$). The same thing can be shown in the same way with respect to player $\mathbf{v}$. $\qquad\square$

**Lemma B.4.** *Let* $\bar{\mathbf{x}}, \mathbf{x}', \mathbf{x}^\star \in \mathcal{X}$ *be such that* $F_{\bar{\mathbf{x}}}(\mathbf{x}') = F(\mathbf{x}^\star) = 0$. *Then,*

$$\|\mathbf{x}' - \mathbf{x}^\star\| \leqslant \kappa_c \|\bar{\mathbf{x}} - \mathbf{x}^\star\|. \tag{16}$$

*Proof.* From Assumption 2.1 and the Cauchy–Schwarz inequality, it follows that $\|F_{\bar{\mathbf{x}}}(\mathbf{x}') - F_{\bar{\mathbf{x}}}(\mathbf{x}^\star)\| \geqslant \bar{\mu}\|\mathbf{x}' - \mathbf{x}^\star\|$. Hence,

$$\|\mathbf{x}' - \mathbf{x}^\star\| \leqslant \frac{1}{\bar{\mu}} \|F_{\bar{\mathbf{x}}}(\mathbf{x}') - F_{\bar{\mathbf{x}}}(\mathbf{x}^\star)\| = \frac{1}{\bar{\mu}} \|F(\mathbf{x}^\star) - F_{\bar{\mathbf{x}}}(\mathbf{x}^\star)\| \leqslant \frac{L_c}{\bar{\mu}} \|\bar{\mathbf{x}} - \mathbf{x}^\star\| = \kappa_c \|\bar{\mathbf{x}} - \mathbf{x}^\star\|. \qquad\square$$

**Lemma B.5.** *For any* $\bar{\mathbf{x}} \in \mathcal{X}$, *the operator* $F_{\bar{\mathbf{x}}}$ *is* $\bar{\mu}$-*strongly monotone with*

$$\bar{\mu} = \min\left\{\frac{\mu_u}{\alpha_u}, \frac{\mu_v}{\alpha_v}\right\}.$$

*Proof.* Recall that function $f$ is $\mu_u$-strongly convex in $\mathbf{u}$ and $\mu_v$-strongly concave in $\mathbf{v}$ meaning that:

$$\langle \nabla_u f(\mathbf{u}, \mathbf{v}) - \nabla_u f(\mathbf{u}', \mathbf{v}), \mathbf{u} - \mathbf{u}' \rangle \geqslant \mu_u \|\mathbf{u} - \mathbf{u}'\|_u^2$$
$$\langle \nabla_v f(\mathbf{u}, \mathbf{v}) - \nabla_v f(\mathbf{u}, \mathbf{v}'), \mathbf{v}' - \mathbf{v} \rangle \geqslant \mu_v \|\mathbf{v} - \mathbf{v}'\|_v^2$$

Therefore,

$$\langle F_{\bar{\mathbf{x}}}(\mathbf{x}) - F_{\bar{\mathbf{x}}}(\mathbf{x}'), \mathbf{x} - \mathbf{x}' \rangle$$
$$= \langle \nabla_u f(\mathbf{u}, \bar{\mathbf{v}}) - \nabla_u f(\mathbf{u}', \bar{\mathbf{v}}), \mathbf{u} - \mathbf{u}' \rangle + \langle \nabla_v f(\bar{\mathbf{u}}, \mathbf{v}) - \nabla_v f(\bar{\mathbf{u}}, \mathbf{v}'), \mathbf{v}' - \mathbf{v} \rangle$$
$$\geqslant \mu_u \|\mathbf{u} - \mathbf{u}'\|_u^2 + \mu_v \|\mathbf{v} - \mathbf{v}'\|_v^2 = \frac{\mu_u}{\alpha_u} \alpha_u \|\mathbf{u} - \mathbf{u}'\|_u^2 + \frac{\mu_v}{\alpha_v} \alpha_v \|\mathbf{v} - \mathbf{v}'\|_v^2$$
$$\geqslant \min\left\{\frac{\mu_u}{\alpha_u}, \frac{\mu_v}{\alpha_v}\right\} \|\mathbf{x} - \mathbf{x}'\|^2. \qquad\square$$

**Lemma B.6** (two-player). *For any* $\mathbf{x}, \bar{\mathbf{x}} \in \mathcal{X}$, *parameter* $L_c$ *can be expressed as:*

$$L_c = \frac{1}{\sqrt{\alpha_u \alpha_v}} \max\{L_{uv}, L_{vu}\}.$$

*Proof.* Recall that:

$$\|\nabla_u f(\mathbf{u}, \mathbf{v}) - \nabla_u f(\mathbf{u}, \mathbf{v}')\|_{u,*} \leqslant L_{uv} \|\mathbf{v} - \mathbf{v}'\|_v$$
$$\|\nabla_v f(\mathbf{u}, \mathbf{v}) - \nabla_v f(\mathbf{u}', \mathbf{v})\|_{v,*} \leqslant L_{vu} \|\mathbf{u} - \mathbf{u}'\|_u$$

Next we have:

$$\|F(\mathbf{x}) - F_{\bar{\mathbf{x}}}(\mathbf{x})\|_*^2 = \frac{1}{\alpha_u} \|\nabla_u f(\mathbf{u}, \mathbf{v}) - \nabla_u f(\mathbf{u}, \bar{\mathbf{v}})\|_{u,*}^2 + \frac{1}{\alpha_v} \|\nabla_v f(\mathbf{u}, \mathbf{v}) - \nabla_v f(\bar{\mathbf{u}}, \mathbf{v})\|_{v,*}^2$$
$$\leqslant \frac{L_{uv}^2}{\alpha_u} \|\mathbf{v} - \bar{\mathbf{v}}\|_v^2 + \frac{L_{vu}^2}{\alpha_v} \|\mathbf{u} - \bar{\mathbf{u}}\|_u^2$$
$$= \frac{L_{uv}^2}{\alpha_v \alpha_u} \alpha_v \|\mathbf{v} - \bar{\mathbf{v}}\|_v^2 + \frac{L_{vu}^2}{\alpha_u \alpha_v} \alpha_u \|\mathbf{u} - \bar{\mathbf{u}}\|_u^2$$
$$\leqslant \max\left\{\frac{L_{uv}^2}{\alpha_u \alpha_v}, \frac{L_{vu}^2}{\alpha_u \alpha_v}\right\} \left[\alpha_v \|\mathbf{v} - \bar{\mathbf{v}}\|_v^2 + \alpha_u \|\mathbf{u} - \bar{\mathbf{u}}\|_u^2\right] = L_c^2 \|\bar{\mathbf{x}} - \mathbf{x}\|^2. \qquad\square$$

**Lemma B.7** (two-player). *Let $\bar{\mathbf{x}}, \mathbf{x}_0^\star, \mathbf{x}^\star \in \mathcal{X}$ be such that $F_{\bar{\mathbf{x}}}(\mathbf{x}_0^\star) = 0$ and $F(\mathbf{x}^\star) = 0$. Then $\kappa_c$ can be expressed as:*

$$\kappa_c = \max\left\{ \sqrt{\frac{\alpha_u}{\alpha_v}}\frac{L_{uv}}{\mu_u}, \sqrt{\frac{\alpha_v}{\alpha_u}}\frac{L_{vu}}{\mu_v} \right\}. \tag{17}$$

*Proof.* Indeed,

$$
\begin{aligned}
\|\mathbf{x}_0^\star - \mathbf{x}^\star\|^2 &= \alpha_u \|\mathbf{u}' - \mathbf{u}^\star\|_u^2 + \alpha_v \|\mathbf{v}' - \mathbf{v}^\star\|_v^2 \\
&\leqslant \frac{\alpha_u}{\mu_u^2} \|\nabla_u f(\mathbf{u}', \bar{\mathbf{v}}) - \nabla_u f(\mathbf{u}^\star, \bar{\mathbf{v}})\|_{u,*}^2 + \frac{\alpha_v}{\mu_v^2} \|\nabla_v f(\bar{\mathbf{u}}, \mathbf{v}') - \nabla_v f(\bar{\mathbf{u}}, \mathbf{v}^\star)\|_{v,*}^2 \\
&= \frac{\alpha_u}{\mu_u^2} \|\nabla_u f(\mathbf{u}^\star, \mathbf{v}^\star) - \nabla_u f(\mathbf{u}^\star, \bar{\mathbf{v}})\|_{u,*}^2 + \frac{\alpha_v}{\mu_v^2} \|\nabla_v f(\mathbf{u}^\star, \mathbf{v}^\star) - \nabla_v f(\bar{\mathbf{u}}, \mathbf{v}^\star)\|_{v,*}^2 \\
&\leqslant \frac{\alpha_u L_{uv}^2}{\mu_u^2} \|\bar{\mathbf{v}} - \mathbf{v}^\star\|_v^2 + \frac{\alpha_v L_{vu}^2}{\mu_v^2} \|\bar{\mathbf{u}} - \mathbf{u}^\star\|_u^2 \\
&= \frac{\alpha_u L_{uv}^2}{\alpha_v \mu_u^2} \alpha_v \|\bar{\mathbf{v}} - \mathbf{v}^\star\|_v^2 + \frac{\alpha_v L_{vu}^2}{\alpha_u \mu_v^2} \alpha_u \|\bar{\mathbf{u}} - \mathbf{u}^\star\|_u^2 \\
&\leqslant \max\left\{ \frac{\alpha_u L_{uv}^2}{\alpha_v \mu_u^2}, \frac{\alpha_v L_{vu}^2}{\alpha_u \mu_v^2} \right\} \left[ \alpha_v \|\bar{\mathbf{v}} - \mathbf{v}^\star\|_v^2 + \alpha_u \|\bar{\mathbf{u}} - \mathbf{u}^\star\|_u^2 \right] = \kappa_c^2 \|\bar{\mathbf{x}} - \mathbf{x}^\star\|^2. \qquad \square
\end{aligned}
$$

### B.1. Proof of Theorem 4.1 in Weakly Coupled Regime

In this section, we provide the proof of convergence for our method in the **weakly coupled regime**. In contrast to the common assumptions and proof techniques, we utilize our new parameter $L_c$, which quantifies the level of interaction in the game. We demonstrate that if the fraction $\frac{L_c}{\bar{\mu}}$ is sufficiently small (i.e., the game is weakly coupled), we can achieve **communication acceleration**. This aspect is often overlooked in the analysis of games, as existing works tend to disregard the possibility that player interactions might be very low which results in a pessimistic rate. We start with the following auxiliary lemma.

**Lemma B.8.** *For any $\mathbf{x}_0^\star \in \mathcal{X}$ that satisfies $F_0(\mathbf{x}_0^\star) = 0$ where $F_0(\mathbf{x}) = F_{\bar{\mathbf{x}} = \mathbf{x}_0}(\mathbf{x})$, after $K$ steps of Decoupled SGDA starting from $\mathbf{x}_0$ with a stepsize of $\gamma \leqslant \frac{\bar{\mu}}{L^2}$, we have:*

$$\mathbb{E}\left[ \|\mathbf{x}_K - \mathbf{x}_0^\star\|^2 \right] \leqslant (1 - \gamma\bar{\mu})^K \mathbb{E}\|\mathbf{x}_0 - \mathbf{x}_0^\star\|^2 + \frac{\gamma\bar{\sigma}^2}{\bar{\mu}} \tag{18}$$

*Proof.* We start by upper bounding the iterates generated by our method from $\mathbf{x}_0^\star$ at a time step $0 < t + 1 \leqslant K$ using the update rule of our method. Recall that $G_0(\mathbf{x}, \xi) \equiv G_{\bar{\mathbf{x}} = \mathbf{x}_0}(\mathbf{x}, \xi)$ where $\mathbf{x}_0 = (\mathbf{u}_0, \mathbf{v}_0)$.

$$
\begin{aligned}
\|\mathbf{x}_{t+1} - \mathbf{x}_0^\star\|^2 &= \|\mathbf{x}_t - \gamma \mathbf{P}^{-1} G_0(\mathbf{x}_t, \xi_t) - \mathbf{x}_0^\star\|^2 \\
&= \|\mathbf{x}_t - \mathbf{x}_0^\star\|^2 + \gamma^2 \|G_0(\mathbf{x}_t, \xi)\|_*^2 - 2\gamma\langle G_0(\mathbf{x}_t, \xi_t), \mathbf{x}_t - \mathbf{x}_0^\star\rangle.
\end{aligned}
$$

By taking the conditional expectation on previous iterates we have:

$$
\begin{aligned}
\mathbb{E}_{\xi_t} &\|\mathbf{x}_{t+1} - \mathbf{x}_0^\star\|^2 \\
&\leqslant \|\mathbf{x}_t - \mathbf{x}_0^\star\|^2 + \gamma^2 \|F_0(\mathbf{x}_t) - F_0(\mathbf{x}_0^\star)\|_*^2 - 2\gamma\langle F_0(\mathbf{x}_t) - F_0(\mathbf{x}_0^\star), \mathbf{x}_t - \mathbf{x}_0^\star\rangle + \gamma^2\bar{\sigma}^2 \\
&\leqslant (1 - 2\gamma\bar{\mu} + \gamma^2\bar{L}^2) \|\mathbf{x}_t - \mathbf{x}_0^\star\|^2 + \gamma^2\bar{\sigma}^2
\end{aligned}
$$

With the choice of $\gamma \leqslant \frac{\bar{\mu}}{L^2}$ and taking the unconditional expectation we have:

$$\mathbb{E}\|\mathbf{x}_{t+1} - \mathbf{x}_0^\star\|^2 \leqslant (1 - \gamma\bar{\mu})\mathbb{E}\|\mathbf{x}_t - \mathbf{x}_0^\star\|^2 + \gamma^2\bar{\sigma}^2$$

After unrolling the recursion for $K$ steps using Lemma B.1 we have:

$$
\begin{aligned}
\mathbb{E}\|\mathbf{x}_K - \mathbf{x}_0^\star\|^2 &\leqslant (1 - \gamma\bar{\mu})^K \mathbb{E}\|\mathbf{x}_0 - \mathbf{x}_0^\star\|^2 + \sum_{i=0}^{K-1}(1 - \gamma\bar{\mu})^i \gamma^2\bar{\sigma}^2 \\
&\leqslant (1 - \gamma\bar{\mu})^K \mathbb{E}\|\mathbf{x}_0 - \mathbf{x}_0^\star\|^2 + \frac{\gamma\bar{\sigma}^2}{\bar{\mu}}
\end{aligned}
$$

$$\square$$

Now we are ready to prove the main theorem.

**Theorem B.9.** *For any* $R, K \geqslant \frac{1}{\gamma\mu} \log\left(\frac{4}{\kappa_c}\right)$, *after running Decoupled SGDA for a total of* $T = KR$ *iterations on a function* $f$, *with the stepsize* $\gamma \leqslant \frac{\bar{\mu}}{L^2}$ *in weakly coupled regime* ($4\kappa_c < 1$), *we get a rate of:*

$$\mathbb{E}\left[\left\|\mathbf{x}_K^R - \mathbf{x}^\star\right\|^2\right] \leqslant D^2 (4\kappa_c)^R + \frac{8\gamma\bar{\sigma}^2}{\bar{\mu}} \cdot \frac{\kappa_c}{1 - 4\kappa_c}$$

*where* $D = \|\mathbf{x}_0 - \mathbf{x}^\star\|$.

*Proof.* We start by upper bounding the following term where $\mathbf{x}_t \equiv (\mathbf{u}_t, \mathbf{v}_t)$ is the parameters of players at some round $r$ after $t$ local steps.

$$\|\mathbf{x}_{t+1} - \mathbf{x}^\star\|^2 \leqslant 2 \|\mathbf{x}_{t+1} - \mathbf{x}_0^\star\|^2 + 2 \|\mathbf{x}_0^\star - \mathbf{x}^\star\|^2 \tag{19}$$

where $\mathbf{x}_0^\star \in \mathcal{X}$ satisfies $F_{\bar{\mathbf{x}}}(\mathbf{x}_0^\star) = 0$. Recall that $F_{\bar{\mathbf{x}}}(\mathbf{x}_0^\star) = (\nabla_u f(\mathbf{u}_0^\star, \bar{\mathbf{v}}), -\nabla_v f(\bar{\mathbf{u}}, \mathbf{v}_0^\star))$. The point $\mathbf{u}_0^\star$ is the minimizer of $f$ given a fixed $\mathbf{v} = \bar{\mathbf{v}}$ meaning that $\mathbf{u}_0^\star = \arg\min_{\mathbf{u}\in\mathcal{X}_u} f(\mathbf{u}, \bar{\mathbf{v}})$ and $\mathbf{v}_0^\star$ is the maximizer of $f$ given a fixed $\mathbf{u} = \bar{\mathbf{u}}$ meaning that $\mathbf{v}_0^\star = \arg\max_{\mathbf{v}\in\mathcal{X}_v} f(\bar{\mathbf{u}}, \mathbf{v})$. Note that we know such minimizer and maximizer exists as the function is strongly convex in $\mathbf{u}$ and strongly concave in $\mathbf{v}$.

For the first term we use Lemma B.8 and we get:

$$\mathbb{E}\left[\|\mathbf{x}_{t+1} - \mathbf{x}_0^\star\|^2\right] \leqslant (1 - \gamma\bar{\mu})^K \mathbb{E} \|\mathbf{x}_0 - \mathbf{x}_0^\star\|^2 + \frac{\gamma\bar{\sigma}^2}{\bar{\mu}}$$

Putting this back in (19) gives us:

$$\mathbb{E} \|\mathbf{x}_{t+1} - \mathbf{x}^\star\|^2$$

$$\leqslant 2(1 - \gamma\bar{\mu})^K \mathbb{E} \|\mathbf{x}_0 - \mathbf{x}_0^\star\|^2 + 2\mathbb{E} \|\mathbf{x}_0^\star - \mathbf{x}^\star\|^2 + \frac{2\gamma\bar{\sigma}^2}{\bar{\mu}}$$

$$\leqslant 2(1 - \gamma\bar{\mu})^K \mathbb{E} \|\mathbf{x}_0 - \mathbf{x}_0^\star\|^2 + 2\kappa_c \mathbb{E} \|\mathbf{x}_0 - \mathbf{x}^\star\|^2 + \frac{2\gamma\bar{\sigma}^2}{\bar{\mu}}$$

$$\leqslant 4(1 - \gamma\bar{\mu})^K \mathbb{E} \|\mathbf{x}_0 - \mathbf{x}^\star\|^2 + 4(1 - \gamma\bar{\mu})^K \mathbb{E} \|\mathbf{x}_0^\star - \mathbf{x}^\star\|^2 + 2\kappa_c \mathbb{E} \|\mathbf{x}_0 - \mathbf{x}^\star\|^2 + \frac{2\gamma\bar{\sigma}^2}{\bar{\mu}}$$

$$\leqslant 4(1 - \gamma\bar{\mu})^K \mathbb{E} \|\mathbf{x}_0 - \mathbf{x}^\star\|^2 + \left(4(1 - \gamma\bar{\mu})^K \kappa_c + 2\kappa_c\right) \mathbb{E} \|\mathbf{x}_0 - \mathbf{x}^\star\|^2 + \frac{2\gamma\bar{\sigma}^2}{\bar{\mu}}$$

$$\leqslant \left(4(1 - \gamma\bar{\mu})^K + 4(1 - \gamma\bar{\mu})^K \kappa_c + 2\kappa_c\right) \mathbb{E} \|\mathbf{x}_0 - \mathbf{x}^\star\|^2 + \frac{2\gamma\bar{\sigma}^2}{\bar{\mu}}$$

$$\leqslant \left(4\exp\left(-\gamma\bar{\mu}K\right) + 4\exp\left(-\gamma\bar{\mu}K\right)\kappa_c + 2\kappa_c\right) \mathbb{E} \|\mathbf{x}_0 - \mathbf{x}^\star\|^2 + \frac{2\gamma\bar{\sigma}^2}{\bar{\mu}}$$

where we used Lemma B.4 in the third line. Now we need to make sure that $4\exp\left(-\gamma\bar{\mu}K\right) \leqslant \kappa_c \leqslant 1$ which is implied by $K \geqslant \frac{1}{\gamma\bar{\mu}} \log\left(\frac{4}{\kappa_c}\right)$. Next we have:

$$\mathbb{E} \|\mathbf{x}_{t+1} - \mathbf{x}^\star\|^2 \leqslant 4\kappa_c \mathbb{E} \|\mathbf{x}_0 - \mathbf{x}^\star\|^2 + \frac{2\gamma\bar{\sigma}^2}{\bar{\mu}}$$

The above recursion can be re-written in terms of two consecutive rounds:

$$\mathbb{E} \|\mathbf{x}^{r+1} - \mathbf{x}^\star\|^2 \leqslant 4\kappa_c \mathbb{E} \|\mathbf{x}^r - \mathbf{x}^\star\|^2 + \frac{2\gamma\bar{\sigma}^2}{\bar{\mu}}$$

After unrolling the recursion for $R$ rounds using Lemma B.1 we have:

$$\mathbb{E}\left\|\mathbf{x}^R - \mathbf{x}^\star\right\|^2 \leqslant (4\kappa_c)^R \, \mathbb{E}\left\|\mathbf{x}_0 - \mathbf{x}^\star\right\|^2 + \frac{2\gamma\bar{\sigma}^2}{\bar{\mu}} \sum_{i=1}^R (4\kappa_c)^i$$

Note that we assumed the game is weakly coupled which implies that $4\kappa_c \leqslant 1$. Finally we have:

$$\mathbb{E}\left\|\mathbf{x}^R - \mathbf{x}^\star\right\|^2 \leqslant (4\kappa_c)^R \, \mathbb{E}\left\|\mathbf{x}_0 - \mathbf{x}^\star\right\|^2 + \frac{2\gamma\bar{\sigma}^2}{\bar{\mu}} \sum_{i=1}^R (4\kappa_c)^i$$

$$\leqslant D^2 \, (4\kappa_c)^R + \frac{8\gamma\bar{\sigma}^2}{\bar{\mu}} \cdot \frac{\kappa_c}{1 - 4\kappa_c}$$

$\square$

## B.2. Proof of Theorem 4.1 in Non Weakly Coupled Regime

We start with some auxiliary lemmas.

**Lemma B.10.** *Let $\{r_t\}_{t\geqslant 0}$ be a non-negative sequence of numbers that satisfy*

$$r_{t+1} \leqslant (1 - a\gamma)r_t + \frac{b}{K}\gamma \sum_{i=\max\{0,t-K+1\}}^t r_i + c\gamma^2 \,,$$

*for constants $a > 0$, $b, c \geqslant 0$ and integer $K \geqslant 1$ and a parameter $\gamma \geqslant 0$, such that $a\gamma \leqslant \frac{1}{K}$. If $b \leqslant \frac{a}{4}$, then it holds*

$$r_t \leqslant \left(1 - \frac{a}{2}\gamma\right)^t r_0 + \frac{2c}{a}\gamma \,. \tag{20}$$

*Proof.* By assumption on $r_t$:

$$r_{t+1} \leqslant \left(1 - \frac{a\gamma}{2}\right) r_t - \frac{a\gamma}{2}r_t + \frac{b}{K}\gamma \sum_{i=\max\{0,t-K+1\}}^t r_i + c\gamma^2 \,,$$

and by unrolling the recursion:

$$r_{t+1} \leqslant \left(1 - a\frac{\gamma}{2}\right)^t r_0 + \sum_{i=0}^t \left(1 - \frac{a\gamma}{2}\right)^{t-i} \left[ -\frac{a\gamma}{2}r_i + \frac{b}{K}\gamma \sum_{j=\max\{0,i-K+1\}}^i r_j \right] + \sum_{i=0}^t \left(1 - \frac{a\gamma}{2}\right)^{t-i} c\gamma^2$$

$$\leqslant \left(1 - \frac{a\gamma}{2}\right)^t r_0 + \sum_{i=0}^t \left(1 - \frac{a\gamma}{2}\right)^{t-i} \left[ -\frac{a\gamma}{2}r_i + \frac{b}{K}\gamma \sum_{j=\max\{0,i-K+1\}}^i r_j \right] + \frac{2c}{a}\gamma$$

$$= \left(1 - a\frac{\gamma}{2}\right)^t r_0 + \sum_{i=0}^t \left(1 - \frac{a\gamma}{2}\right)^{t-i} \left[ -\frac{a\gamma}{2}r_i + \frac{b}{K}\gamma \sum_{j=\max\{0,i-K-1\}}^i \left(1 - \frac{a\gamma}{2}\right)^{i-j} r_i \right] + \frac{2c}{a}\gamma$$

where we used $\sum_{i=0}^t (1 - \frac{a\gamma}{2})^i \leqslant \frac{2}{a\gamma}$ (for $(\frac{a\gamma}{2}) < 1$) for the second inequality.

By estimating

$$-\frac{a\gamma}{2}r_i + \frac{b}{K}\gamma \sum_{j=\max\{0,i-K-1\}}^i (1 - \frac{a\gamma}{2})^{i-j}r_i \leqslant -\frac{a\gamma}{2}r_i + \frac{b}{K}\gamma \sum_{j=\max\{0,i-K-1\}}^i \left(1 - \frac{a\gamma}{2}\right)^{1-K} r_i$$

$$\leqslant -\frac{a\gamma}{2}r_i + b\gamma r_i \left(1 - \frac{a\gamma}{2}\right)^{1-K} r_i$$

$$\leqslant -\frac{a\gamma}{2}r_i + 2b\gamma r_i \leqslant 0 \,,$$

with and $(1 - \frac{a\gamma}{2})^{1-K} \leqslant 2$ for $a\gamma \leqslant \frac{1}{K}$, and the assumption $b \leqslant \frac{a}{4}$ (and $r_i \geqslant 0$).

The validity of the inequality, $(1 - \frac{a\gamma}{2})^{1-K} \leqslant 2$ for $a\gamma \leqslant \frac{1}{K}$ can be shown in the following way:

$$\left(1 - \frac{a\gamma}{2}\right)^{1-K} \leqslant \left(1 - \frac{a\gamma}{2}\right)^{-K} \leqslant e^{\frac{a\gamma K}{2}}$$

For the last inequality above we used the approximation $(1-x)^{-n} \leqslant e^{nx}$ for $x \geqslant 0$ and $n \geqslant 0$:

Given that $a\gamma \leqslant \frac{1}{K}$, we have:

$$e^{\frac{a\gamma K}{2}} \leqslant e^{\frac{1}{2}}.$$

Thus, we have

$$\left(1 - \frac{a\gamma}{2}\right)^{1-K} \leqslant 2$$

Going back to the main proof, we conclude

$$r_{t+1} \leqslant \left(1 - \frac{a\gamma}{2}\right)^t r_0 + \frac{2c}{a}\gamma.$$

as claimed. $\qquad\square$

**Lemma B.11** (**Consensus error**). *After running Decoupled SGDA for $K$ local steps at some round $r$ with a step-size of $\gamma \leqslant \frac{a}{32LL_cK}$ for any constant $0 < a < L$, the consensus error can be upper bounded as follows:*

$$\mathbb{E}\left\|\mathbf{x}_{t+1} - \mathbf{x}_0\right\|^2 \leqslant \sum_{i=t+1-K}^{t} \frac{a^2}{64KL_c^2}\mathbb{E}\left\|\mathbf{x}_i - \mathbf{x}^\star\right\|^2 + 4K\gamma^2\bar{\sigma}^2 \tag{21}$$

*Proof.* Recall that $G_0(\mathbf{x}, \xi) \equiv G_{\bar{\mathbf{x}}=\mathbf{x}_0}(\mathbf{x}, \xi)$ and $\mathbb{E}_\xi[G_0(\mathbf{x}, \xi)] = F_0(\mathbf{x})$ where $\mathbf{x}_0 = (\mathbf{u}_0, \mathbf{v}_0)$ refers to the parameters of each player at the beginning of some round $r$. Using the update rule of our method we have:

$$\mathbb{E}\left\|\mathbf{x}_{t+1} - \mathbf{x}_0\right\|^2$$
$$= \mathbb{E}\left\|\mathbf{x}_t - \gamma\mathbf{P}^{-1}G_0(\mathbf{x}_t, \xi) - \mathbf{x}_0\right\|^2$$
$$\leqslant \mathbb{E}\left\|\mathbf{x}_t - \gamma\mathbf{P}^{-1}F_0(\mathbf{x}_t) - \mathbf{x}_0\right\|^2 + \gamma^2\bar{\sigma}^2$$
$$\leqslant \left(1 + \frac{1}{K}\right)\mathbb{E}\left\|\mathbf{x}_t - \mathbf{x}_0\right\|^2 + 2K\gamma^2\mathbb{E}\left\|F_0(\mathbf{x}_t)\right\|_*^2 + \gamma^2\bar{\sigma}^2$$
$$\leqslant \left(1 + \frac{1}{K}\right)\mathbb{E}\left\|\mathbf{x}_t - \mathbf{x}_0\right\|^2 + 2K\gamma^2\mathbb{E}\left\|F_0(\mathbf{x}_t) - F(\mathbf{x}_t) + F(\mathbf{x}_t)\right\|_*^2 + \gamma^2\bar{\sigma}^2$$
$$\leqslant \left(1 + \frac{1}{K}\right)\mathbb{E}\left\|\mathbf{x}_t - \mathbf{x}_0\right\|^2 + 4K\gamma^2\mathbb{E}\left\|F_0(\mathbf{x}_t) - F(\mathbf{x}_t)\right\| + 4K\gamma^2\mathbb{E}\left\|F(\mathbf{x}_t)\right\|_*^2 + \gamma^2\bar{\sigma}^2$$
$$\leqslant \left(1 + \frac{1}{K}\right)\mathbb{E}\left\|\mathbf{x}_t - \mathbf{x}_0\right\|^2 + 4KL_c^2\gamma^2\mathbb{E}\left\|\mathbf{x}_t - \mathbf{x}_0\right\|^2 + 4KL^2\gamma^2\mathbb{E}\left\|\mathbf{x}_t - \mathbf{x}^\star\right\|^2 + \gamma^2\bar{\sigma}^2$$

With the choice of $\gamma \leqslant \frac{a}{32KLL_c}$, we get:

$$\mathbb{E}\left\|\mathbf{x}_{t+1} - \mathbf{x}_0\right\|^2$$
$$\leqslant \left(1 + \frac{1}{K}\right)\mathbb{E}\left\|\mathbf{x}_t - \mathbf{x}_0\right\|^2 + \frac{a^2}{256KL^2}\mathbb{E}\left\|\mathbf{x}_t - \mathbf{x}_0\right\|^2 + \frac{a^2}{256KL_c^2}\mathbb{E}\left\|\mathbf{x}_t - \mathbf{x}^\star\right\|^2 + \gamma^2\bar{\sigma}^2$$
$$\leqslant \left(1 + \frac{1}{K}\right)\mathbb{E}\left\|\mathbf{x}_t - \mathbf{x}_0\right\|^2 + \frac{1}{256K}\mathbb{E}\left\|\mathbf{x}_t - \mathbf{x}_0\right\|^2 + \frac{a^2}{256KL_c^2}\mathbb{E}\left\|\mathbf{x}_t - \mathbf{x}^\star\right\|^2 + \gamma^2\bar{\sigma}^2$$
$$\leqslant \left(1 + \frac{1}{K} + \frac{1}{256K}\right)\mathbb{E}\left\|\mathbf{x}_t - \mathbf{x}_0\right\|^2 + \frac{a^2}{256KL_c^2}\mathbb{E}\left\|\mathbf{x}_t - \mathbf{x}^\star\right\|^2 + \gamma^2\bar{\sigma}^2$$

where in the third line we used the fact that $\frac{a^2}{L^2} \leqslant 1$ due to the assumption $a < L$. By unrolling the recursion for the last $K$ steps and considering the fact that $\left(1 + \frac{1}{K} + \frac{1}{256K}\right)^K \leqslant 4$ we get:

$$\mathbb{E}\left\|\mathbf{x}_{t+1} - \mathbf{x}_0\right\|^2 \leqslant \sum_{i=t+1-K}^{t} \frac{a^2}{64KL_c^2} \mathbb{E}\left\|\mathbf{x}_i - \mathbf{x}^\star\right\|^2 + 4K\gamma^2\bar{\sigma}^2$$

$\square$

Now we are ready to prove the following theorem.

**Theorem B.12** (Decoupled SGDA for two-player Games). *For any $R, K$, after running Decoupled SGDA for a total of $T = KR$ iterations on a function $f$, with the stepsize $\gamma \leqslant \min\left\{\frac{\mu}{L^2}, \frac{\mu}{KLL_c}, \frac{\mu}{KL_c^2}\right\}$ in the non weakly coupled regime, we get a rate of:*

$$\mathbb{E}\left[\|\mathbf{x}_K^R - \mathbf{x}^\star\|^2\right] \leqslant D^2 \exp\left(-\frac{\gamma\mu}{2}KR\right) + \frac{2\bar{\sigma}^2\gamma}{\mu},$$

*where $D = \|\mathbf{x}_0 - \mathbf{x}^\star\|$.*

*Proof.* We start by upper bounding the iterate $\mathbf{x}$ at time step $t + 1$ from the equilibrium. Recall that $F_0(\mathbf{x}) \equiv F_{\bar{\mathbf{x}}=\mathbf{x}_0}(\mathbf{x})$ where $\mathbf{x}_0$ refers to the parameters of players at the beginning of some round $r$.

$$\mathbb{E}\left\|\mathbf{x}_{t+1} - \mathbf{x}^\star\right\|^2$$
$$\leqslant \mathbb{E}\left\|\mathbf{x}_t - \gamma\mathbf{P}^{-1}G_0(\mathbf{x}_t, \xi) - \mathbf{x}^\star\right\|^2 + \gamma^2\bar{\sigma}^2$$
$$\leqslant \mathbb{E}\left\|\mathbf{x}_t - \gamma\mathbf{P}^{-1}F_0(\mathbf{x}_t) - \mathbf{x}^\star\right\|^2 + \gamma^2\bar{\sigma}^2$$
$$= \mathbb{E}\left\|\mathbf{x}_t - \gamma\mathbf{P}^{-1}F(\mathbf{x}_t) - \mathbf{x}^\star + \gamma\mathbf{P}^{-1}F(\mathbf{x}_t) - \gamma\mathbf{P}^{-1}F_0(\mathbf{x}_t)\right\|^2 + \gamma^2\bar{\sigma}^2$$
$$\leqslant \left(1 + \frac{\gamma\mu}{2}\right)\left[\mathbb{E}\left\|\mathbf{x}_t - \gamma\mathbf{P}^{-1}F(\mathbf{x}_t) - \mathbf{x}^\star\right\|^2\right] + \gamma\left(\gamma + \frac{2}{\mu}\right)\mathbb{E}\left\|F(\mathbf{x}_t) - F_0(\mathbf{x}_t)\right\|_*^2 + \gamma^2\bar{\sigma}^2$$
$$= \left(1 + \frac{\gamma\mu}{2}\right)\left[\mathbb{E}\left\|\mathbf{x}_t - \mathbf{x}^\star\right\|^2 + \gamma^2\mathbb{E}\left\|F(\mathbf{x}_t) - F(\mathbf{x}^\star)\right\|_*^2 - 2\gamma\langle F(\mathbf{x}_t) - F(\mathbf{x}^\star), \mathbf{x}_t - \mathbf{x}^\star\rangle\right] +$$
$$\gamma\left(\gamma + \frac{2}{\mu}\right)\mathbb{E}\left\|F(\mathbf{x}_t) - F_0(\mathbf{x}_t)\right\|_*^2 + \gamma^2\bar{\sigma}^2$$
$$= \left(1 + \frac{\gamma\mu}{2}\right)\left[(1 + \gamma^2L^2 - 2\gamma\mu)\mathbb{E}\left\|\mathbf{x}_t - \mathbf{x}^\star\right\|^2\right] + \gamma\left(\gamma + \frac{2}{\mu}\right)\mathbb{E}\left\|F(\mathbf{x}_t) - F_0(\mathbf{x}_t)\right\|_*^2 + \gamma^2\bar{\sigma}^2$$
$$\leqslant \left(1 + \frac{\gamma\mu}{2}\right)\left[(1 - \gamma\mu)\mathbb{E}\left\|\mathbf{x}_t - \mathbf{x}^\star\right\|^2\right] + \frac{3\gamma}{\mu}\mathbb{E}\left\|F(\mathbf{x}_t) - F_0(\mathbf{x}_t)\right\|_*^2 + \gamma^2\bar{\sigma}^2$$
$$\leqslant \left(1 - \frac{\gamma\mu}{2}\right)\mathbb{E}\left\|\mathbf{x}_t - \mathbf{x}^\star\right\|^2 + \frac{3\gamma L_c^2}{\mu}\mathbb{E}\left\|\mathbf{x}_t - \mathbf{x}_0\right\|^2 + \gamma^2\bar{\sigma}^2$$

Where we assumed that $\gamma \leqslant \frac{\mu}{L^2}$. Now by using the upper bound on consensus error from Lemma B.11 and setting $a = \mu$ we get:

$$\mathbb{E}\left\|\mathbf{x}_{t+1} - \mathbf{x}^\star\right\|^2$$
$$= \left(1 - \frac{\gamma\mu}{2}\right)\mathbb{E}\left\|\mathbf{x}_t - \mathbf{x}^\star\right\|^2 + \frac{\gamma\mu}{16K}\sum_{i=\max\{0,t-K\}}^{t}\mathbb{E}\left\|\mathbf{x}_i - \mathbf{x}^\star\right\|^2 + \left(1 + \frac{12K\gamma L_c^2}{\mu}\right)\gamma^2\bar{\sigma}^2$$

With the choice of $\gamma \leqslant \frac{\mu}{12KL_c^2}$ we have:

$$\mathbb{E}\left\|\mathbf{x}_{t+1} - \mathbf{x}^\star\right\|^2$$

$$\left(1 - \frac{\gamma\mu}{2}\right)\mathbb{E}\left\|\mathbf{x}_t - \mathbf{x}^\star\right\|^2 + \frac{\gamma\mu}{16K}\sum_{i=\max\{0,t-K\}}^{t}\mathbb{E}\left\|\mathbf{x}_i - \mathbf{x}^\star\right\|^2 + 2\frac{\gamma\bar{\sigma}^2}{\mu}$$

By unrolling the recursion using Lemma B.10 we get:

$$\mathbb{E}\left\|\mathbf{x}_K^R - \mathbf{x}^\star\right\|^2 \leqslant D^2\exp\left(-\frac{\mu^2}{L^2}R\right) + 2\frac{\gamma^2}{\mu}\bar{\sigma}^2$$

$\square$

# C. Decoupled SGD for $N$-player games

In this section, we generalize all previous results on two-player games to $N$-player games. We first introduce the notation that is needed to define $N$-player games and will be used to establish our convergence guarantees.

## C.1. Setting and Preliminaries

**Notation.** We consider unconstrained $N$-player games where each player $\mathbf{x}^i$ belongs to the space $\mathcal{X}_i = \mathbb{R}^{d_i}$. The vector $\mathbf{x} = (\mathbf{x}^1, \ldots, \mathbf{x}^N) \in \mathbb{R}^d$ is defined in the space $\mathcal{X} = \mathcal{X}_1 \times \ldots \times \mathcal{X}_N = \mathbb{R}^d$ with $d = \sum_{i=1}^N d_i$. The space $\mathcal{X}_i$ for all $i \in [N]$ is equipped with a certain Euclidean norm, $\|\mathbf{x}^i\|_i := \langle\mathbf{P}_i\mathbf{x}^i, \mathbf{x}^i\rangle^{1/2}$, where $\mathbf{P}_i$ is a positive definite matrix. The norm in the space $\mathcal{X}$ is then defined by $\|\mathbf{x}\| = (\sum_{i=1}^N \alpha_i\|\mathbf{x}^i\|_i^2)^{1/2}$ where $\alpha_i > 0$; thus, $\|\mathbf{x}\| = \langle\mathbf{P}\mathbf{x}, \mathbf{x}\rangle^{1/2}$, where $\mathbf{P}$ is the block-diagonal matrix with blocks $\alpha_i\mathbf{P}_i$ ($\mathbf{P} = \text{diag}(\alpha_1\mathbf{P}_1, \ldots, \alpha_N\mathbf{P}_N)$). The dual norms are defined as: $\|\mathbf{g}_i\|_{i,*} := \max_{\|\mathbf{x}^i\|_i=1}\langle\mathbf{g}_i, \mathbf{x}^i\rangle = \langle\mathbf{g}_i, \mathbf{P}_i^{-1}\mathbf{g}_i\rangle^{1/2}$ ($\mathbf{g}_i \in \mathcal{X}_{d_i}$) and $\|\mathbf{g}\|_* := \max_{\|\mathbf{x}\|=1}\langle\mathbf{g}, \mathbf{x}\rangle = (\sum_{i=1}^N \frac{1}{\alpha_i}\|\mathbf{g}_i\|_{i,*}^2)^{1/2} = \langle\mathbf{g}, \mathbf{P}^{-1}\mathbf{g}\rangle^{1/2}$ ($\mathbf{g} \equiv (\mathbf{g}_1, \ldots, \mathbf{g}_N) \in \mathcal{X}$).

Similar to the work (Nesterov, 2012), we define the following partitioning of the identity matrix:

$$\mathbf{I}_d = (\mathbf{U}_1, \mathbf{U}_2, \ldots, \mathbf{U}_N) \in \mathbb{R}^{d\times d}, \ d = \sum_{i=1}^N d_i, \ \mathbf{U}_i \in \mathbb{R}^{d\times d_i}$$

Now we can represent the vector $\mathbf{x}$ as follows:

$$\mathbf{x} = \sum_{i=1}^N \mathbf{U}_i\mathbf{x}^i \in \mathcal{X}.$$

We can extract the parameters of one player as follows:

$$\mathbf{x}^n = \mathbf{U}_n^\top\mathbf{x} \in \mathcal{X}_n$$

**Problem Formulation.** An $N$-player games is defined as:

$$\left(\min_{\mathbf{x}^1}f_1(\mathbf{x}), \ldots, \min_{\mathbf{x}^N}f_N(\mathbf{x})\right) \tag{$N$-player}$$

Where $f_n\colon \mathcal{X} \to \mathbb{R}$.

The goal is to find the Nash Equilibrium in $\mathbf{x}^\star = (\mathbf{x}^{\star 1}, \ldots, \mathbf{x}^{\star N})$ like in the work (Bravo et al., 2018), which has the property that if one player changes their strategy, their payoff function will increase. In other words, there is no incentive to change one strategy alone: for all $\mathbf{h}_n \in \mathcal{X}_n$, it holds that

$$f_n(\mathbf{x}^\star) \leqslant f_n(\mathbf{x}^\star + \mathbf{U}_n\mathbf{h}_n). \tag{22}$$

Moreover, we define the operator $F_{\bar{\mathbf{x}}}(\mathbf{x}) : \mathcal{X} \to \mathcal{X}$ with respect to the fixed point $\bar{\mathbf{x}} \in \mathcal{X}$ which denotes the stack of gradients with respect to each player's parameters as follows:

$$F_{\bar{\mathbf{x}}}(\mathbf{x}) := (\nabla_1 f_1(\bar{\mathbf{x}} + \mathbf{U}_1 \mathbf{h}), \ldots, \nabla_N f_N(\bar{\mathbf{x}} + \mathbf{U}_N \mathbf{h}))$$

Where $\mathbf{h} = \mathbf{x} - \bar{\mathbf{x}}$. We can recover the commonly used operator as $F(\mathbf{x}) \equiv F_{\bar{\mathbf{x}}=\mathbf{x}}(\mathbf{x}) = (\nabla_1 f_1(\mathbf{x}), \ldots, \nabla_N f_N(\mathbf{x}))$. Note that for the equilibrium, it holds that $F(\mathbf{x}^\star) = 0$ while in general $F_{\bar{\mathbf{x}}}(\mathbf{x}^\star) \neq 0$. We can extract the partial gradient with respect to one player as follows:

$$\nabla_n f_n(\mathbf{x}) = \mathbf{U}_n^\top F_{\bar{\mathbf{x}}}(\mathbf{x}).$$

We now present the assumptions required for the convergence of our method.

**Assumption C.1** (Strong monotonicity). Operators $F_{\bar{\mathbf{x}}}$ and $F$ are strongly monotone with parameters $\bar{\mu}, \mu > 0$, i.e., for all $\mathbf{x}, \bar{\mathbf{x}}, \mathbf{x}' \in \mathcal{X}$, the following inequalities hold:

$$\begin{aligned}
\langle F_{\bar{\mathbf{x}}}(\mathbf{x}) - F_{\bar{\mathbf{x}}}(\mathbf{x}'), \mathbf{x} - \mathbf{x}' \rangle &\geqslant \bar{\mu} \|\mathbf{x} - \mathbf{x}'\|^2. \\
\langle F(\mathbf{x}) - F(\mathbf{x}'), \mathbf{x} - \mathbf{x}' \rangle &\geqslant \mu \|\mathbf{x} - \mathbf{x}'\|^2.
\end{aligned} \tag{23}$$

We can show that $\bar{\mu} = \min_{1 \leqslant i \leqslant N} \{\frac{\mu_i}{\alpha_i}\}$ (Proof in Lemma C.8) where $\langle \nabla_n f_n(\mathbf{x}) - \nabla_n f_n(\mathbf{x} + \mathbf{U}_n \mathbf{d}_n), \mathbf{x}^n - \mathbf{x}'^n \rangle \geqslant \mu_n \|\mathbf{x}'^n - \mathbf{x}^n\|_n^2$.

**Assumption C.2** (Lipschitz gradients). Operators $F_{\bar{\mathbf{x}}}, F : \mathcal{X} \to \mathcal{X}$ are Lipschitz with parameters $\bar{L}$ and $L$ if for all $\mathbf{x}, \bar{\mathbf{x}}, \mathbf{x}' \in \mathcal{X}$, the following inequality holds:

$$\begin{aligned}
\|F_{\bar{\mathbf{x}}}(\mathbf{x}) - F_{\bar{\mathbf{x}}}(\mathbf{x}')\|_* &\leqslant \bar{L} \|\mathbf{x} - \mathbf{x}'\| \\
\|F(\mathbf{x}) - F(\mathbf{x}')\|_* &\leqslant L \|\mathbf{x} - \mathbf{x}'\|
\end{aligned} \tag{24}$$

**Assumption C.3.** The norm of the difference between operators $F_{\bar{\mathbf{x}}}(\mathbf{x})$ and $F(\mathbf{x})$ is upper bounded with parameter $L_c$ for all $\mathbf{x}, \bar{\mathbf{x}} \in \mathcal{X}$ as follows:

$$\|F_{\bar{\mathbf{x}}}(\mathbf{x}) - F(\mathbf{x})\|_* \leqslant L_c \|\mathbf{x} - \bar{\mathbf{x}}\| \tag{25}$$

It's possible to show $L_c = (\max_{1 \leqslant i \leqslant N} \sum_{j \neq i} \frac{\bar{L}_j^2}{\alpha_j})^{1/2}$ where $\bar{L}_n$ is defined as $\|\nabla_n f_n(\mathbf{x}) - \nabla_n f_n(\mathbf{x} + \sum_{i \neq n} \mathbf{U}_i \mathbf{h}_i)\|_{n,*} \leqslant \bar{L}_n \|\sum_{i \neq n} \mathbf{U}_i \mathbf{h}_i\|$ for any $\mathbf{x} \in \mathcal{X}$ and any $\mathbf{h}_n \in \mathcal{X}_n$ (Proof in Lemma C.9). The parameter $\bar{L}_n$ corresponds to smoothness parameter of $f_n$ when we take gradient with respect to player $n$ while varying all other parameters (and fixing the parameters of player $n$). If $\bar{L}_n = 0$ for all $n$, it means for any two players $i, j \in [N]$, $i \neq j$, they have no interaction.

**Assumption C.4.** There exists finite constants $\bar{\sigma}^2$ such that for all $\mathbf{x}, \bar{\mathbf{x}} \in \mathcal{X}$:

$$\mathbb{E}_\xi \left[ \|G_{\bar{\mathbf{x}}}(\mathbf{x}, \xi) - F_{\bar{\mathbf{x}}}(\mathbf{x})\|_*^2 \right] \leqslant \bar{\sigma}^2. \tag{26}$$

Where $\mathbb{E}[G_{\bar{\mathbf{x}}}(\mathbf{x}, \xi)] = F_{\bar{\mathbf{x}}}(\mathbf{x})$.

As we assumed that the above inequality holds for all $\bar{\mathbf{x}} \in \mathcal{X}$, we also cover the common operator $F$ and we denote $G(\mathbf{x}, \xi) \equiv G_{\bar{\mathbf{x}}=\mathbf{x}}(\mathbf{x}, \xi)$.

### C.2. Method

**Local update methods.** As discussed in the two-player section, Local update methods proposed to reduce the communication overhead in distributed optimization. In this context, it's reasonable to assume each player has access to their own stochastic oracle $G_i(\mathbf{x}, \xi)$ for all $i \in [N]$ with the property $\mathbb{E}[G_i(\mathbf{x}, \xi)] = F(\mathbf{x})$ and following bound on the variance of the noise:

$$\begin{aligned}
\mathbb{E}_\xi[\|[G_i(\mathbf{x}, \xi)]_i - [F(\mathbf{x})]_i\|_{i,*}^2] &\leqslant \sigma_{ii}^2, \\
\mathbb{E}_\xi[\|[G_i(\mathbf{x}, \xi)]_j - [F(\mathbf{x})]_j\|_{j,*}^2] &\leqslant \sigma_{ij}^2 \quad \text{for } i \neq j
\end{aligned}$$

However, as we discussed we are considering a setting where the players may not have access to other players' strategies or gradients, and only assume that the private components of the gradients have bounded variance ($\bar{\sigma}^2 \leqslant \sum_{i=1}^N \sigma_{ii}^2$). On

the other hand, local update methods require the variance $\sigma_{ij}^2$ to be bounded as well while we allow that to be arbitrarily large. We introduce the oracle $G_0(\mathbf{x}, \xi) \equiv G_{\bar{\mathbf{x}} = \mathbf{x}_0^r}(\mathbf{x}, \xi)$ for $i = \{u, v\}$ where $\mathbf{x}_0 = (\mathbf{u}_0, \mathbf{v}_0)$ refers to the parameters of each player at the beginning of the round. Our operator only uses the reliable information which is $[G_i(\mathbf{x}, \xi)]_i$ for player $i$. Now we can write the update rule of our method as:

$$\mathbf{x}_{t+1}^r = \mathbf{x}_t^r - \gamma \mathbf{P}^{-1} G_0(\mathbf{x}_t^r, \xi_t), \tag{27}$$

where

$$G_0(\mathbf{x}, \xi) \equiv \left( \nabla_i f_i(\mathbf{x}_0 + \mathbf{U}_i \mathbf{U}_i^\top (\mathbf{x} - \mathbf{x}_0)); \xi) \right)_{1 \leqslant i \leqslant N}.$$

Here, the index $t$ denotes the local update step in the current local update phase on player $i$, and the superscript $r$ indexes the local phases. One communication round is needed for exchanging the updated parameters $\mathbf{x}_K^r$ when passing to the next round. Note that $\mathbf{x}_t^{r,i} \in \mathcal{X}_i$ and $\mathbf{x}_t^r \in \mathcal{X}$.

---

**Algorithm 2** Decoupled SGD for $N$-player games

---

1: **Input:** step size $\gamma$, initialization $\mathbf{x}_0 = (\mathbf{x}_0^1, \ldots, \mathbf{x}_0^N), R, K$
2: **for** $r \in \{1, \ldots, R\}$ **do**
3:     **for** $t \in \{1, \ldots, K\}$ **do**
4:         **for** $n \in \{1, \ldots, N\}$ **in parallel do**
5:             Update local model $\mathbf{x}_{t+1}^{n,r} \leftarrow \mathbf{x}_t^{n,r} - \gamma \alpha_n^{-1} \nabla_n f_n(\mathbf{x}_0 + \mathbf{U}_n(\mathbf{x}_t^r - \mathbf{x}_0); \xi_t)$
6:         **end for**
7:     **end for**
8:     **Communicate** $\left[ \mathbf{x}_K^{1,r}, \ldots, \mathbf{x}_K^{N,r} \right]^\top$ to all players
9: **end for**
10: **Output:** $\mathbf{x}_K^R = (\mathbf{x}_K^{1,R}, \ldots, \mathbf{x}_K^{N,R})$

---

### C.3. Convergence Guarantee

Now we out to a change in the definition of weakly coupled games in $N$-player setting.

**Definition C.5** (Weakly Coupled and Fully Decoupled Games). Given an $N$-player game in the form of $N$-player. We define the **coupling degree** parameter $\kappa_c$ for this game as follows:

$$\boxed{\kappa_c := \frac{L_c}{\bar{\mu}}} \tag{28}$$

This variable measures the level of interaction in the game. A smaller value of $\kappa_c$ indicates less interaction. We say the game is **Weakly Coupled** if the following inequality holds:

$$\kappa_c \leqslant \frac{1}{4} \tag{29}$$

We say the game is **Fully Decoupled** if we have $\kappa_c = 0$ which implies each player is minimizing their own pay-off function independently.

**Theorem C.6.** *For any $R \geqslant 1$ and any $K \geqslant \frac{1}{\gamma \mu} \log \left( \frac{4}{\kappa_c} \right)$, after running Decoupled SGDA for a total of $T = KR$ iterations on a function $f$, with the stepsize $\gamma \leqslant \frac{\bar{\mu}}{L^2}$ in the weakly coupled regime ($4\kappa_c \leqslant 1$), we get a rate of:*

$$\mathbb{E}\left[ \|\mathbf{x}_K^R - \mathbf{x}^\star\|^2 \right] \leqslant D^2 \exp\left( -(1 - 4\kappa_c)R \right) + \frac{\bar{\sigma}^2 \gamma}{\mu} \cdot \frac{8\kappa_c}{1 - 4\kappa_c}.$$

*Moreover, For any $R, K \geqslant 1$, after running Decoupled SGDA for a total of $T = KR$ iterations on a function $f$, with the stepsize $\gamma \leqslant \min \left\{ \frac{\mu}{L^2}, \frac{\mu}{KLL_c} \right\}$ in the non-weakly coupled regime, we get a rate of:*

$$\mathbb{E}\left[ \|\mathbf{x}_K^R - \mathbf{x}^\star\|^2 \right] \leqslant D^2 \exp\left( -\frac{\gamma \mu}{2} KR \right) + \frac{2\bar{\sigma}^2 \gamma}{\mu}.$$

where $D = \|\mathbf{x}_0 - \mathbf{x}^\star\|$.

**Corollary C.7.** *With the choice of $\gamma = \frac{\bar{\mu}}{RL^2}$ if the game is weakly coupled we get:*

$$\mathbb{E}\big[\|\mathbf{x}_K^R - \mathbf{x}^\star\|^2\big] \leqslant D^2 \exp\Big(-(1 - 4\kappa_c)R\Big) + \bar{\sigma}^2 \frac{8\bar{\mu}\kappa_c}{R\mu L^2(1 - 4\kappa_c)}.$$

*Consequently, to reach $\mathbb{E}[\|\mathbf{x}_K^R - \mathbf{x}^\star\|^2] \leqslant \epsilon$, it suffices to perform $R = \max\{\frac{1}{4-\kappa_c} \log(\frac{2D^2}{\epsilon}), \frac{16\bar{\mu}\kappa_c\bar{\sigma}^2}{\mu L^2(1-4\kappa_c)\epsilon}\}$ rounds with $K = \frac{L^2}{\mu\bar{\mu}} \log(\frac{4}{\kappa_c})$. Moreover, with the choice of $\gamma = \min\{\frac{\mu}{32KL^2}, \frac{1}{\mu KR} \log(\max\{2, \frac{\mu^2 D^2}{\bar{\sigma}^2} KR\})\}$ if the game is not weakly coupled we get:*

$$\mathbb{E}\big[\|\mathbf{x}_K^R - \mathbf{x}^\star\|^2\big] \leqslant D^2 \exp\Big(-\frac{\mu^2}{2L^2}R\Big) + \frac{\bar{\sigma}^2}{\mu^2 KR}.$$

*Consequently, to reach $\mathbb{E}[\|\mathbf{x}_K^R - \mathbf{x}^\star\|^2] \leqslant \epsilon$, it suffices to perform $R = \frac{2L^2}{\mu^2} \log(\frac{D^2}{\epsilon})$ with $K = \frac{2\bar{\sigma}^2}{\mu^2\epsilon}$.*

## C.4. Missing Proofs for Section C.3

Before establishing the convergence results, we first need a couple of auxiliary lemmas for $N$-player games.

**Lemma C.8.** *for all $\mathbf{x}, \bar{\mathbf{x}}, \mathbf{x}' \in \mathcal{X}$, the operator $F_{\bar{\mathbf{x}}}$ is $\bar{\mu}$-strongly monotone where $\bar{\mu}$ can be expressed as: , we have*

$$\bar{\mu} = \min_{1 \leqslant i \leqslant N} \Big\{\frac{\mu_i}{\alpha_i}\Big\}. \tag{30}$$

*Proof.* First recall that that each $f_n$ is $\mu_n$-strongly convex in $\mathbf{x}^n \in \mathcal{X}_n$ meaning that:

$$\langle \nabla_n f_n(\mathbf{x}) - \nabla_n f_n(\mathbf{x} + \mathbf{U}_n\mathbf{d}_n), \mathbf{x}^n - \mathbf{x}'^n \rangle \geqslant \mu_n \|\mathbf{x}'^n - \mathbf{x}^n\|_n^2,$$

where $\mathbf{d}_n := \mathbf{x}'^n - \mathbf{x}^n$. Next we have:

$$
\begin{aligned}
\langle F_{\bar{\mathbf{x}}}(\mathbf{x}) &- F_{\bar{\mathbf{x}}}(\mathbf{x}'), \mathbf{x} - \mathbf{x}' \rangle \\
&= \sum_{i=1}^N \langle \nabla_i f(\mathbf{x}) - \nabla_i f(\mathbf{x} + \mathbf{U}_i^\top \mathbf{d}_i), \mathbf{x}^i - \mathbf{x}'^i \rangle \\
&\geqslant \sum_{i=1}^N \mu_i \|\mathbf{x} - \mathbf{x}'\|_i^2 = \sum_{i=1}^N \frac{\mu_i}{\alpha_i}\alpha_i \|\mathbf{x} - \mathbf{x}'^i\|_i^2 \\
&\geqslant \min_{1 \leqslant i \leqslant N} \Big\{\frac{\mu_i}{\alpha_i}\Big\} \|\mathbf{x} - \mathbf{x}'\|^2. \qquad \square
\end{aligned}
$$

where $\mathbf{d}_i := \mathbf{x}'^i - \mathbf{x}^i$.

**Lemma C.9** ($N$-player). *For all $\mathbf{x}, \bar{\mathbf{x}} \in \mathcal{X}$, parameter $L_c$ can be expressed as:*

$$\bar{L} = \big(\max_{1 \leqslant i \leqslant N} \sum_{j \neq i} \frac{\bar{L}_j^2}{\alpha_j}\big)^{1/2}. \tag{31}$$

*Proof.* Recall that for each $n \in [N]$, there exist constants $\hat{L}_n, \bar{L}_n \geqslant 0$ such that, for any $\mathbf{x} \in \mathcal{X}$, any $\mathbf{h}_1 \in \mathcal{X}_1, \ldots, \mathbf{h}_N \in \mathcal{X}_N$ and any $n \in [N]$, it holds that:

$$
\begin{aligned}
\|\nabla_n f_n(\mathbf{x}) - \nabla_n f_n(\mathbf{x} + \mathbf{U}_n\mathbf{h}_n)\|_{n,*} &\leqslant \hat{L}_n \|\mathbf{h}_n\|_n, \\
\|\nabla_n f_n(\mathbf{x}) - \nabla_n f_n(\mathbf{x} + \textstyle\sum_{i \neq n}\mathbf{U}_i\mathbf{h}_i)\|_{n,*} &\leqslant \bar{L}_n \|\textstyle\sum_{i \neq n}\mathbf{U}_i\mathbf{h}_i\|.
\end{aligned}
$$

Next we define $\mathbf{h}_i := \bar{\mathbf{x}}^i - \mathbf{x}^i$ and $\mathbf{v} := \sum_{j \neq i} \mathbf{U}_j \mathbf{h}_j$ and we have:

$$\|F_{\bar{\mathbf{x}}}(\mathbf{x}) - F(\mathbf{x})\|_*^2 = \sum_{i=1}^N \frac{1}{\alpha_i} \|\nabla_i f_i(\mathbf{x}) - \nabla_i f_i(\mathbf{x} + \mathbf{v})\|_{i,*}^2 \leqslant \sum_{i=1}^N \frac{\bar{L}_i^2}{\alpha_i} \|\mathbf{v}\|^2$$

$$= \sum_{i=1}^N \frac{\bar{L}_i^2}{\alpha_i} \sum_{j \neq i} \alpha_j \|\mathbf{h}_j\|_j^2 = \sum_{i=1}^N \beta_i \alpha_i \|\mathbf{h}_i\|_i^2,$$

where $\beta_i = \sum_{j \neq i} \frac{\bar{L}_j^2}{\alpha_j}$. Defining now $L_c^2 = \max_{1 \leqslant i \leqslant N} \beta_i$, we get $\sum_{i=1}^N \beta_i \alpha_i \|\mathbf{h}_i\|_i^2 \leqslant L_c^2 \|\mathbf{h}\|^2$. $\qquad \square$

**Lemma C.10** ($N$-player)**.** *Let $\bar{\mathbf{x}}, \mathbf{x}', \mathbf{x}^\star \in \mathcal{X}$ be such that $F_{\bar{\mathbf{x}}}(\mathbf{x}') = 0$ and $F(\mathbf{x}^\star) = 0$. Then $\kappa_c$ can be expressed as:*

$$\kappa_c = \max_{1 \leqslant i \leqslant N} \left( \sum_{j \neq i} \frac{\alpha_i \bar{L}_i^2}{\mu_i^2} \right)^{1/2}. \tag{32}$$

*Proof.* Let's define $\mathbf{h}_i := \mathbf{x}_0^i - \mathbf{x}^i$, $\mathbf{d}_i := \mathbf{x}'^i - \mathbf{x}^i$, $\mathbf{r}_i := \mathbf{x}^{\star i} - \mathbf{x}^i$, $\mathbf{s}_i := \mathbf{x}_0^i - \mathbf{x}^{\star i}$. We first introduce the point $\mathbf{x}' \in \mathbb{R}^d$ as follows:

$$\mathbf{x}' = (\mathbf{x}'^1, \ldots, \mathbf{x}'^N), \quad \mathbf{x}'^i = \arg\min_{\mathbf{x}^i \in \mathbb{R}^{d_i}} f_i \left( \mathbf{x} + \sum_{j \neq i} \mathbf{U}_j \mathbf{h}_j \right)$$

$$\left\| \mathbf{x}' - \mathbf{x}^\star \right\|^2$$

$$= \sum_{i=1}^N \alpha_i \left\| \mathbf{x}'^i - \mathbf{x}^\star \right\|_i^2$$

$$\leqslant \sum_{i=1}^N \frac{\alpha_i}{\mu_i^2} \left\| \nabla_i f_i \left( \mathbf{x} + \mathbf{U}_i \mathbf{d}_i + \sum_{j \neq i} \mathbf{U}_j \mathbf{h}_j \right) - \nabla_i f_i \left( \mathbf{x} + \mathbf{U}_i \mathbf{r}_i + \sum_{j \neq i} \mathbf{U}_j \mathbf{h}_j \right) \right\|_{i,*}^2$$

$$= \sum_{i=1}^N \frac{\alpha_i}{\mu_i^2} \left\| \nabla_i f_i(\mathbf{x}^\star) - \nabla_i f_i \left( \mathbf{x} + \mathbf{U}_i \mathbf{r}_i + \sum_{j \neq i} \mathbf{U}_j \mathbf{h}_j \right) \right\|_{i,*}^2 \leqslant \sum_{i=1}^N \frac{\alpha_i \bar{L}_i^2}{\mu_i^2} \left\| \sum_{j \neq i} \mathbf{U}_j \mathbf{s}_j \right\|^2$$

$$\leqslant \sum_{i=1}^N \frac{\alpha_i \bar{L}_i^2}{\mu_i^2} \sum_{j \neq i} \alpha_j \|\mathbf{s}_j\|_j^2 \leqslant \sum_{i=1}^N \beta_i \alpha_i \|\mathbf{s}_i\|_i^2 \leqslant \kappa_c^2 \|\mathbf{s}\|^2$$

where $\beta_i = \sum_{j \neq i} \frac{\alpha_i \bar{L}_i^2}{\mu_i^2}$ and $\kappa_c := \max_{1 \leqslant i \leqslant N} \sqrt{\beta_i}$. $\qquad \square$

**More details on smoothness parameters for $N$-player games.** For $N$-player games, we can define the following matrix for the better understanding of the smoothness parameters:

$$\mathbf{L} = \begin{pmatrix} \hat{L}_1 & & & & \\ & \hat{L}_2 & & \boxed{\bar{L}_n} & \\ & & \ddots & & \\ & \boxed{\bar{L}_n} & & \hat{L}_{N-1} & \\ & & & & \hat{L}_N \end{pmatrix} \tag{33}$$

In the above matrix, the row number corresponds to the player with respect to whom we are taking the derivative, while the column number corresponds to the player that is fixed, with all other parameters changing. All the elements $L_{ii}$ on the main diagonal of the matrix measure the strength of each individual player, while the off-diagonal elements $L_{ij}$ for $i \neq j$ measure the interaction between players $i$ and $j$. We assume that all the diagonal elements are upper bounded by $\hat{L}_n$ and all off-diagonal elements are upper bounded by $\bar{L}_n$. Here $n$ is the player which is being fixed. The parameter $\bar{L}_n$ measures the interaction of the $n$th player with all other players.

With the use of these Lemmas, one can easily extend the proof of two player game to the general $N$-player games.

# D. Decoupled GDA for Quadratic Games

To provide an extra insight for the results we showed so far in Section 4 and support them with a separate analysis, we additionally consider analysing our method for quadratic games with bi-linear coupling between the players which are a sub-class of SCSC functions.

**Problem formulation and Notation.** Recall that we consider unconstrained two-player games denoted by $\mathbf{u}$ and $\mathbf{v}$ in the spaces $\mathcal{X}_u = \mathbb{R}^{d_u}$ and $\mathcal{X}_v = \mathbb{R}^{d_v}$, respectively. The corresponding product space $\mathcal{X} = \mathcal{X}_u \times \mathcal{X}_v = \mathbb{R}^d$ (with $d = d_u + d_v$) consists of vectors $\mathbf{x} = (\mathbf{u}, \mathbf{v}) \in \mathbb{R}^d$, where $\mathbf{u} \in \mathcal{X}_u$ and $\mathbf{v} \in \mathcal{X}_v$. We assume that the spaces $\mathcal{X}_u$ and $\mathcal{X}_v$ are equipped with Euclidean norms, denoted by $\|\mathbf{u}\|_u$, $\|\mathbf{v}\|_v$. The norm in the space $\mathcal{X}$ is then defined by $\|\mathbf{x}\| = (\alpha\|\mathbf{u}\|_u^2 + \beta\|\mathbf{v}\|_v^2)^{1/2}$ where $\alpha, \beta > 0$.

We aim to find the saddle point of the following function:

$$\min_{\mathbf{u} \in \mathcal{X}_u} \max_{\mathbf{v} \in \mathcal{X}_v} \left[ f(\mathbf{u}, \mathbf{v}) = \frac{1}{2}\langle \mathbf{u}, \mathbf{A}\mathbf{u} \rangle - \frac{1}{2}\langle \mathbf{v}, \mathbf{B}\mathbf{v} \rangle + \langle \mathbf{u}, \mathbf{C}\mathbf{v} \rangle \right], \tag{QG}$$

where $\mathbf{A} \in \mathbb{S}_{++}^{d_u}$ and $\mathbf{B} \in \mathbb{S}_{++}^{d_v}$ and $\mathbf{C} \in \mathbb{R}^{d_u \times d_v}$. Recall that we defined a general two player game as $f(\mathbf{u}, \mathbf{v}) = g(\mathbf{u}) - h(\mathbf{v}) + r(\mathbf{u}, \mathbf{v})$. For the class of quadratic games, we can be more specific as functions $g(\cdot)$ and $h(\cdot)$ are quadratic functions and $r(\cdot)$ is just a linear term. Moreover, we can be more precise about the smoothness and strong convexity parameters as they are correspond to the maximum and minimum singular values of the matrices $\mathbf{A}, \mathbf{B}$ and $\mathbf{C}$. The matrix $\mathbf{C}$ can be seen as the interaction between two players as it's the only term which involves both $\mathbf{u}$ and $\mathbf{v}$. It's also easy to verify that for quadratic games in the form of (QG), the saddle point is at $\mathbf{x}^\star = (0, 0)$. For clarity and ease of comparison other related works, in this section we assume $\alpha = \beta = 1$.

**Definition D.1.** Consider a function $f\colon \mathcal{X} \to \mathbb{R}$ in the form of (QG) for some $\mathbf{u} \in \mathcal{X}_u, \mathbf{v} \in \mathcal{X}_v$. The Lipschitzness and strong convexity / concavity parameters can be defined as:

$$\mu_u \preceq \mathbf{A} \preceq L_u \quad \text{and} \quad \mu_v \preceq \mathbf{B} \preceq L_v$$

Moreover, we use the norm of matrix $\mathbf{C}$ to measure the strength of the interactive part of the game.

$$L_{uv} = L_{vu} := \|\mathbf{C}\|$$

Note that we assume $L_{uv} = L_{vu}$ which always holds for twice differentiable functions. Now we give an explicit formula for the iterates generated by our method on quadratic games.

**Lemma D.2.** *Given a two-player quadratic game in the form of* (QG). *At some round $r$ after $K$ local steps with a stepsize of $\gamma \leqslant \max\{\frac{1}{L_u}, \frac{1}{L_v}\}$, the exact iterate generated by Decoupled GDA is given as follows:*

$$\mathbf{x}_K^r = \left[ \mathbf{Q}^K + \mathbf{E} \right] \mathbf{x}_0^r$$

$$\mathbf{Q} := \begin{pmatrix} (\mathbf{I} - \gamma\mathbf{A}) & \mathbf{0} \\ \mathbf{0} & (\mathbf{I} - \gamma\mathbf{B}) \end{pmatrix}, \quad \mathbf{E} := \begin{pmatrix} \mathbf{0} & -\mathbf{E}_u \\ \mathbf{E}_v & \mathbf{0} \end{pmatrix} \tag{34}$$

$$\mathbf{E}_u := \left[ \mathbf{I} - (\mathbf{I} - \gamma\mathbf{A})^K \right] \mathbf{A}^{-1}\mathbf{C}, \quad \mathbf{E}_v := \left[ \mathbf{I} - (\mathbf{I} - \gamma\mathbf{B})^K \right] \mathbf{B}^{-1}\mathbf{C}^\top$$

*After taking the norm of both sides we have:*

$$\|\mathbf{x}_K^r\| \leqslant \max\left\{ (1 - \gamma\mu_u)^K, (1 - \gamma\mu_v)^K \right\} + \|\mathbf{C}\| \cdot \max\left\{ \delta(\mathbf{A}), \delta(\mathbf{B}) \right\}^{1/2}$$

$$\delta(\mathbf{A}) := \frac{(1 - (1 - \gamma L_u)^K)^2}{\mu_u^2}, \quad \delta(\mathbf{B}) := \frac{(1 - (1 - \gamma L_v)^K)^2}{\mu_v^2} \tag{35}$$

*Remark* D.3. For a quadratic game in the form of (QG), the saddle point is $\mathbf{x}^\star = (0, 0)$. We expect our method to shrink the norm of $\mathbf{x}_t^r$ in each round by a factor less than 1 so that we converge to the saddle point.

Lemma D.2 shows the dynamics of Decoupled GDA for quadratic functions. We can decompose the exact iterates and write it as the sum of two matrices $\mathbf{Q}$ and $\mathbf{E}$. As $\mathbf{Q}$ is a diagonal matrix to the power of $K$ and we have that $\gamma \leqslant \max\{\frac{1}{L_u}, \frac{1}{L_v}\}$, we know that when $K \to \infty$ then $\mathbf{Q} \to \mathbf{0}$. The second matrix $\mathbf{E}$ can be seen as an error matrix which is caused by the interactive part of the game. It is clear that if the game is fully decoupled which implies $\mathbf{C} = \mathbf{0}$, we get the trivial result that we converge only with local steps **without** the need for communicating. However, for the case that we have this interactive term and the game is weakly coupled, we have to upper bound the norm of this error matrix to derive the convergence rate. We first re-state the notion of weakly coupled games for quadratic games and then provide the convergence rate of Decoupled GDA for quadratic games.

**Definition D.4** (Weakly Coupled and Fully Decoupled Games). Given a quadratic game in the form of (QG). We define the **coupling degree** parameter $\kappa_c$ for this game as follows:

$$\kappa_c := \|\mathbf{C}\| \cdot \max\left\{\frac{1}{\mu_u}, \frac{1}{\mu_v}\right\}. \tag{36}$$

This variable measures the level of interaction in the game. A smaller value of $\kappa_c$ indicates less interaction. For any quadratic game, we say the game is **Weakly Coupled** if the following inequality holds:

$$\kappa_c \leqslant \frac{1}{2} \tag{37}$$

We say the game is **Fully Decoupled** if we have $\kappa_c = 0$ which implies $r(\mathbf{u}, \mathbf{v}) = 0$.

**Theorem D.5.** *For any $R$ and $K \geqslant 1$ with a stepsize of $\gamma \leqslant \max\{\frac{1}{L_u}, \frac{1}{L_v}\}$ which ensures $\delta(\mathbf{A}) \leqslant 1$ and $\delta(\mathbf{B}) \leqslant 1$, after running Decoupled GDA for a total of $T = KR$ iterations on a quadratic game in the form of (QG) assuming the game is weakly coupled, we get a rate of:*

$$\|\mathbf{x}^R - \mathbf{x}^\star\| \leqslant D\left(\exp\left(-\frac{\mu_{\min}}{L_{\max}}K\right) + \frac{L_{uv}}{\mu_{\min}}\right)^R \tag{38}$$

*Where $L_{\max} := \max\{L_u, L_v\}$ and $\mu_{\min} := \min\{\mu_u, \mu_v\}$.*

Theorem D.5 clearly shows the effect of local steps and communication rounds which gives more insights about our method compared to the SCSC case. We can see that the first term in the rate goes to zero with taking more local steps while there is another term that is not affected by local steps. It's indeed intuitive as we don't expect our method to converge with only local steps in general. The remaining error is do to the interactive part. All the previous results discussed for SCSC case can be applied to the quadratic setting as well.

## D.1. Missing Proofs from Section D

We first introduce some auxiliary lemmas that are needed for proofs.

**Lemma D.6.** *Let $\mathbf{A}$ be a positive definite matrix and $\gamma \geqslant 0$. Then matrices $\mathbf{A}^{-1}$ and $(\mathbf{I} - \gamma\mathbf{A})$ are commutative meaning that:*

$$\mathbf{A}^{-1}(\mathbf{I} - \gamma\mathbf{A}) = (\mathbf{I} - \gamma\mathbf{A})\mathbf{A}^{-1} \tag{39}$$

*Proof.*

$$\mathbf{A}^{-1}(\mathbf{I} - \gamma\mathbf{A})$$
$$= \mathbf{A}^{-1} - \gamma\mathbf{I}$$
$$= (\mathbf{I} - \gamma\mathbf{A})\mathbf{A}^{-1}$$

$\square$

**Lemma D.7.** *Let $\mathbf{A}$ be a positive definite matrix and $\gamma \geqslant 0$. Then matrices $\mathbf{A}^{-1}$ and $(\mathbf{I} - \gamma\mathbf{A})^K$ are commutative meaning that:*

$$\mathbf{A}^{-1}(\mathbf{I} - \gamma\mathbf{A})^K = (\mathbf{I} - \gamma\mathbf{A})^K\mathbf{A}^{-1} \tag{40}$$

*Proof.* By induction we assume that this statement holds for $K$ which means $\mathbf{A}^{-1}(\mathbf{I} - \gamma\mathbf{A})^K = (\mathbf{I} - \gamma\mathbf{A})^K\mathbf{A}^{-1}$. Now we show that this statement holds for $K + 1$.

$$\begin{aligned}
&\mathbf{A}^{-1}(\mathbf{I} - \gamma\mathbf{A})^{K+1} \\
&= \mathbf{A}^{-1}(\mathbf{I} - \gamma\mathbf{A})(\mathbf{I} - \gamma\mathbf{A})^K \\
&= (\mathbf{I} - \gamma\mathbf{A})\mathbf{A}^{-1}(\mathbf{I} - \gamma\mathbf{A})^K \\
&= (\mathbf{I} - \gamma\mathbf{A})(\mathbf{I} - \gamma\mathbf{A})^K\mathbf{A}^{-1} \\
&= (\mathbf{I} - \gamma\mathbf{A})^{K+1}\mathbf{A}^{-1}
\end{aligned}$$

For the case of $K = 1$ we use the previous Lemma. □

**Lemma D.8.** *Let $\mathbf{A}$ be a positive definite matrix and $\gamma \geqslant 0$. Then we have that:*

$$\mathbf{A}^{-1}\left((\mathbf{I} - \gamma\mathbf{A})^K - \mathbf{I}\right) = \left((\mathbf{I} - \gamma\mathbf{A})^K - \mathbf{I}\right)\mathbf{A}^{-1} \tag{41}$$

*Proof.*

$$\begin{aligned}
&\mathbf{A}^{-1}\left((\mathbf{I} - \gamma\mathbf{A})^K - \mathbf{I}\right) \\
&= \mathbf{A}^{-1}(\mathbf{I} - \gamma\mathbf{A})^K - \mathbf{A}^{-1} \\
&= (\mathbf{I} - \gamma\mathbf{A})^K\mathbf{A}^{-1} - \mathbf{A}^{-1} \\
&= \left((\mathbf{I} - \gamma\mathbf{A})^K - \mathbf{I}\right)\mathbf{A}^{-1}
\end{aligned}$$

□

### D.2. Explicit Iterates Generated by Decoupled GDA

**Lemma D.9.** *Given a general quadratic game in the form of* (QG). *After $k$ steps of Decoupled GDA at some round $r$ we can compute the explicit form of iterates as follows:*

$$\begin{aligned}
\mathbf{u}_k^r &= -\mathbf{A}^{-1}\mathbf{C}\mathbf{v}_0^r + \mathbf{A}^{-1}\left(\mathbf{I} - \gamma\alpha^{-1}\mathbf{A}\right)^k\left(\mathbf{A}\mathbf{u}_0^r + \mathbf{C}\mathbf{v}_0^r\right) \\
\mathbf{v}_k^r &= \mathbf{B}^{-1}\mathbf{C}^\top\mathbf{u}_0^r + \mathbf{B}^{-1}\left(\mathbf{I} - \gamma\beta^{-1}\mathbf{B}\right)^k\left(\mathbf{B}\mathbf{v}_0^r - \mathbf{C}^\top\mathbf{u}_0^r\right)
\end{aligned}$$

*Proof.* We use induction for the proof of this lemma. By using the update rule of Decoupled GDA we have:

$$
\begin{aligned}
\mathbf{u}_{k+1}^r &= \mathbf{u}_k - \gamma \nabla_u f(\mathbf{u}_k^r, \mathbf{v}_0^r) \\
&= \mathbf{u}_k - \gamma \left( \mathbf{A} \mathbf{u}_k^r + \mathbf{C} \mathbf{v}_0^r \right) \\
&= -\mathbf{A}^{-1} \mathbf{C} \mathbf{v}_0^r + \mathbf{A}^{-1} \left( \mathbf{I} - \gamma \mathbf{A} \right)^k \left( \mathbf{A} \mathbf{u}_0^r + \mathbf{C} \mathbf{v}_0^r \right) \\
&\quad - \gamma \left( \mathbf{A} \left[ -\mathbf{A}^{-1} \mathbf{C} \mathbf{v}_0^r + \mathbf{A}^{-1} \left( \mathbf{I} - \gamma \mathbf{A} \right)^k \left( \mathbf{A} \mathbf{u}_0^r + \mathbf{C} \mathbf{v}_0 \right) \right] + \mathbf{C} \mathbf{v}_0^r \right) \\[2mm]
&= -\mathbf{A}^{-1} \mathbf{C} \mathbf{v}_0^r + \mathbf{A}^{-1} \left( \mathbf{I} - \gamma \mathbf{A} \right)^k \left( \mathbf{A} \mathbf{u}_0^r + \mathbf{C} \mathbf{v}_0^r \right) \\
&\quad - \gamma \left( -\mathbf{C} \mathbf{v}_0^r + \left( \mathbf{I} - \gamma \mathbf{A} \right)^k \left( \mathbf{A} \mathbf{u}_0^r + \mathbf{C} \mathbf{v}_0^r \right) + \mathbf{C} \mathbf{v}_0^r \right) \\[2mm]
&= -\mathbf{A}^{-1} \mathbf{C} \mathbf{v}_0^r + \mathbf{A}^{-1} \left( \mathbf{I} - \gamma \mathbf{A} \right)^k \left( \mathbf{A} \mathbf{u}_0^r + \mathbf{C} \mathbf{v}_0^r \right) - \gamma \left( \mathbf{I} - \gamma \mathbf{A} \right)^k \left( \mathbf{A} \mathbf{u}_0^r + \mathbf{C} \mathbf{v}_0^r \right) \\
&= -\mathbf{A}^{-1} \mathbf{C} \mathbf{v}_0^r + \left( \mathbf{A}^{-1} - \gamma \mathbf{I} \right) \left[ \left( \mathbf{I} - \gamma \mathbf{A} \right)^k \left( \mathbf{A} \mathbf{u}_0^r + \mathbf{C} \mathbf{v}_0^r \right) \right] \\
&= -\mathbf{A}^{-1} \mathbf{C} \mathbf{v}_0^r + \mathbf{A}^{-1} \left( \mathbf{I} - \gamma \mathbf{A} \right) \left[ \left( \mathbf{I} - \gamma \mathbf{A} \right)^k \left( \mathbf{A} \mathbf{u}_0^r + \mathbf{C} \mathbf{v}_0^r \right) \right] \\
&= -\mathbf{A}^{-1} \mathbf{C} \mathbf{v}_0^r + \mathbf{A}^{-1} \left( \mathbf{I} - \gamma \mathbf{A} \right)^{k+1} \left( \mathbf{A} \mathbf{u}_0^r + \mathbf{C} \mathbf{v}_0^r \right)
\end{aligned}
$$

Now we only need to show that our claim also works for $k = 0$,

$$
\begin{aligned}
\mathbf{u}_0^r &= -\mathbf{A}^{-1} \mathbf{C} \mathbf{v}_0^r + \mathbf{A}^{-1} \left( \mathbf{I} - \gamma \mathbf{A} \right)^0 \left( \mathbf{A} \mathbf{u}_0^r + \mathbf{C} \mathbf{v}_0^r \right) \\
&= -\mathbf{A}^{-1} \mathbf{C} \mathbf{v}_0^r + \mathbf{u}_0^r + \mathbf{A}^{-1} \mathbf{C} \mathbf{v}_0^r \\
&= \mathbf{u}_0^r
\end{aligned}
$$

Also, we do the computation with respect to $\mathbf{v}$:

$$
\mathbf{v}_k^r = \mathbf{B}^{-1} \mathbf{C}^\top \mathbf{u}_0^r + \mathbf{B}^{-1} \left( \mathbf{I} - \gamma \mathbf{B} \right)^k \left( \mathbf{B} \mathbf{v}_0^r - \mathbf{C}^\top \mathbf{u}_0^r \right)
$$

By using the update rule of Decoupled GDA we get:

$$
\begin{aligned}
\mathbf{v}_{k+1}^r &= \mathbf{v}_k - \gamma \nabla_\mathbf{v} f(\mathbf{u}_0^r, \mathbf{u}_k^r) \\
&= \mathbf{v}_k + \gamma \left( -\mathbf{B} \mathbf{v}_k^r + \mathbf{C}^\top \mathbf{u}_0^r \right) \\
&= \mathbf{v}_k - \gamma \left( \mathbf{B} \mathbf{v}_k^r - \mathbf{C}^\top \mathbf{u}_0^r \right) \\
&= \mathbf{B}^{-1} \mathbf{C}^\top \mathbf{u}_0^r + \mathbf{B}^{-1} \left( \mathbf{I} - \gamma \mathbf{B} \right)^k \left( \mathbf{B} \mathbf{v}_0^r - \mathbf{C}^\top \mathbf{u}_0 \right) \\
&\quad - \gamma \left( \mathbf{B} \left[ \mathbf{B}^{-1} \mathbf{C}^\top \mathbf{u}_0^r + \mathbf{B}^{-1} \left( \mathbf{I} - \gamma \mathbf{B} \right)^k \left( \mathbf{B} \mathbf{v}_0^r - \mathbf{C}^\top \mathbf{u}_0^r \right) \right] - \mathbf{C}^\top \mathbf{u}_0^r \right) \\[2mm]
&= \mathbf{B}^{-1} \mathbf{C}^\top \mathbf{u}_0^r + \mathbf{B}^{-1} \left( \mathbf{I} - \gamma \mathbf{B} \right)^k \left( \mathbf{B} \mathbf{v}_0^r - \mathbf{C}^\top \mathbf{u}_0 \right) \\
&\quad - \gamma \left( \mathbf{C}^\top \mathbf{u}_0^r + \left( \mathbf{I} - \gamma \mathbf{B} \right)^k \left( \mathbf{B} \mathbf{v}_0^r - \mathbf{C}^\top \mathbf{u}_0^r \right) - \mathbf{C}^\top \mathbf{u}_0^r \right) \\[2mm]
&= \mathbf{B}^{-1} \mathbf{C}^\top \mathbf{u}_0^r + \mathbf{B}^{-1} \left( \mathbf{I} - \gamma \mathbf{B} \right)^k \left( \mathbf{B} \mathbf{v}_0^r - \mathbf{C}^\top \mathbf{u}_0 \right) - \gamma \left( \mathbf{I} - \gamma \mathbf{B} \right)^k \left( \mathbf{B} \mathbf{v}_0^r - \mathbf{C}^\top \mathbf{u}_0^r \right) \\
&= \mathbf{B}^{-1} \mathbf{C}^\top \mathbf{u}_0^r + \left( \mathbf{B}^{-1} - \gamma \mathbf{I} \right) \left[ \left( \mathbf{I} - \gamma \mathbf{B} \right)^k \left( \mathbf{B} \mathbf{v}_0^r - \mathbf{C}^\top \mathbf{u}_0^r \right) \right] \\
&= \mathbf{B}^{-1} \mathbf{C}^\top \mathbf{u}_0^r + \mathbf{B}^{-1} \left( \mathbf{I} - \gamma \mathbf{B} \right) \left[ \left( \mathbf{I} - \gamma \mathbf{B} \right)^k \left( \mathbf{B} \mathbf{v}_0^r - \mathbf{C}^\top \mathbf{u}_0^r \right) \right] \\
&= \mathbf{B}^{-1} \mathbf{C}^\top \mathbf{u}_0^r + \mathbf{B}^{-1} \left( \mathbf{I} - \gamma \mathbf{B} \right)^{k+1} \left( \mathbf{B} \mathbf{v}_0^r - \mathbf{C}^\top \mathbf{u}_0^r \right)
\end{aligned}
$$

Now we only need to show this our claim also works for $k = 0$,

$$\mathbf{v}_0^r = \mathbf{B}^{-1}\mathbf{C}^\top\mathbf{u}_0^r + \mathbf{B}^{-1}\left(\mathbf{I} - \gamma\mathbf{B}\right)^0\left(\mathbf{B}\mathbf{v}_0^r - \mathbf{C}^\top\mathbf{u}_0^r\right)$$
$$= \mathbf{B}^{-1}\mathbf{C}^\top\mathbf{u}_0^r + \mathbf{v}_0^r - \mathbf{B}^{-1}\mathbf{C}^\top\mathbf{u}_0^r$$
$$= \mathbf{v}_0^r$$

$\square$

### D.3. Proof of Lemma D.2

Given a two-player quadratic game in the form of (QG). At some round $r$ after $K$ local steps with a stepsize of $\gamma \leqslant \max\{\frac{1}{L_u}, \frac{1}{L_v}\}$, the exact iterate generated by Decoupled GDA is given as follows:

$$\mathbf{x}_K^r = \left[\mathbf{Q}^K + \mathbf{E}\right]\mathbf{x}_0^r$$

$$\mathbf{Q} := \begin{pmatrix} (\mathbf{I} - \gamma\mathbf{A}) & \mathbf{0} \\ \mathbf{0} & (\mathbf{I} - \gamma\mathbf{B}) \end{pmatrix}, \quad \mathbf{E} := \begin{pmatrix} \mathbf{0} & -\mathbf{E}_u \\ \mathbf{E}_v & \mathbf{0} \end{pmatrix} \tag{42}$$

$$\mathbf{E}_u := \left[\mathbf{I} - (\mathbf{I} - \gamma\mathbf{A})^K\right]\mathbf{A}^{-1}\mathbf{C}, \quad \mathbf{E}_v := \left[\mathbf{I} - (\mathbf{I} - \gamma\mathbf{B})^K\right]\mathbf{B}^{-1}\mathbf{C}^\top$$

After taking the norm of both sides we have:

$$\|\mathbf{x}_K^r\| \leqslant \max\left\{(1 - \gamma\mu_u)^K, (1 - \gamma\mu_v)^K\right\} + \|\mathbf{C}\| \cdot \max\left\{\delta(\mathbf{A}), \delta(\mathbf{B})\right\}^{1/2}$$

$$\delta(\mathbf{A}) := \frac{(1 - (1 - \gamma L_u)^K)^2}{\mu_u^2}, \quad \delta(\mathbf{B}) := \frac{(1 - (1 - \gamma L_v)^K)^2}{\mu_v^2} \tag{43}$$

*Proof.* From Lemma D.9 we can write the explicit iterates for the variable $\mathbf{x}$:

$$\|\mathbf{x}_k^r\| = \left\|\begin{pmatrix} (\mathbf{I} - \gamma\mathbf{A}) & \mathbf{0} \\ \mathbf{0} & (\mathbf{I} - \gamma\mathbf{B}) \end{pmatrix}^k + \begin{pmatrix} \mathbf{0} & -\mathbf{E}_u \\ \mathbf{E}_v & \mathbf{0} \end{pmatrix}\right\| \cdot \|\mathbf{x}_0^r\|$$

$$\leqslant \left\|\begin{pmatrix} (\mathbf{I} - \gamma\mathbf{A}) & \mathbf{0} \\ \mathbf{0} & (\mathbf{I} - \gamma\mathbf{B}) \end{pmatrix}^k\right\| + \left\|\begin{pmatrix} \mathbf{0} & -\mathbf{E}_u \\ \mathbf{E}_v & \mathbf{0} \end{pmatrix}\right\| \cdot \|\mathbf{x}_0^r\|$$

$$\leqslant \max\left\{(1 - \gamma\mu_u)^k, (1 - \gamma\mu_v)^k\right\} \cdot \|\mathbf{x}_0^r\| + \left\|\begin{pmatrix} \mathbf{0} & -\mathbf{E}_u \\ \mathbf{E}_v & \mathbf{0} \end{pmatrix}\right\| \cdot \|\mathbf{x}_0^r\|$$

For computing the norm of the error matrix we need to compute $\sqrt{\lambda_{\max}(\mathbf{E}^\top\mathbf{E})}$. We first form $\mathbf{E}^\top\mathbf{E}$:

$$\mathbf{E}^\top\mathbf{E} = \begin{pmatrix} \mathbf{E}_v^\top\mathbf{E}_v & \mathbf{0} \\ \mathbf{0} & \mathbf{E}_u^\top\mathbf{E}_u \end{pmatrix}$$

So we have:

$$\lambda_{\max}(\mathbf{E}^\top\mathbf{E}) = \max\left\{\lambda_{\max}(\mathbf{E}_u^\top\mathbf{E}_u), \lambda_{\max}(\mathbf{E}_v^\top\mathbf{E}_v)\right\}$$

For computing the $\lambda_{\max}(\mathbf{E}_u^\top \mathbf{E}_u)$ we have:

$$
\begin{aligned}
\lambda_{\max}\left(\mathbf{E}_u^\top \mathbf{E}_u\right) &= \lambda_{\max}\left(\mathbf{C}^\top \mathbf{A}^{-\top}\left[\mathbf{I} - (\mathbf{I} - \gamma\mathbf{A})^k\right]^\top \left[\mathbf{I} - (\mathbf{I} - \gamma\mathbf{A})^k\right]\mathbf{A}^{-1}\mathbf{C}\right) \\
&\leqslant \|\mathbf{C}\|^2\, \lambda_{\max}\left(\mathbf{A}^{-\top}\left[\mathbf{I} - (\mathbf{I} - \gamma\mathbf{A})^k\right]^\top \left[\mathbf{I} - (\mathbf{I} - \gamma\mathbf{A})^k\right]\mathbf{A}^{-1}\right) \\
&\leqslant \|\mathbf{C}\|^2\, \lambda_{\max}\left(\mathbf{A}^{-\top}\right)\lambda_{\max}\left(\left[\mathbf{I} - (\mathbf{I} - \gamma\mathbf{A})^k\right]^\top\right)\lambda_{\max}\left(\left[\mathbf{I} - (\mathbf{I} - \gamma\mathbf{A})^k\right]\right)\lambda_{\max}\left(\mathbf{A}^{-1}\right) \\
&\leqslant \|\mathbf{C}\|^2\, \frac{(1 - (1 - \gamma\lambda_{\max}(\mathbf{A}))^k)^2}{\lambda_{\min}^2(\mathbf{A})} \\
&\leqslant \frac{\|\mathbf{C}\|^2}{\mu_u^2}
\end{aligned}
$$

We have the same computation with respect to player $\mathbf{v}$ as well which gives us:

$$
\begin{aligned}
\lambda_{\max}\left(\mathbf{E}_v^\top \mathbf{E}_v\right) &= \|\mathbf{C}\|^2\, \frac{(1 - (1 - \gamma\lambda_{\max}(\mathbf{B}))^k)^2}{\lambda_{\min}^2(\mathbf{B})} \\
&\leqslant \frac{\|\mathbf{C}\|^2}{\mu_v^2}
\end{aligned}
$$

Note that using the assumption $\gamma \leqslant \max\{\frac{1}{L_u}, \frac{1}{L_v}\}$ we make sure that $\delta(\mathbf{A}) \leqslant 1$ and $\delta(\mathbf{B}) \leqslant 1$. $\qquad \square$

### D.4. Proof of Theorem D.5

For any $R$ and $K \geqslant 1$ with a stepsize of $\gamma \leqslant \max\{\frac{1}{L_u}, \frac{1}{L_v}\}$ which ensures $\delta(\mathbf{A}) \leqslant 1$ and $\delta(\mathbf{B}) \leqslant 1$, after running Decoupled GDA for a total of $T = KR$ iterations on a quadratic game in the form of (QG) assuming the game is weakly coupled with $c = 1$, we get a rate of:

$$
\left\|\mathbf{x}^R - \mathbf{x}^\star\right\| \leqslant D\left(\exp\left(-\frac{\mu_{\min}}{L_{\max}}K\right) + \frac{L_{uv}}{\mu_{\min}}\right)^R \tag{44}
$$

Where $L_{\max} := \max\{L_u, L_v\}$ and $\mu_{\min} := \min\{\mu_u, \mu_v\}$.

*Proof.* Using previous Lemmas we have:

$$
\begin{aligned}
\|\mathbf{x}_k^r\| \\
&\leqslant \max\left\{(1 - \gamma\mu_u)^K, (1 - \gamma\mu_v)^K\right\}\cdot\|\mathbf{x}_0^r\| + \left\|\begin{pmatrix} \mathbf{0} & -\mathbf{E}_u \\ \mathbf{E}_v & \mathbf{0} \end{pmatrix}\right\|\cdot\|\mathbf{x}_0^r\| \\
&\leqslant \max\left\{(1 - \gamma\mu_u)^K, (1 - \gamma\mu_v)^K\right\}\cdot\|\mathbf{x}_0^r\| + \|\mathbf{C}\|\max\left\{\frac{1}{\lambda_{\min}(\mathbf{A})}, \frac{1}{\lambda_{\min}(\mathbf{B})}\right\}\cdot\|\mathbf{x}_0^r\| \\
&\leqslant \left((1 - \gamma\mu_{\min})^K + \frac{\|\mathbf{C}\|}{\mu_{\min}}\right)\cdot\|\mathbf{x}_0^r\|
\end{aligned}
$$

After unrolling the above recursion for $R$ rounds we get:

$$
\|\mathbf{x}_k^r\| \leqslant B\left((1 - \gamma\mu_{\min})^K + \frac{\|\mathbf{C}\|}{\mu_{\min}}\right)^R
$$

$\qquad \square$

## E. Additional Related Works & Discussion

### E.1. Decentralized optimization

The key difference between decentralized and distributed minimax approaches is the presence of a central server. In the former, there is no central server, and nodes communicate directly with their neighbors, whereas in the latter, a central server

aggregates the parameters. Our method belongs to the category of distributed methods. However, we will discuss later on that our approach is completely different from the general idea of distributed / federated optimization.

Decentralized optimization is widely studied for the case of minimization (Xiao & Boyd, 2004; Tsitsiklis, 1984) with the goal of not relying on a central node or server. This idea is also applied to the case of minimax optimization problems. The paper (Liu et al., 2020) is the first who studied non-convex-non-concave decentralized minimax. They also used the idea of optimistic gradient descent and achieved a rate of $\mathcal{O}(\epsilon^{-12})$. In (Xian et al., 2021), authors proposed an algorithm called DM-HSGD for non-convex decentralized minimax by utilizing variance reduction and achieved a rate of $\mathcal{O}(\kappa^3 \epsilon^{-3})$. Recently, authors in (Liu et al., 2023) proposed an algorithm named Precision for the non-convex-strongly-concave objectives which has a two-stage local updates and gives a rate of $\mathcal{O}(\frac{1}{T})$.

### E.2. Comparison Between Decoupled SGDA and Federated Minimax (Local SGDA)

In this section, we aim to highlight the key differences between our method and existing distributed or decentralized methods in the literature. As mentioned earlier, our method can be classified as distributed, though it has a major difference from others. In fact, this difference lies in the problem formulation.

**Decentralized / Distributed minimax formulation.** In these settings, we aim to solve the following finite-sum optimization problem over $M$ clients:

$$f(\mathbf{u}, \mathbf{v}) = \frac{1}{M} \sum_{m=1}^{M} f_m(\mathbf{u}, \mathbf{v}) \tag{45}$$

In the above formulation, it is assumed that each client has a different data distribution $\mathcal{D}_m$ and tries to solve the game based on this data. It means that each client keeps updating **both** $\mathbf{u}$ and $\mathbf{v}$ at the same time for several steps. Then the server aggregates the parameters and sends them back to clients. The ultimate goal is to find the saddle point $\mathbf{x}^\star = (\mathbf{u}^\star, \mathbf{v}^\star)$ of the global function $f$, as if the entire dataset $\mathcal{D} = \mathcal{D}_1 \cup \cdots \cup \mathcal{D}_M$ were on a single machine running GDA on it. In this setting, each client is allowed to update both players meaning that it has access to the gradient of $f_m$ with respect to $\mathbf{u}$ and $\mathbf{v}$. However in our approach, instead of splitting the data over clients, we split the parameter space. It means one machine is responsible for **only** updating $\mathbf{u}$ and another for $\mathbf{v}$. Our method also allows to have several machines for $\mathbf{u}$ and several machines for $\mathbf{v}$. An important point to consider is that the notions of *client* and *player* should not be intermixed. When the number of players is fixed, the distributed minimax approach essentially runs several instances $(f_m)$ of the main game $(f)$ in parallel to ultimately find the saddle point of $f$. In contrast, our method directly finds the saddle point of $f$ by splitting the parameter space across different machines. Figure 7 illustrates the difference between these two methods.

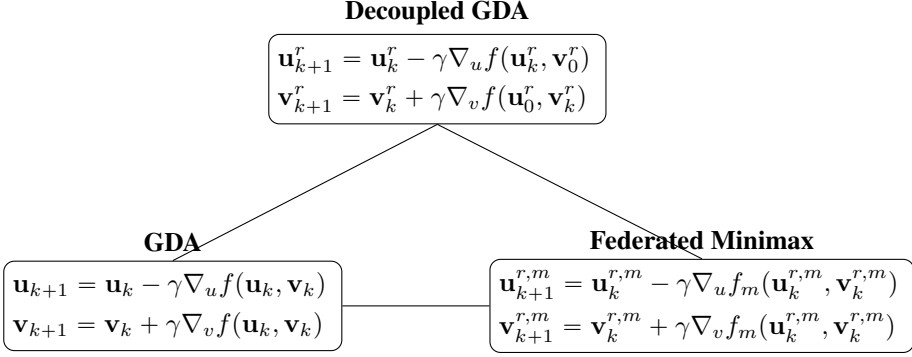

*Figure 6.* Comparison of different gradient descent ascent (GDA) approaches: Decoupled GDA, standard GDA, and Federated Minimax. The top box represents Decoupled GDA, where $\mathbf{u}$ and $\mathbf{v}$ gradients are separated, while the bottom left and right boxes represent the standard GDA and Federated Minimax approaches, respectively.

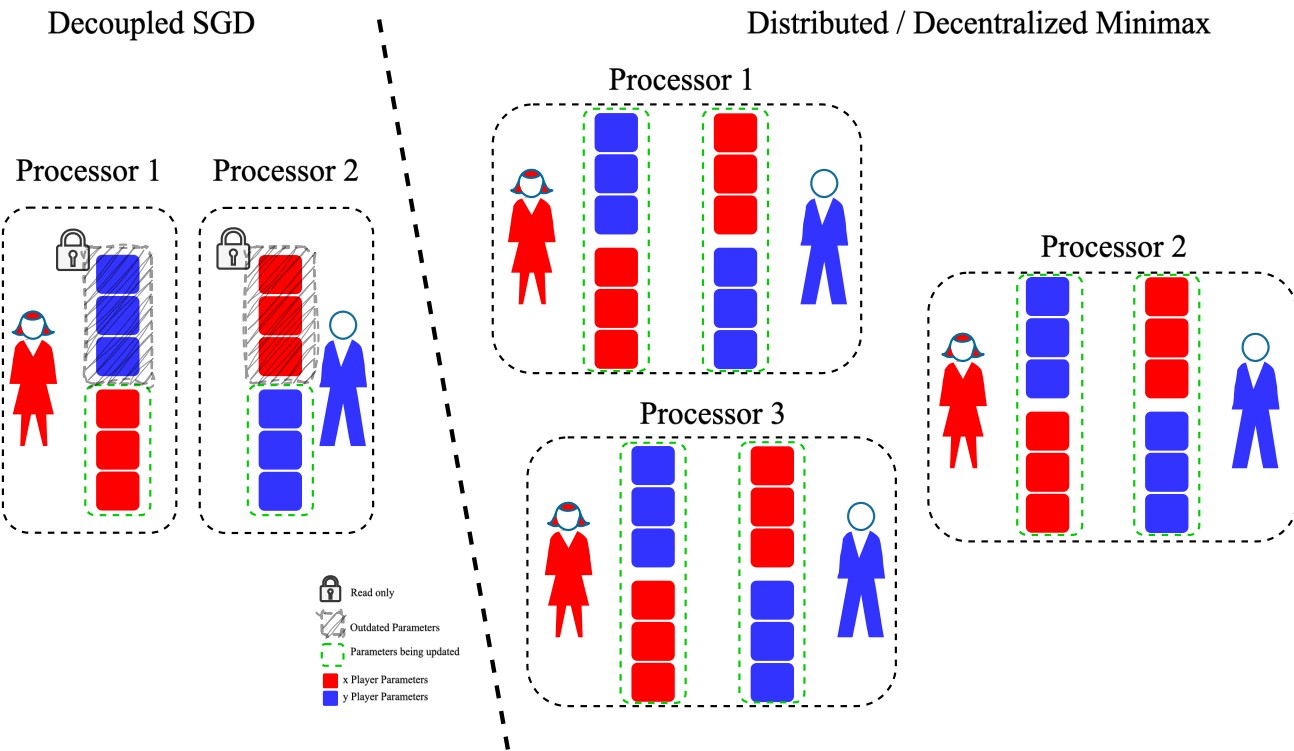

*Figure 7.* Comparison of our method with the federated minimax formulation: Our method splits the parameter space, while the federated formulation splits the data. Moreover, our method only allows each player to access the gradient with respect to their own parameters, whereas in federated minimax, each player can compute the gradient with respect to both their own parameters and the other player's parameters.

# F. Federated Decoupled SGDA

### F.1. Comparing Decoupled SGDA with Federated Learning for Minimax Optimization

Federated learning (FL) builds on the foundational work in distributed minimization, exploring various settings. In the context of minimax optimization, methods like Local SGD have been extended to achieve convergence rates for different classes of functions in both heterogeneous and homogeneous regimes. FL methods for games differ from the setting considered in this work. In FL, multiple copies of all strategies (parameters) are trained locally on different machines and datasets and periodically aggregated. FL is suited for scenarios where a single local machine runs a multi-player algorithm and has access to all players' loss functions, with "collaboration" built into the design. In contrast, our method suits competitive distributed players (local machines) where each player has noisy or outdated strategies of the remaining players. For further discussion, see Appendix E. Additionally, federated learning assumes balanced noise across players, which is not required in our setting; revisited in § 3 and § 5. Finally, in § 4, we identify a class of games where our approach leads to faster convergence, even if fully centralized training is possible, which class similarly arises in non-convex settings–§ 5.In the rest of this section, we study Federated Decoupled SGDA, which is a combination of Federated Minimax and Decoupled SGDA algorithms, and can benefit from the advantages of both approaches. In the next section we propose this method with details.

### F.2. Federated Decoupled SGDA method

In this section, we study an extension of our method in the context of federated minimax optimization aligned with the works (Deng & Mahdavi, 2021; Sharma et al., 2022). This line of work is closely related to finite-sum minimization, a well-studied topic. Local SGD (Stich, 2019a) is the most popular method used to solve finite-sum minimization problems in a distributed fashion. As an extension of this method to finite-sum minimax problems, researchers have studied Local SGDA, which is a

straightforward extension of Local SGD, incorporating gradient ascent steps in addition to gradient descent. We extend our method to this setting for the sake of completeness and provide a convergence rate that matches the state-of-the-art results for Local SGD while also improving the conditioning of the existing rates for Local SGDA.

**Notation.** We consider unconstrained two-player games denoted by $\mathbf{u}$ and $\mathbf{v}$ in the spaces $\mathcal{X}_u = \mathbb{R}^{d_u}$ and $\mathcal{X}_v = \mathbb{R}^{d_v}$, respectively. The corresponding product space $\mathcal{X} = \mathcal{X}_u \times \mathcal{X}_v = \mathbb{R}^d$ (with $d = d_u + d_v$) consists of vectors $\mathbf{x} = (\mathbf{u}, \mathbf{v}) \in \mathbb{R}^d$, where $\mathbf{u} \in \mathcal{X}_u$ and $\mathbf{v} \in \mathcal{X}_v$. For this section, we consider the common $\ell_2$ norm $\|\mathbf{x}\|_2 = \sqrt{\langle \mathbf{x}, \mathbf{x} \rangle}$.

**Problem formulation** In distributed minimax optimization, we aim to solve the following problem:

$$
\min_{\mathbf{u} \in \mathcal{X}_u} \max_{\mathbf{v} \in \mathcal{X}_v} \left[ f(\mathbf{u}, \mathbf{v}) = \frac{1}{M} \sum_{m=1}^M f_m(\mathbf{u}, \mathbf{v}) = \frac{1}{M} \sum_{m=1}^M \mathbb{E}_{\xi_m \sim \mathcal{D}_m} f_m(\mathbf{u}, \mathbf{v}, \xi_m) \right]
\tag{46}
$$

In this setting, we assume that each player's data is distributed across $M$ clients / processors. So each processor has access to a function $f_m : \mathcal{X} \to \mathbb{R}$ on which it can perform stochastic gradient steps. We denote $\mathbf{u}_k^{m,r}$ and $\mathbf{v}_k^{m,r}$ as the parameters of players $\mathbf{u}$ and $\mathbf{v}$ on client $m$ in some round $r$ after $k$ local steps. We also use the notation $\bar{\mathbf{u}}_k^r = \frac{1}{M} \sum_{m=1}^M \mathbf{u}_k^{m,r}$ and $\bar{\mathbf{v}}_k^r = \frac{1}{M} \sum_{m=1}^M \mathbf{v}_k^{m,r}$ to denote the average of parameters over clients at some round $r$ after $k$ local steps. Data distribution across processors can be either homogeneous or heterogeneous. In the heterogeneous regime, which is the case of study in this paper, each processor holds a different payoff function. To measure the heterogeneity of the problem, it's common to use the following assumption:

**Assumption F.1.** There exists a constant $\zeta_\star > 0$ satisfying the following inequality in distributed minimax games:

$$
\max \left\{ \sup_m \|\nabla_u f_m(\mathbf{x}^\star)\|^2, \sup_m \|\nabla_v f_m(\mathbf{x}^\star)\|^2 \right\} \leqslant \zeta_\star^2
\tag{47}
$$

Assumption F.1 is very common in federated learning and it has been used in many works (Koloskova et al., 2020; Deng & Mahdavi, 2021; Khaled et al., 2020). Another common assumption in the literature (Woodworth et al., 2020b; Patel et al., 2024; Zindari et al., 2023) is gradient similarity $\zeta$ for every point $\mathbf{x} \in \mathcal{X}$ which is a stronger assumption and cannot be satisfied for quadratic functions. In this work, we use Assumption F.1 to provide our convergence guarantee for our method.

**Assumption F.2.** We assume that each local function $f_m$ is $L$-smooth meaning that for all $\mathbf{u}, \mathbf{u}' \in \mathcal{X}_u$ and $\mathbf{v}, \mathbf{v}' \in \mathcal{X}_v$ it holds that:

$$
\begin{aligned}
\left\|\nabla_u f_m(\mathbf{u}, \mathbf{v}) - \nabla_u f_m(\mathbf{u}', \mathbf{v}')\right\| &\leqslant L \left[\|\mathbf{u} - \mathbf{u}'\| + \|\mathbf{v} - \mathbf{v}'\|\right] \\
\left\|\nabla_v f_m(\mathbf{u}, \mathbf{v}) - \nabla_v f_m(\mathbf{u}', \mathbf{v}')\right\| &\leqslant L \left[\|\mathbf{u} - \mathbf{u}'\| + \|\mathbf{v} - \mathbf{v}'\|\right]
\end{aligned}
\tag{48}
$$

**Assumption F.3.** We assume that the global function $f$ is $\mu$ strongly convex in $\mathbf{u}$ and $\mu$ strongly concave in $\mathbf{v}$ if for all $\mathbf{u}, \mathbf{u}' \in \mathcal{X}_u$ and $\mathbf{v}, \mathbf{v}' \in \mathcal{X}_v$ it holds that:

$$
\begin{aligned}
\langle \mathbf{u} - \mathbf{u}', \nabla_u f(\mathbf{u}, \mathbf{v}) - \nabla_u f(\mathbf{u}', \mathbf{v}) \rangle &\geqslant \mu \|\mathbf{u} - \mathbf{u}'\|^2 \\
\langle \mathbf{v} - \mathbf{v}', \nabla_v f(\mathbf{u}, \mathbf{v}') - \nabla_v f(\mathbf{u}, \mathbf{v}) \rangle &\geqslant \mu \|\mathbf{v} - \mathbf{v}'\|^2
\end{aligned}
\tag{49}
$$

**Assumption F.4.** The variance of the noise of stochastic gradients on each client is uniformly upper bounded by $\sigma^2$.

$$
\begin{aligned}
\mathbb{E}_{\xi_m} \left[\|\nabla_u f_m(\mathbf{u}, \mathbf{v}; \xi_m) - \nabla_u f_m(\mathbf{u}, \mathbf{v})\|^2\right] &\leqslant \sigma^2 \\
\mathbb{E}_{\xi_m} \left[\|\nabla_v f_m(\mathbf{u}, \mathbf{v}; \xi_m) - \nabla_v f_m(\mathbf{u}, \mathbf{v})\|^2\right] &\leqslant \sigma^2
\end{aligned}
\tag{50}
$$

**Method** Note that in this section we drop the superscript $r$ for convenience. It's expected to first define operators $F(\mathbf{x}), F_m(\mathbf{x})$ as follows:

$$
G_m(\mathbf{x}; \xi_m) := \begin{pmatrix} \nabla_u f_m(\mathbf{u}, \mathbf{v}; \xi_m) \\ -\nabla_v f_m(\mathbf{u}, \mathbf{v}; \xi_m) \end{pmatrix}, \quad G(\mathbf{x}; \xi_m) := \begin{pmatrix} \frac{1}{M} \sum_{m=1}^M \nabla_u f_m(\mathbf{u}, \mathbf{v}; \xi_m) \\ -\frac{1}{M} \sum_{m=1}^M \nabla_v f_m(\mathbf{u}, \mathbf{v}; \xi_m) \end{pmatrix}
\tag{51}
$$

Where we assume $\mathbb{E}_{\xi_m}[G_m(\mathbf{x}; \xi_m)] = F_m(\mathbf{x})$ and $\mathbb{E}_{\xi_m}[G(\mathbf{x}; \xi_m)] = F(\mathbf{x})$.

---

**Algorithm 3** Decoupled SGDA for two-player federated minimax games

1: **Input:** step size $\gamma$, initialization $\mathbf{u}_0, \mathbf{v}_0$
2: **Initialize:** $\forall m \in [M], \mathbf{u}_0^{r,0} \leftarrow \mathbf{u}_0, \quad \mathbf{y}_0^{r,0} \leftarrow \mathbf{v}_0$
3: **for** $r \in \{1, \ldots, R\}$ **do**
4: $\quad \forall m \in [M], \mathbf{u}_0^{m,r} \leftarrow \bar{\mathbf{u}}_0^r, \quad \mathbf{y}_0^{m,r} \leftarrow \bar{\mathbf{v}}_0^r$
5: $\quad$ **for** $k \in \{0, \ldots, K-1\}$ **do**
6: $\quad\quad$ **for** $m \in \{1, \ldots, M\}$ **in parallel do**
7: $\quad\quad\quad$ Update local model $\mathbf{u}_{k+1}^{m,r} \leftarrow \mathbf{u}_k^{m,r} - \gamma \nabla_u f_m(\mathbf{u}_k^{m,r}, \mathbf{v}_0^{m,r}; \xi_m)$
8: $\quad\quad\quad$ Update local model $\mathbf{v}_{k+1}^{m,r} \leftarrow \mathbf{v}_k^{m,r} + \gamma \nabla_v f_m(\mathbf{u}_0^{m,r}, \mathbf{v}_k^{m,r}; \xi_m)$
9: $\quad\quad$ **end for**
10: $\quad$ **end for**
11: $\quad \bar{\mathbf{u}}_0^{r+1} \leftarrow \frac{1}{M} \sum_{m=1}^{M} \mathbf{u}_K^{m,r}, \quad \bar{\mathbf{v}}_0^{r+1} \leftarrow \frac{1}{M} \sum_{m=1}^{M} \mathbf{v}_K^{m,r}$
12: $\quad$ **Communicate** $\bar{\mathbf{u}}_K^r$ to all processors with $\mathbf{v}$ player and $\bar{\mathbf{v}}_K^r$ to all processors with $\mathbf{u}$ player
13: **end for**
14: **Output:** $\bar{\mathbf{u}}_K^R, \bar{\mathbf{v}}_K^R$

---

*Remark* F.5. Note that $F_m(\mathbf{x}^\star) \neq 0$ and $F(\mathbf{x}^\star) = 0$.

The conventional method for solving (46) is Local SGDA which performs $K$ gradient descent and ascent local steps on each client followed by an averaging on parameters $\mathbf{u}$ and $\mathbf{v}$ over all clients which is done by a central server.

$$\mathbf{u}_K^m = \mathbf{u}_0^m - \gamma \sum_{i=0}^{K-1} \nabla_u f_m(\mathbf{u}_i^m, \mathbf{v}_i^m; \xi_m), \quad \mathbf{v}_K^m = \mathbf{v}_0^m + \gamma \sum_{i=0}^{K-1} \nabla_v f_m(\mathbf{u}_i^m, \mathbf{v}_i^m; \xi_m) \tag{52}$$

Then the server computes the average of parameters $\bar{\mathbf{u}}_K = \frac{1}{M} \sum_{m=1}^{M} \mathbf{u}_K^m$ and $\bar{\mathbf{v}}_K = \frac{1}{M} \sum_{m=1}^{M} \mathbf{v}_K^m$ and sends them back to all clients to start from these points. On the other hand, our method uses a different operator which contains the outdated gradients.

$$G_m^0(\mathbf{x}; \xi_m) := \begin{pmatrix} \nabla_u f_m(\mathbf{u}, \mathbf{v}_0; \xi_m) \\ -\nabla_v f_m(\mathbf{u}_0, \mathbf{v}; \xi_m) \end{pmatrix}, \quad G^0(\mathbf{x}; \xi_m) := \begin{pmatrix} \frac{1}{M} \sum_{m=1}^{M} \nabla_u f_m(\mathbf{u}, \mathbf{v}_0; \xi_m) \\ -\frac{1}{M} \sum_{m=1}^{M} \nabla_v f_m(\mathbf{u}_0, \mathbf{v}; \xi_m) \end{pmatrix} \tag{53}$$

Where we assume $\mathbb{E}_{\xi_m}[G_m^0(\mathbf{x}; \xi_m)] = F_m^0(\mathbf{x})$ and $\mathbb{E}_{\xi_m}[G^0(\mathbf{x}; \xi_m)] = F^0(\mathbf{x})$. The update rule of our in some round $r$ can be written as:

$$\mathbf{u}_K^m = \mathbf{u}_0^m - \gamma \sum_{i=0}^{K-1} \nabla_u f_m(\mathbf{u}_i^m, \mathbf{v}_0^m; \xi_m), \quad \mathbf{v}_K^m = \mathbf{v}_0^m + \gamma \sum_{i=0}^{K-1} \nabla_v f_m(\mathbf{u}_0^m, \mathbf{v}_i^m; \xi_m) \tag{54}$$

Assuming that all clients started with the parameters $\mathbf{u}_0 = \mathbf{u}_0^m$ and $\mathbf{v}_0 = \mathbf{v}_0^m$ at the beginning of the round.

In Algorithm 3, we discuss our method, where two players $\mathbf{u}$ and $\mathbf{v}$ have their data distributed across $M$ processors each. At every round, each set of processors update their local models while having access to an **outdated** version of the other opponent parameters which was received at the beginning of the round. By the end of the round, both set of $\mathbf{u}$ and $\mathbf{v}$ processors send the their parameters to a central server which will compute the average of the parameters and send them back to all processors.

### F.3. Convergence Guarantees

**Theorem F.6.** *For any $K, R, L > 0, \mu > 0$ after running Decoupled SGDA for a total of $T = KR$ iterations on the problems in the form of (46) in a distributed setting with $2M$ clients using a stepsize of $\gamma \leqslant \frac{\mu}{32L^2 K}$, assuming that $\|\mathbf{x}_0 - \mathbf{x}^\star\|^2 \leqslant D^2$, we have the following convergence rate:*

$$\mathbb{E}\left[\left\|\bar{\mathbf{x}}_K^R - \mathbf{x}^\star\right\|^2\right] \leqslant D^2 \exp\left(-\frac{\gamma \mu KR}{2}\right) + \frac{96K^2 L^2 \gamma^2 \zeta_\star^2}{\mu^2} + \frac{6KL^2 \gamma^2 \sigma^2}{\mu^2} + \frac{2\gamma \sigma^2}{M\mu} \tag{55}$$

**Corollary F.7.** *After choosing a stepsize of* $\gamma = \min\left\{\frac{\mu}{32KL^2}, \frac{\ln(\max\{2,\mu^2B^2KR/\sigma^2\})}{\mu KR}\right\}$, *we get a rate of:*

$$\mathbb{E}\left[\left\|\bar{\mathbf{x}}_K^R - \mathbf{x}^\star\right\|^2\right] = \tilde{\mathcal{O}}\left(D^2\exp\left(-\frac{\mu^2}{L^2}R\right) + \frac{L^2\zeta_\star^2}{\mu^4R^2} + \frac{L^2\sigma^2}{\mu^4KR^2} + \frac{\sigma^2}{\mu^2MKR}\right) \tag{56}$$

| Method | Heterogeneous | Homogeneous |
|---|---|---|
| Local SGDA 
 (Deng & Mahdavi, 2021) | $\mathcal{O}\left(\frac{L^6}{\mu^6R^3} + \frac{\sigma^2}{\mu^2MKR} + \frac{L^2\zeta_\star^2}{\mu^3MKR} + \frac{L^2\sigma^2}{\mu^3MKR}\right)$ | $\tilde{\mathcal{O}}\left(\frac{1}{K^2R^2} + \frac{\sigma^2}{\mu^2MKR} + \frac{L^2\sigma^2}{\mu^4MKR} + \frac{L^2\sigma^2}{\mu^4MK^2R^2}\right)$ |
| Local SGD 
 (Koloskova et al., 2020) 
 (Yuan & Ma, 2020) | $\mathcal{O}\left(LD^2\exp\left(-\frac{\mu}{L}R\right) + \frac{\sigma^2}{\mu MKR} + \frac{L\zeta_\star^2}{\mu^2R^2} + \frac{L\sigma^2}{\mu^2KR^2}\right)$ | $\mathcal{O}\left(LD^2\exp\left(-\frac{\mu}{L}KR\right) + \frac{\sigma^2}{\mu KMR} + \frac{Q^2\sigma^4}{\mu^5K^2R^4}\right)$ |
| **Ours** 
 Fed. Decoupled SGDA | $\tilde{\mathcal{O}}\left(D^2\exp\left(-\frac{\mu^2}{L^2}R\right) + \frac{\sigma^2}{\mu^2MKR} + \frac{L^2\zeta_\star^2}{\mu^4R^2} + \frac{L^2\sigma^2}{\mu^4KR^2}\right)$ | - |

*Table 4.* Comparison of Methods in Heterogeneous and Homogeneous Settings

Table 4 compares state-of-the-art rates for Local SGD, Local SGDA with Federated Decoupled SGDA. The first term in our rate enjoys an exponential decrease which matches the rate of Local SGD. However, the rate for Local SGDA has a rate of $\mathcal{O}\left(\frac{L^6}{\mu^6R^3}\right)$ which is worse. In addition, in this term we have a better conditioning of $\kappa^2$ compared to $\kappa^6$ in Local SGDA. Note that the condition number $\kappa^2$ in the first term of our rate matches the baseline SGDA which considering the fact that SGD has a conditioning of $\kappa$ which also appears in Local SGD due to the fact that the problem is minimization (not minimax). Our method also matches the rate of Local SGD for the noise terms and the term with heterogeneity. However, it seems that for this term the rate in (Deng & Mahdavi, 2021) contracts the existing lower bound proposed for Local SGD in (Patel et al., 2024) which is:

$$\mathbb{E}[f(\mathbf{x}^R) - f(\mathbf{x}^\star)] \geq \frac{LD^2}{R} + \frac{(L\sigma^2D^4)^{1/3}}{K^{1/3}R^{2/3}} + \frac{\sigma D}{\sqrt{MKR}} + \frac{(L\zeta_\star^2D^4)^{1/3}}{R^{2/3}} \tag{57}$$

As it's clear from the lower bound, the term with heterogeneity cannot be improved using local steps (there is no $K$ in the denominator) while in the rate of (Deng & Mahdavi, 2021) this term can be decreased if $K \to \infty$, which contracts the lower bound.

### F.4. Missing Proofs for Section F

**Lemma F.8.** *For a set of $M$ vectors $\mathbf{a}_1, \mathbf{a}_2, \ldots, \mathbf{a}_M \in \mathbb{R}^d$ we have:*

$$\left\|\sum_{m=1}^M \mathbf{a}_m\right\| \leq \sum_{m=1}^M \|\mathbf{a}_m\| . \tag{58}$$

**Lemma F.9.** *For a set of $M$ vectors $\mathbf{a}_1, \mathbf{a}_2, \ldots, \mathbf{a}_M \in \mathbb{R}^d$ we have:*

$$\left\|\sum_{m=1}^M \mathbf{a}_m\right\|^2 \leq M\sum_{m=1}^M \|\mathbf{a}_m\|^2 . \tag{59}$$

**Lemma F.10.** *For two arbitrary vectors $\mathbf{a}, \mathbf{b} \in \mathbb{R}^d$ and $\forall \gamma > 0$ we have:*

$$\|\mathbf{a} + \mathbf{b}\|^2 \leq (1+\gamma)\|\mathbf{a}\|^2 + (1+\gamma^{-1})\|\mathbf{b}\|^2 . \tag{60}$$

**Lemma F.11.** *Let Assumption 2.4 holds. Then we have:*

$$\mathbb{E}_{\xi_m}\left\|\frac{1}{M}\sum_{m=1}^M \nabla_u f_m(\mathbf{u}, \mathbf{v}, \xi_m) - \frac{1}{M}\sum_{m=1}^M \nabla_u f_m(\mathbf{u}, \mathbf{v})\right\|^2 \leq \frac{\sigma^2}{M} . \tag{61}$$

*The same argument holds for gradient with respect to $\mathbf{v}$.*

**Lemma F.12** (Consensus Error). *After running Decoupled Local SGDA for $k$ local steps at some round $r$ with a step-size of $\gamma \leqslant \frac{\mu}{32L^2K}$, the error $\Psi(\mathbf{x}_k^{m,r}) + \Phi(\bar{\mathbf{x}}_k^r)$ can be upper bounded as follows:*

$$\mathbb{E}[\Psi(\mathbf{x}_k^{m,r}) + \Phi(\bar{\mathbf{x}}_k^r)] \leqslant \sum_{i=1}^{K} \frac{\mu^2}{8KL^2} \|\bar{\mathbf{x}}_i^r - \mathbf{x}^\star\|^2 + 32K^2\gamma^2\zeta_\star^2 + \frac{2K\gamma^2\sigma^2}{M} + 2K\gamma^2\sigma^2 \tag{62}$$

In this setting, we have two different errors related to the use of outdated gradients and deviation from the average iterates. Total error is the sum of both errors. We define the consensus error in this setting as follows:

$$\Psi(\mathbf{u}_k^{m,r}) := \frac{1}{M}\sum_{m=1}^{M} \|\mathbf{u}_k^{m,r} - \bar{\mathbf{u}}_k^r\|^2, \quad \Psi(\mathbf{v}_k^{m,r}) := \frac{1}{M}\sum_{m=1}^{M} \|\mathbf{v}_k^{m,r} - \bar{\mathbf{v}}_k^r\|^2$$

$$\Phi(\bar{\mathbf{u}}_k^r) := \|\bar{\mathbf{u}}_0^r - \bar{\mathbf{u}}_k^r\|^2, \quad \Phi(\bar{\mathbf{v}}_k^r) := \|\bar{\mathbf{v}}_0^r - \bar{\mathbf{v}}_k^r\|^2$$

$$\Psi(\mathbf{x}_k^{m,r}) = \Psi(\mathbf{u}_k^{m,r}) + \Psi(\mathbf{v}_k^{m,r}), \quad \Phi(\bar{\mathbf{x}}_k^r) = \Phi(\bar{\mathbf{u}}_k^r) + \Phi(\bar{\mathbf{v}}_k^r)$$

The total consensus error can be computed by summing both errors with respect to $\mathbf{u}$ and $\mathbf{v}$:

$$\text{Consensus error} := \boxed{\begin{array}{c} \Psi(\mathbf{u}_k) + \Psi(\mathbf{v}_k) \\ \text{error caused by} \\ \text{deviation from average} \end{array}} + \boxed{\begin{array}{c} \Phi(\mathbf{u}_k) + \Phi(\mathbf{v}_k) \\ \text{error caused by} \\ \text{outdated gradients} \end{array}}$$

In the following, the upper bound for consensus error in different settings will be discussed. Note that in the case of multi client, we get different upper bounds based on the assumption on data heterogeneity.

*Proof.*

$$\mathbb{E}[\Psi(\mathbf{u}_{k+1}^{m,r}) + \Phi(\bar{\mathbf{u}}_{k+1}^r)]$$

$$= \frac{1}{M} \sum_{m=1}^{M} \mathbb{E} \left\| \mathbf{u}_k^{m,r} - \gamma \nabla_u f_m(\mathbf{u}_k^{m,r}, \mathbf{v}_0^{m,r}; \xi_m) - \bar{\mathbf{u}}_k^r + \frac{\gamma}{M} \sum_{m=1}^{M} \nabla_u f_m(\mathbf{u}_k^{m,r}, \mathbf{v}_0^{m,r}; \xi_m) \right\|^2 +$$

$$\mathbb{E} \left\| \bar{\mathbf{u}}_0^r - \bar{\mathbf{u}}_k^r + \frac{\gamma}{M} \sum_{m=1}^{M} \nabla_u f_m(\mathbf{u}_k^{m,r}, \mathbf{v}_0^{m,r}; \xi_m) \right\|^2$$

$$= \frac{1}{M} \sum_{m=1}^{M} \mathbb{E} \left\| \mathbf{u}_k^{m,r} - \gamma \nabla_u f_m(\mathbf{u}_k^{m,r}, \mathbf{v}_0^{m,r}) - \bar{\mathbf{u}}_k^r + \frac{\gamma}{M} \sum_{m=1}^{M} \nabla_u f_m(\mathbf{u}_k^{m,r}, \mathbf{v}_0^{m,r}) \right\|^2 +$$

$$\mathbb{E} \left\| \bar{\mathbf{u}}_0^r - \bar{\mathbf{u}}_k^r + \frac{\gamma}{M} \sum_{m=1}^{M} \nabla_u f_m(\mathbf{u}_k^{m,r}, \mathbf{v}_0^{m,r}) \right\|^2 + \frac{\gamma^2 \sigma^2}{M} + \gamma^2 \sigma^2$$

$$\leqslant \left(1 + \frac{1}{K}\right) \mathbb{E}[\Psi(\mathbf{u}_k^{m,r}) + \Phi(\bar{\mathbf{u}}_k^r)] + \frac{2K\gamma^2}{M} \sum_{m=1}^{M} \mathbb{E} \left\| \nabla_u f_m(\mathbf{u}_k^{m,r}, \mathbf{v}_0^{m,r}) - \frac{1}{M} \sum_{m=1}^{M} \nabla_u f_m(\mathbf{u}_k^{m,r}, \mathbf{v}_0^{m,r}) \right\|^2 +$$

$$\frac{2K\gamma^2}{M} \sum_{m=1}^{M} \mathbb{E} \left\| \nabla_u f_m(\mathbf{u}_k^{m,r}, \mathbf{v}_0^{m,r}) \right\|^2 + \frac{\gamma^2 \sigma^2}{M} + \gamma^2 \sigma^2$$

$$\leqslant \left(1 + \frac{1}{K}\right) \mathbb{E}[\Psi(\mathbf{u}_k^{m,r}) + \Phi(\bar{\mathbf{u}}_k^r)] + \frac{4K\gamma^2}{M} \sum_{m=1}^{M} \mathbb{E} \left\| \nabla_u f_m(\mathbf{u}_k^{m,r}, \mathbf{v}_0^{m,r}) \right\|^2 + \frac{\gamma^2 \sigma^2}{M} + \gamma^2 \sigma^2$$

$$= \left(1 + \frac{1}{K}\right) \mathbb{E}[\Psi(\mathbf{u}_k^{m,r}) + \Phi(\bar{\mathbf{u}}_k^r)] +$$

$$\frac{4K\gamma^2}{M} \sum_{m=1}^{M} \mathbb{E} \left\| \nabla_u f_m(\mathbf{u}_k^{m,r}, \mathbf{v}_0^{m,r}) - \nabla_u f_m(\bar{\mathbf{u}}_k^r, \bar{\mathbf{v}}_k^r) + \nabla_u f_m(\bar{\mathbf{u}}_k^r, \bar{\mathbf{v}}_k^r) \right\|^2 + \frac{\gamma^2 \sigma^2}{M} + \gamma^2 \sigma^2$$

$$\leqslant \left(1 + \frac{1}{K}\right) \mathbb{E}[\Psi(\mathbf{u}_k^{m,r}) + \Phi(\bar{\mathbf{u}}_k^r)] + \frac{8K\gamma^2}{M} \sum_{m=1}^{M} \mathbb{E} \left\| \nabla_u f_m(\mathbf{u}_k^{m,r}, \mathbf{v}_0^{m,r}) - \nabla_u f_m(\bar{\mathbf{u}}_k^r, \bar{\mathbf{v}}_k^r) \right\|^2 +$$

$$\frac{8K\gamma^2}{M} \sum_{m=1}^{M} \mathbb{E} \left\| \nabla_u f_m(\bar{\mathbf{u}}_k^r, \bar{\mathbf{v}}_k^r) \right\|^2 + \frac{\gamma^2 \sigma^2}{M} + \gamma^2 \sigma^2$$

$$\leqslant \left(1 + \frac{1}{K}\right) \mathbb{E}[\Psi(\mathbf{u}_k^{m,r}) + \Phi(\bar{\mathbf{u}}_k^r)] + 8KL^2\gamma^2 \mathbb{E}[\Psi(\mathbf{u}_k^{m,r})] + 8KL^2\gamma^2 \mathbb{E}[\Phi(\bar{\mathbf{v}}_k^r)] +$$

$$\frac{8K\gamma^2}{M} \sum_{m=1}^{M} \mathbb{E} \left\| \nabla_u f_m(\bar{\mathbf{u}}_k^r, \bar{\mathbf{v}}_k^r) \right\|^2 + \frac{\gamma^2 \sigma^2}{M} + \gamma^2 \sigma^2$$

$$= \left(1 + \frac{1}{K}\right) \mathbb{E}[\Psi(\mathbf{u}_k^{m,r}) + \Phi(\bar{\mathbf{u}}_k^r)] + 8KL^2\gamma^2 \mathbb{E}[\Psi(\mathbf{u}_k^{m,r}) + \Phi(\bar{\mathbf{v}}_k^r)] +$$

$$\frac{8K\gamma^2}{M} \sum_{m=1}^{M} \mathbb{E} \left\| \nabla_u f_m(\bar{\mathbf{u}}_k^r, \bar{\mathbf{v}}_k^r) - \nabla_u f_m(\mathbf{u}^\star, \mathbf{v}^\star) + \nabla_u f_m(\mathbf{u}^\star, \mathbf{v}^\star) \right\|^2 + \frac{\gamma^2 \sigma^2}{M} + \gamma^2 \sigma^2$$

$$\left(1 + \frac{1}{K}\right) \mathbb{E}[\Psi(\mathbf{u}_k^{m,r}) + \Phi(\bar{\mathbf{u}}_k^r)] + 8KL^2\gamma^2 \mathbb{E}[\Psi(\mathbf{u}_k^{m,r}) + \Phi(\bar{\mathbf{v}}_k^r)] +$$

$$\frac{16K\gamma^2}{M} \sum_{m=1}^{M} \mathbb{E} \left\| \nabla_u f_m(\bar{\mathbf{u}}_k^r, \bar{\mathbf{v}}_k^r) - \nabla_u f_m(\mathbf{u}^\star, \mathbf{v}^\star) \right\|^2 + 16K\gamma^2 \zeta_\star^2 + \frac{\gamma^2 \sigma^2}{M} + \gamma^2 \sigma^2$$

we continue:

$$\mathbb{E}[\Psi(\mathbf{u}_{k+1}^{m,r}) + \Phi(\bar{\mathbf{u}}_{k+1}^r)] \leqslant \left(1 + \frac{1}{K}\right) \mathbb{E}[\Psi(\mathbf{u}_k^{m,r}) + \Phi(\bar{\mathbf{u}}_k^r)] + 8KL^2\gamma^2 \mathbb{E}[\Psi(\mathbf{u}_k^{m,r}) + \Phi(\bar{\mathbf{v}}_k^r)] +$$

$$16KL^2\gamma^2 \mathbb{E}\left\|\bar{\mathbf{x}}_k^r - \mathbf{x}^\star\right\|^2 + 16K\gamma^2\zeta_\star^2 + \frac{\gamma^2\sigma^2}{M} + \gamma^2\sigma^2$$

After doing the same computation with respect to $\mathbf{v}$ we get:

$$\mathbb{E}[\Psi(\mathbf{v}_{k+1}^{m,r}) + \Phi(\bar{\mathbf{v}}_{k+1}^r)]$$

$$\leqslant \left(1 + \frac{1}{K}\right) \mathbb{E}[\Psi(\mathbf{v}_k^{m,r}) + \Phi(\bar{\mathbf{v}}_k^r)] + 8KL^2\gamma^2 \mathbb{E}[\Psi(\mathbf{v}_k^{m,r}) + \Phi(\bar{\mathbf{u}}_k^r)] +$$

$$16KL^2\gamma^2 \mathbb{E}\left\|\bar{\mathbf{x}}_k^r - \mathbf{x}^\star\right\|^2 + 16K\gamma^2\zeta_\star^2 + \frac{\gamma^2\sigma^2}{M} + \gamma^2\sigma^2$$

Now we sum up both inequalities and we get:

$$\mathbb{E}[\Psi(\mathbf{x}_{k+1}^{m,r}) + \Phi(\bar{\mathbf{x}}_{k+1}^r)]$$

$$\leqslant \left(1 + \frac{1}{K}\right) \mathbb{E}[\Psi(\mathbf{x}_k^{m,r}) + \Phi(\bar{\mathbf{x}}_k^r)] + 8KL^2\gamma^2 \mathbb{E}[\Psi(\mathbf{x}_k^{m,r}) + \Phi(\bar{\mathbf{x}}_k^r)] +$$

$$32KL^2\gamma^2 \mathbb{E}\left\|\bar{\mathbf{x}}_k^r - \mathbf{x}^\star\right\|^2 + 32K\gamma^2\zeta_\star^2 + \frac{2\gamma^2\sigma^2}{M} + 2\gamma^2\sigma^2$$

With the choice of $\gamma \leqslant \frac{\mu}{32L^2K}$ we simplify the above inequality as:

$$\mathbb{E}[\Psi(\mathbf{x}_{k+1}^{m,r}) + \Phi(\bar{\mathbf{x}}_{k+1}^r)]$$

$$\leqslant \left(1 + \frac{1}{K} + \frac{1}{128K}\right) \mathbb{E}[\Psi(\mathbf{x}_k^{m,r}) + \Phi(\bar{\mathbf{x}}_k^r)] + \frac{\mu^2}{32KL^2} \mathbb{E}\left\|\bar{\mathbf{x}}_k^r - \mathbf{x}^\star\right\|^2 + 32K\gamma^2\zeta_\star^2 + \frac{2\gamma^2\sigma^2}{M} + 2\gamma^2\sigma^2$$

After unrolling the recursion for the last $K$ steps and considering the fact that $\left(1 + \frac{1}{K} + \frac{1}{128K}\right)^K \leqslant 4$ we have:

$$\mathbb{E}[\Psi(\mathbf{x}_{k+1}^{m,r}) + \Phi(\bar{\mathbf{x}}_{k+1}^r)] \leqslant \sum_{i=1}^{K} \frac{\mu^2}{8KL^2} \mathbb{E}\left\|\bar{\mathbf{x}}_i^r - \mathbf{x}^\star\right\|^2 + 32K^2\gamma^2\zeta_\star^2 + \frac{2K\gamma^2\sigma^2}{M} + 2K\gamma^2\sigma^2$$

$\square$

### F.5. Proof of Theorem F.6

For any $K, R, L > 0, \mu > 0$ after running Decoupled SGDA for a total of $T = KR$ iterations on the problems in the form of (46) in a distributed setting with $2M$ clients using a stepsize of $\gamma \leqslant \frac{\mu}{32L^2K}$, assuming that $\left\|\mathbf{x}_0 - \mathbf{x}^\star\right\|^2 \leqslant B^2$, we have the following convergence rate:

$$\mathbb{E}\left\|\bar{\mathbf{x}}_K^R - \mathbf{x}^\star\right\|^2 \leqslant B^2 \exp\left(-\frac{\gamma\mu KR}{2}\right) + \frac{96K^2L^2\gamma^2\zeta_\star^2}{\mu^2} + \frac{6KL^2\gamma^2\sigma^2}{\mu^2} + \frac{2\gamma\sigma^2}{M\mu} \tag{63}$$

*Proof.* We start by upper bounding the distance between the average iterate $\bar{\mathbf{u}}_{k+1}^r$ and the saddle point.

$$
\begin{aligned}
&\mathbb{E}\left\|\bar{\mathbf{u}}_{k+1}^r - \mathbf{u}^\star\right\|^2 \\
&= \mathbb{E}\left\|\bar{\mathbf{u}}_k^r - \frac{\gamma}{M}\sum_{m=1}^M \nabla_u f_m(\mathbf{u}_k^{m,r}, \mathbf{v}_0^{m,r}; \xi_m) - \mathbf{u}^\star\right\|^2 \\
&\leqslant \mathbb{E}\left\|\bar{\mathbf{u}}_k^r - \frac{\gamma}{M}\sum_{m=1}^M \nabla_u f_m(\mathbf{u}_k^{m,r}, \mathbf{v}_0^{m,r}) - \mathbf{u}^\star\right\|^2 + \frac{\gamma^2\sigma^2}{M} \\
&= \mathbb{E}\left\|\bar{\mathbf{u}}_k^r + \frac{\gamma}{M}\sum_{m=1}^M \nabla_u f_m(\bar{\mathbf{u}}_k^r, \bar{\mathbf{v}}_k^r) - \frac{\gamma}{M}\sum_{m=1}^M \nabla_u f_m(\mathbf{u}_k^{m,r}, \mathbf{v}_0^{m,r}) - \frac{\gamma}{M}\sum_{m=1}^M \nabla_u f_m(\bar{\mathbf{u}}_k^r, \bar{\mathbf{v}}_k^r) - \mathbf{u}^\star\right\|^2 + \frac{\gamma^2\sigma^2}{M} \\
&\leqslant \left(1+\frac{\gamma\mu}{2}\right)\mathbb{E}\left\|\bar{\mathbf{u}}_k^r - \frac{\gamma}{M}\sum_{m=1}^M \nabla_u f_m(\bar{\mathbf{u}}_k^r, \bar{\mathbf{v}}_k^r) - \mathbf{u}^\star\right\|^2 + \\
&\qquad \left(1+\frac{2}{\gamma\mu}\right)\frac{\gamma^2}{M}\sum_{m=1}^M \mathbb{E}\left\|\nabla_u f_m(\bar{\mathbf{u}}_k^r, \bar{\mathbf{v}}_k^r) - \nabla_u f_m(\mathbf{u}_k^{m,r}, \mathbf{v}_0^{m,r})\right\|^2 + \frac{\gamma^2\sigma^2}{M}
\end{aligned}
$$

For the first term in the above inequality we have:

$$
\begin{aligned}
&\left(1+\frac{\gamma\mu}{2}\right)\mathbb{E}\left\|\bar{\mathbf{u}}_k^r - \frac{\gamma}{M}\sum_{m=1}^M \nabla_u f_m(\bar{\mathbf{u}}_k^r, \bar{\mathbf{v}}_k^r) - \mathbf{u}^\star\right\|^2 \\
&= \left(1+\frac{\gamma\mu}{2}\right)\mathbb{E}\left\|\bar{\mathbf{u}}_k^r - \gamma\nabla_u f(\bar{\mathbf{u}}_k^r, \bar{\mathbf{v}}_k^r) - \mathbf{u}^\star\right\|^2 \\
&= \left(1+\frac{\gamma\mu}{2}\right)\mathbb{E}\left[\left\|\bar{\mathbf{u}}_k^r - \mathbf{u}^\star\right\|^2 + \gamma^2\left\|\nabla_u f(\bar{\mathbf{u}}_k^r, \bar{\mathbf{v}}_k^r)\right\|^2 - 2\gamma\langle\bar{\mathbf{u}}_k^r - \mathbf{u}^\star, \nabla_u f(\bar{\mathbf{u}}_k^r, \bar{\mathbf{v}}_k^r)\rangle\right] \\
&\leqslant \left(1+\frac{\gamma\mu}{2}\right)\mathbb{E}\left[(1+\gamma^2 L^2)\left\|\bar{\mathbf{u}}_k^r - \mathbf{u}^\star\right\|^2 - 2\gamma\langle\bar{\mathbf{u}}_k^r - \mathbf{u}^\star, \nabla_u f(\bar{\mathbf{u}}_k^r, \bar{\mathbf{v}}_k^r)\rangle\right]
\end{aligned}
$$

For the second term we also have:

$$
\begin{aligned}
&\left(1+\frac{2}{\gamma\mu}\right)\frac{\gamma^2}{M}\sum_{m=1}^M \mathbb{E}\left\|\nabla_u f_m(\bar{\mathbf{u}}_k^r, \bar{\mathbf{v}}_k^r) - \nabla_u f_m(\mathbf{u}_k^{m,r}, \mathbf{v}_0^{m,r})\right\|^2 \\
&\leqslant \left(1+\frac{2}{\gamma\mu}\right)\frac{L^2\gamma^2}{M}\sum_{m=1}^M \mathbb{E}\left\|\bar{\mathbf{u}}_k^r - \mathbf{u}_k^{m,r}\right\|^2 + \left(1+\frac{2}{\gamma\mu}\right)\frac{L^2\gamma^2}{M}\sum_{m=1}^M \mathbb{E}\left\|\bar{\mathbf{v}}_k^r - \bar{\mathbf{v}}_0^r\right\|^2 \\
&= \left(1+\frac{2}{\gamma\mu}\right)L^2\gamma^2\,\mathbb{E}\left[\Psi(\mathbf{u}_k^{m,r})\right] + \left(1+\frac{2}{\gamma\mu}\right)L^2\gamma^2\,\mathbb{E}[\Phi(\bar{\mathbf{v}}_k^r)]
\end{aligned}
$$

Where in the last line, we used the fact that $\mathbf{v}_0^{m,r} = \bar{\mathbf{v}}_0^r$. We then repeat the same computation with respect to $\mathbf{v}$.

$$
\mathbb{E}\left\|\bar{\mathbf{v}}_{k+1}^r - \mathbf{v}^\star\right\|^2 =
$$

$$
= \mathbb{E}\left\|\bar{\mathbf{v}}_k^r + \frac{\gamma}{M}\sum_{m=1}^M \nabla_v f_m(\mathbf{u}_0^{m,r}, \mathbf{v}_k^{m,r}; \xi_m) - \mathbf{v}^\star\right\|^2
$$

$$
\leqslant \mathbb{E}\left\|\bar{\mathbf{v}}_k^r + \frac{\gamma}{M}\sum_{m=1}^M \nabla_v f_m(\mathbf{u}_0^{m,r}, \mathbf{v}_k^{m,r}) - \mathbf{v}^\star\right\|^2 + \frac{\gamma\sigma^2}{M}
$$

$$
= \mathbb{E}\left\|\bar{\mathbf{v}}_k^r + \frac{\gamma}{M}\sum_{m=1}^M \nabla_v f_m(\mathbf{u}_0^{m,r}, \mathbf{v}_k^{m,r}) - \frac{\gamma}{M}\sum_{m=1}^M \nabla_v f_m(\bar{\mathbf{u}}_k^r, \bar{\mathbf{v}}_k^r) + \frac{\gamma}{M}\sum_{m=1}^M \nabla_v f_m(\bar{\mathbf{u}}_k^r, \bar{\mathbf{v}}_k^r) - \mathbf{v}^\star\right\|^2 + \frac{\gamma\sigma^2}{M}
$$

$$
\leqslant \left(1 + \frac{\gamma\mu}{2}\right)\mathbb{E}\left\|\bar{\mathbf{v}}_k^r + \frac{\gamma}{M}\sum_{m=1}^M \nabla_v f_m(\bar{\mathbf{u}}_k^r, \bar{\mathbf{v}}_k^r) - \mathbf{v}^\star\right\|^2 +
$$

$$
\left(1 + \frac{2}{\gamma\mu}\right)\frac{\gamma^2}{M}\sum_{m=1}^M \mathbb{E}\left\|\nabla_v f_m(\bar{\mathbf{u}}_k^r, \bar{\mathbf{v}}_k^r) - \nabla_v f_m(\mathbf{u}_0^{m,r}, \mathbf{v}_k^{m,r})\right\|^2 + \frac{\gamma\sigma^2}{M}
$$

For the first term in the above inequality we have:

$$
\left(1 + \frac{\gamma\mu}{2}\right)\mathbb{E}\left\|\bar{\mathbf{v}}_k^r + \frac{\gamma}{M}\sum_{m=1}^M \nabla_v f_m(\bar{\mathbf{u}}_k^r, \bar{\mathbf{v}}_k^r) - \mathbf{v}^\star\right\|^2
$$

$$
= \left(1 + \frac{\gamma\mu}{2}\right)\mathbb{E}\left\|\bar{\mathbf{v}}_k^r + \gamma\nabla_v f(\bar{\mathbf{u}}_k^r, \bar{\mathbf{v}}_k^r) - \mathbf{v}^\star\right\|^2
$$

$$
= \left(1 + \frac{\gamma\mu}{2}\right)\mathbb{E}\left[\left\|\bar{\mathbf{v}}_k^r - \mathbf{v}^\star\right\|^2 + \gamma^2\left\|\nabla_v f(\bar{\mathbf{u}}_k^r, \bar{\mathbf{v}}_k^r)\right\|^2 - 2\gamma\langle\mathbf{v}^\star - \bar{\mathbf{v}}_k^r, \nabla_v f(\bar{\mathbf{u}}_k^r, \bar{\mathbf{v}}_k^r)\rangle\right]
$$

$$
\leqslant \left(1 + \frac{\gamma\mu}{2}\right)\mathbb{E}\left[(1 + \gamma^2 L^2)\left\|\bar{\mathbf{v}}_k^r - \mathbf{v}^\star\right\|^2 - 2\gamma\langle\mathbf{v}^\star - \bar{\mathbf{v}}_k^r, \nabla_v f(\bar{\mathbf{u}}_k^r, \bar{\mathbf{v}}_k^r)\rangle\right]
$$

For the second term we also have:

$$
\left(1 + \frac{2}{\gamma\mu}\right)\frac{\gamma^2}{M}\sum_{m=1}^M \mathbb{E}\left\|\nabla_v f_m(\bar{\mathbf{u}}_k^r, \bar{\mathbf{v}}_k^r) - \nabla_v f_m(\mathbf{u}_0^{m,r}, \mathbf{v}_k^{m,r})\right\|^2
$$

$$
\leqslant \left(1 + \frac{2}{\gamma\mu}\right)\frac{L^2\gamma^2}{M}\sum_{m=1}^M \mathbb{E}\left\|\bar{\mathbf{u}}_k^r - \bar{\mathbf{u}}_0^r\right\|^2 + \left(1 + \frac{2}{\gamma\mu}\right)\frac{L^2\gamma^2}{M}\sum_{m=1}^M \mathbb{E}\left\|\bar{\mathbf{v}}_k^r - \mathbf{v}_k^{m,r}\right\|^2
$$

$$
= \left(1 + \frac{2}{\gamma\mu}\right)L^2\gamma^2\,\mathbb{E}[\Phi(\bar{\mathbf{u}}_k^r)] + \left(1 + \frac{2}{\gamma\mu}\right)L^2\gamma^2\,\mathbb{E}[\Psi(\mathbf{v}_k^{m,r})]
$$

Summing up the results from the inequalities with respect to $\mathbf{u}$ and $\mathbf{v}$ gives us:

$$
\mathbb{E}\left\|\bar{\mathbf{x}}_{k+1}^r - \mathbf{x}^\star\right\|^2
$$

$$
\leqslant \left(1 + \frac{\gamma\mu}{2}\right)\mathbb{E}\left[(1 + \gamma^2 L^2)\left\|\bar{\mathbf{x}}_k^r - \mathbf{x}^\star\right\|^2 - 2\gamma\langle\bar{\mathbf{x}}_k^r - \mathbf{x}^\star, F(\bar{\mathbf{x}}_k^r)\rangle\right] + \gamma\left(\gamma L^2 + \frac{2L^2}{\mu}\right)\mathbb{E}\left[\Phi(\bar{\mathbf{x}}_k^r) + \Psi(\mathbf{x}_k^{m,r})\right] + \frac{\gamma^2\sigma^2}{M}
$$

$$
\leqslant \left(1 + \frac{\gamma\mu}{2}\right)\mathbb{E}\left[(1 + \gamma^2 L^2)\left\|\bar{\mathbf{x}}_k^r - \mathbf{x}^\star\right\|^2 - 2\gamma\mu\left\|\bar{\mathbf{x}}_k^r - \mathbf{x}^\star\right\|^2\right] + \gamma\left(\gamma L^2 + \frac{2L^2}{\mu}\right)\mathbb{E}\left[\Phi(\bar{\mathbf{x}}_k^r) + \Psi(\mathbf{x}_k^{m,r})\right] + \frac{\gamma^2\sigma^2}{M}
$$

$$
= \left(1 + \frac{\gamma\mu}{2}\right)\mathbb{E}\left[(1 - 2\gamma\mu + \gamma^2 L^2)\left\|\bar{\mathbf{x}}_k^r - \mathbf{x}^\star\right\|^2\right] + \gamma\left(\gamma L^2 + \frac{2L^2}{\mu}\right)\mathbb{E}\left[\Phi(\bar{\mathbf{x}}_k^r) + \Psi(\mathbf{x}_k^{m,r})\right] + \frac{\gamma^2\sigma^2}{M}
$$

With the choice of $\gamma \leqslant \frac{\mu}{16L^2}$ we have:

$$
\begin{aligned}
&\mathbb{E}\left\|\bar{\mathbf{x}}_{k+1}^r - \mathbf{x}^\star\right\|^2 \\
&\leqslant \left(1 - \frac{23\gamma\mu}{16}\right) \mathbb{E}\left\|\bar{\mathbf{x}}_k^r - \mathbf{x}^\star\right\|^2 + \frac{33\gamma L^2}{16\mu} \mathbb{E}\left[\Phi(\bar{\mathbf{x}}_k^r) + \Psi(\mathbf{x}_k^{m,r})\right] + \frac{\gamma^2\sigma^2}{M} \\
&\leqslant \left(1 - \frac{23\gamma\mu}{16}\right) \mathbb{E}\left\|\bar{\mathbf{x}}_k^r - \mathbf{x}^\star\right\|^2 + \frac{33\gamma\mu}{128K} \sum_{i=1}^{K} \left\|\bar{\mathbf{x}}_i^r - \mathbf{x}^\star\right\|^2 + \frac{96K^2L^2\gamma^3\zeta_\star^2}{\mu} + \frac{7KL^2\gamma^3\sigma^2}{\mu M} + \frac{6KL^2\gamma^3\sigma^2}{\mu} + \frac{\gamma^2\sigma^2}{M}
\end{aligned}
$$

We change the current notation for simplicity in proof by substituting $r$ and $k$ with $t$. $t$ varies from $0$ to $T = KR$, iterating over all rounds and local steps:

$$
\begin{aligned}
\mathbb{E}\left\|\bar{\mathbf{x}}_{t+1} - \mathbf{x}^\star\right\|^2 &\leqslant \left(1 - \frac{23\gamma\mu}{16}\right) \mathbb{E}\left\|\bar{\mathbf{x}}_t - \mathbf{x}^\star\right\|^2 + \frac{33\gamma\mu}{128K} \sum_{i=\max\{0,t-K+1\}}^{t} \left\|\bar{\mathbf{x}}_i - \mathbf{x}^\star\right\|^2 \\
&\quad + \frac{96K^2L^2\gamma^3\zeta_\star^2}{\mu} + \frac{7KL^2\gamma^3\sigma^2}{\mu M} + \frac{6KL^2\gamma^3\sigma^2}{\mu} + \frac{\gamma^2\sigma^2}{M}
\end{aligned}
$$

Here we use the Lemma B.10 with the following parameters,

$$
s_t = \mathbb{E}\left\|\bar{\mathbf{x}}_t - \mathbf{x}^\star\right\|^2 \;,\; a = \frac{23\mu}{16} \;,\; b = \frac{33\mu}{128} \;,\; c = \frac{96K^2L^2\gamma\zeta_\star^2}{\mu} + \frac{7KL^2\gamma\sigma^2}{\mu M} + \frac{6KL^2\gamma\sigma^2}{\mu} + \frac{\gamma\sigma^2}{M}
$$

The final inequality is:

$$
\begin{aligned}
\mathbb{E}\left\|\bar{\mathbf{x}}_t - \mathbf{x}^\star\right\|^2 &\leqslant \left(1 - \frac{23\gamma\mu}{32}\right)^t \mathbb{E}\left\|\mathbf{x}_0 - \mathbf{x}^\star\right\|^2 + \frac{32}{23\mu}\left(\frac{96K^2L^2\gamma\zeta_\star^2}{\mu} + \frac{7KL^2\gamma\sigma^2}{\mu M} + \frac{6KL^2\gamma\sigma^2}{\mu} + \frac{\gamma\sigma^2}{M}\right)\gamma \\
&\leqslant \left(1 - \frac{\gamma\mu}{2}\right)^t \mathbb{E}\left\|\mathbf{x}_0 - \mathbf{x}^\star\right\|^2 + \frac{96K^2L^2\gamma^2\zeta_\star^2}{\mu^2} + \frac{7KL^2\gamma^2\sigma^2}{\mu^2 M} + \frac{6KL^2\gamma^2\sigma^2}{\mu^2} + \frac{\gamma\sigma^2}{M\mu}
\end{aligned}
$$

Recall that we assumed $\gamma = \frac{\mu}{32KL^2}$ so we have:

$$
\mathbb{E}\left\|\bar{\mathbf{x}}_T - \mathbf{x}^\star\right\|^2 \leqslant \left(1 - \frac{\gamma\mu}{2}\right)^{KR} \mathbb{E}\left\|\mathbf{x}_0 - \mathbf{x}^\star\right\|^2 + \frac{96K^2L^2\gamma^2\zeta_\star^2}{\mu^2} + \frac{6KL^2\gamma^2\sigma^2}{\mu^2} + \frac{2\gamma\sigma^2}{M\mu}
$$

By setting $t = T = RK$, we get:

$$
\begin{aligned}
\mathbb{E}\left\|\bar{\mathbf{x}}_T - \mathbf{x}^\star\right\|^2 &\leqslant \left(1 - \frac{\gamma\mu}{2}\right)^{KR} \mathbb{E}\left\|\mathbf{x}_0 - \mathbf{x}^\star\right\|^2 + \frac{96K^2L^2\gamma^2\zeta_\star^2}{\mu^2} + \frac{6KL^2\gamma^2\sigma^2}{\mu^2} + \frac{2\gamma\sigma^2}{M\mu} \\
&\leqslant \exp\left(-\frac{\gamma\mu}{2}KR\right) \mathbb{E}\left\|\mathbf{x}_0 - \mathbf{x}^\star\right\|^2 + \frac{96K^2L^2\gamma^2\zeta_\star^2}{\mu^2} + \frac{6KL^2\gamma^2\sigma^2}{\mu^2} + \frac{2\gamma\sigma^2}{M\mu}
\end{aligned}
$$

We can see that with this inequality we can only guarantee convergence to a neighborhood of $\mathbf{x}^\star$. To obtain a convergence the final, as discussed in (Stich, 2019b), we need to choose the step size carefully. If $\frac{\mu}{32KL^2} \geqslant \frac{\ln(\max\{2,\mu^4\|\mathbf{x}_0-\mathbf{x}^\star\|^2 T^2/\sigma^2\})}{\mu T}$ then we choose $\gamma = \frac{\ln(\max\{2,\mu^4\|\mathbf{x}_0-\mathbf{x}^\star\|^2 T^2/\sigma^2\})}{\mu T}$, otherwise if $\frac{\mu}{32KL^2} < \frac{\ln(\max\{2,\mu^4\|\mathbf{x}_0-\mathbf{x}^\star\|^2 T^2/\sigma^2\})}{\mu T}$ then we choose $\gamma = \frac{\mu}{32KL^2}$ we can see that with these choices, we would have:

$$
\mathbb{E}\left\|\bar{\mathbf{x}}_T - \mathbf{x}^\star\right\|^2 = \tilde{\mathcal{O}}\left(\exp\left(-\frac{\mu^2}{64L^2}R\right)\|\mathbf{x}_0 - \mathbf{x}^\star\|^2 + \frac{K^2L^2\zeta_\star^2}{\mu^4 T^2} + \frac{KL^2\sigma^2}{\mu^4 T^2} + \frac{2\sigma^2}{M\mu^2 T}\right)
$$

$\square$

# G. Decoupled SGDA with Ghost Sequence

In this section, we introduce a new extension to the Decoupled SGDA algorithm called *Ghost Sequence*. The base Decoupled SGDA algorithm, explained earlier, is designed to take advantage of problems with a dominant separable component. It minimizes communication complexity by reusing outdated strategies, which has already been analyzed theoretically in the prevous sections.

However, we can push this idea further by not just reusing old strategies but also **predicting** the opponent's next move. This smarter approach opens up a new line of research, where more advanced methods can be explored for estimating the opponent's strategy, offering directions for future work.

To demonstrate the potential of this approach, we propose *Decoupled SGDA with Ghost Sequence*. The main idea is for each player to predict (or approximate) the next move of the opponent based on their previous actions and behaviour. This is achieved by computing the difference between successive strategies during synchronization. Using this information, each player can update both their own and their opponent's parameters, leading to improved performance. As shown in Figure 8, Decoupled SGDA with Ghost Sequence can greatly improve the algorithm's performance. It also achieves faster communication, even in highly interactive games, and does not require the problem to be weakly coupled.

For more details, refer to Algorithm 4.

---

**Algorithm 4** Decoupled SGDA with Ghost Sequence

---

1: **Input:** Step size $\gamma$, initial strategies $\mathbf{x}_0 = (\mathbf{u}_0, \mathbf{v}_0)$, total rounds $R$, local updates $K$
2: **for** $r \in \{1, \ldots, R\}$ **do**
3:      Calculate guess $\Delta_{\mathbf{u}}^r \leftarrow \frac{1}{K}(\mathbf{u}_0^r - \mathbf{u}_0^{r-1})$
4:      Calculate guess $\Delta_{\mathbf{v}}^r \leftarrow \frac{1}{K}(\mathbf{v}_0^r - \mathbf{v}_0^{r-1})$
5:      **for** $t \in \{0, \ldots, K-1\}$ **do**
6:          Update ghost sequence $\tilde{\mathbf{v}}_{t+1}^r \leftarrow \tilde{\mathbf{v}}_{t+1}^r + \Delta_{\mathbf{v}}^r$
7:          Update local strategy $\mathbf{u}_{t+1}^r \leftarrow \mathbf{u}_t^r - \gamma \nabla_u f(\mathbf{u}_t^r, \tilde{\mathbf{v}}_{t+1}^r; \xi_t^r)$
8:          Update ghost sequence $\tilde{\mathbf{u}}_{t+1}^r \leftarrow \tilde{\mathbf{u}}_{t+1}^r + \Delta_{\mathbf{u}}^r$
9:          Update local strategy $\mathbf{v}_{t+1}^r \leftarrow \mathbf{v}_t^r + \gamma \nabla_v f(\tilde{\mathbf{u}}_{t+1}^r, \mathbf{v}_t^r; \xi_t^r)$
10:      **end for**
11:      **Communicate** $(\mathbf{u}_K^r, \mathbf{v}_K^r)$ to other players
12: **end for**
13: **Output:** Final strategies $\mathbf{x}_K^R = (\mathbf{u}_K^R, \mathbf{v}_K^R)$

---

# H. Additional Experiments

## H.1. Finding the stationary point Decoupled SGDA for non-convex functions

Here, we add one more figure for the toy GAN problem to provide further insight into the behavior of Decoupled SGDA.

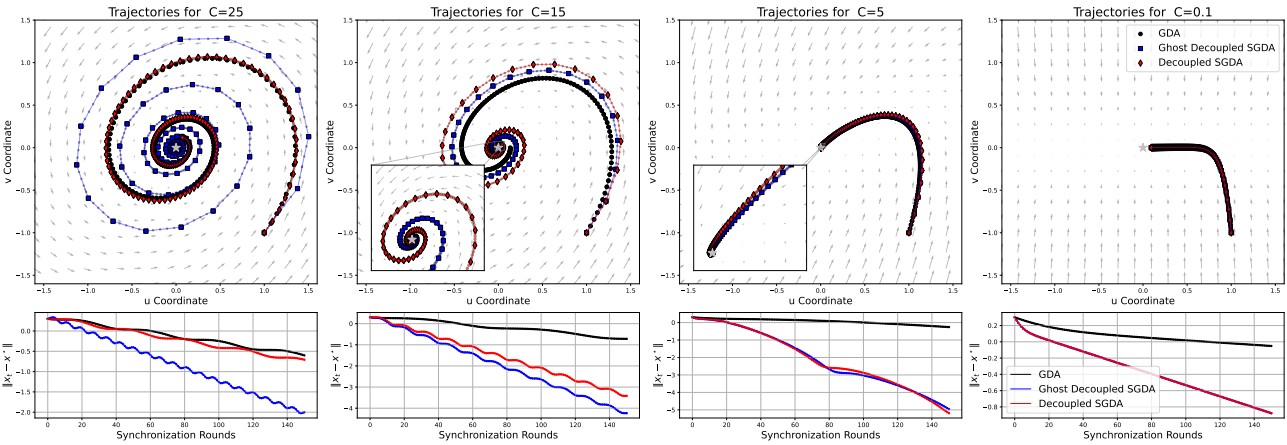

*Figure 8.* Trajectories and convergence comparison of GDA,*Decoupled SGDA* and *Decoupled SGDA with Ghost Sequence* with different values of $\mathbf{C} = C\mathbf{I}$ (interaction strength). The top row shows the trajectories of the different algorithms for $K = \{1, 5\}$ over varying values of $C \in \{25, 15, 5, 0.1\}$. As $C$ decreases, trajectories become more stable, with the Decoupled SGDA with Ghost Sequence (blue) showing more efficient convergence compared to GDA (black) and Decoupled SGDA (red). The bottom row presents the synchronization rounds versus distance to equilibrium for each configuration, highlighting faster convergence of Decoupled SGDA with Ghost Sequence under larger $\mathbf{C}$ values, while Decoupled SGDA with Ghost Sequence and Decoupled SGDA converge similarly for small $\mathbf{C}$.

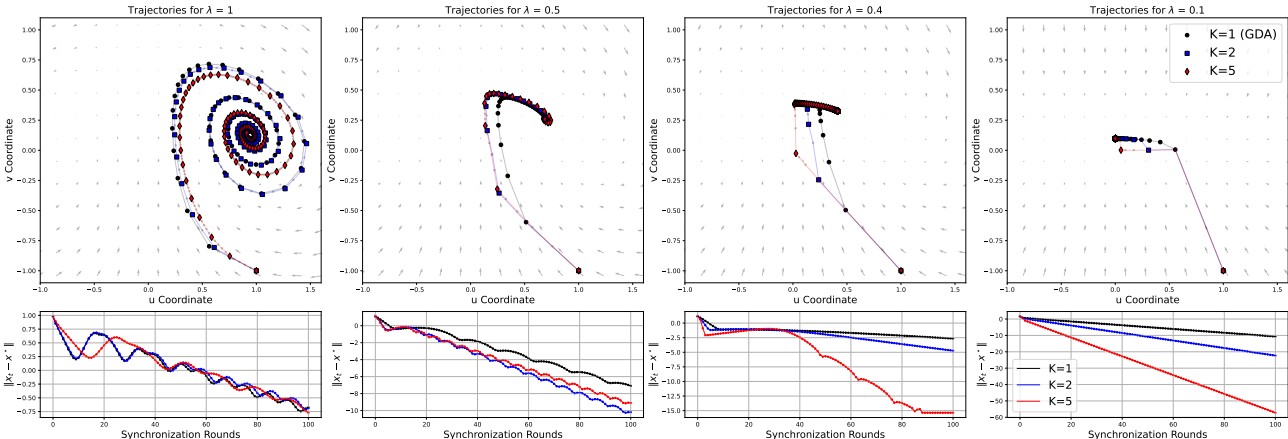

*Figure 9.* **Trajectories (top row) and distance to equilibrium over synchronization rounds (bottom row) of GDA ($K = 1$) and Decoupled SGDA with $K = \{2, 5\}$ on the** (14) **problem** ($d = 2$)**. C** in (13) is a constant here—the larger, the stronger the interactive term. **Left-to-right:** decreasing the constant $c \in \{10, 3.5, 2, 7, 0\}$.

## H.2. More Figures Decoupled SGDA With Gradient Approximation

In this experiment (Figure 10), Decoupled SGDA achieves lower gradient norms in fewer communication rounds compared to Local SGDA, especially as interaction noise increases (larger c). Decoupled SGDA shows much more stability in high-noise environments, highlighting its effectiveness in dealing with noisy gradients when compared to federated minimax settings.

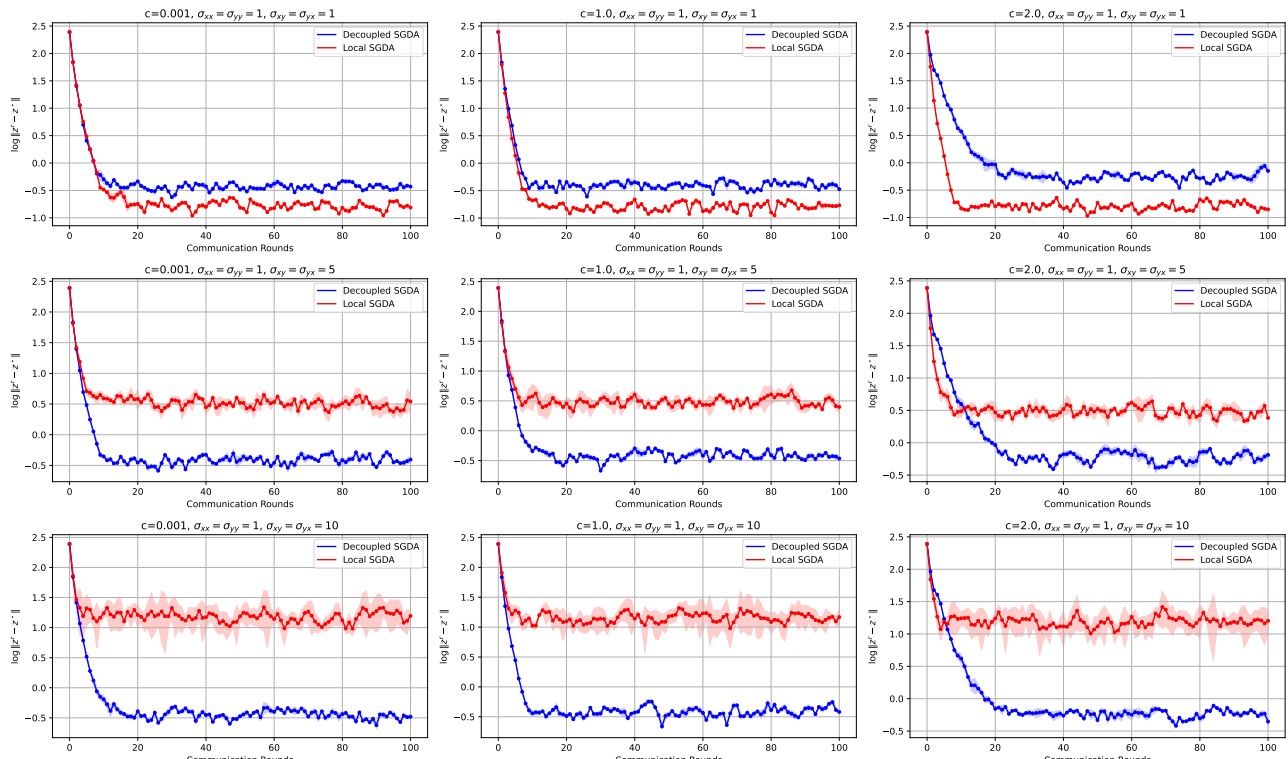

*Figure 10.* **Comparison of Decoupled SGDA and Local SGDA under different noise settings.** Each plot shows the smallest gradient norm achieved by both algorithms over 100 communication rounds, with varying interaction levels and noise variances. **Top Row:** Different settings of noise variances in off-diagonal entries (interaction noise). **Left to Right:** Increasing values of the constant $c$ controlling the interactive term's strength in the game. Decoupled SGDA consistently outperforms Local SGDA in scenarios where off-diagonal noise is significant, achieving lower gradient norms with fewer communication rounds.

# I. Experimental Setup

### I.1. Finding the saddle point of quadratic games

In the first experiment , we conducted tests with a dimensionality of $D = 2$ over $R = 31$ synchronization rounds. The values of $K$ tested were $1, 2$, and $5$, alongside parameter combinations $(a, b, c)$ set as $(1, 10, 10), (1, 10, 3.5), (1, 10, 2.7)$, and $(1, 10, 0)$. For each combination, we explored gamma values uniformly spaced in the interval $[0.0001, 0.1]$. The algorithm initializes $x$ and $y$ at $1$ and $-1$ respectively and updates these variables based on the gradients $g_x$ and $g_y$ computed using the defined parameters.

For the second experiment, in the left figure, eigenvalues were sampled logarithmically between $10^{-1.5}$ and $10^{1.5}$, with random symmetric positive definite matrices generated for each. We tested agent counts $K$ as $[1, 2, 5, 10, 50]$ and learning rates $\gamma$ from $10^{-10}$ to $1$. The algorithm ran for $R = 10^5$ rounds, adjusted based on eigenvalue size, to measure the average distance from equilibrium until it fell below $\epsilon = 10^{-6}$. Results were plotted to illustrate the relationship between $\lambda_{max}(C)$ and the number of rounds required for convergence.

For the left figure, we generated random symmetric positive definite matrices as oracles, varying the maximum eigenvalue of the matrix $C$ using logarithmic spacing between $10^{-1.5}$ and $10^{1.5}$. The accuracy threshold is set to $\epsilon = 10^{-4}$. We evaluated five algorithms: GDA, Decoupled GDA, Optimistic, Alternating Gradient Descent, and Extragradient, with $K$ fixed at 50. Each algorithm was executed for $R = 10^5$ rounds, determined based on the maximum eigenvalue, and their performance was assessed by the number of rounds required to achieve $\epsilon$ accuracy.

### I.2. Decoupled SGDA with Gradient Approximation

In this experiment, we analyze the performance of Decoupled and Local Stochastic Gradient Descent (SGDA) algorithms under varying conditions. We define oracles based on random symmetric positive definite matrices, with a fixed number of rounds $R = 100$ and $K = 40$. In the first experiment (left figure) The maximum eigenvalues of matrices $C$ are sampled logarithmically between $10^{-0.25}$ and $10^1$. For each maximum eigenvalue, we generate corresponding matrices and evaluate the algorithms across five trials to determine the lowest gradient norm achieved. We reported the mean of these five experiments. In the second experiment (right figure), off-diagonal variances ($\sigma_{\mathbf{vu}}^2$ and $\sigma_{\mathbf{uv}}^2$) range linearly from 1 to 10. In this experiment they are assumed to be equal. Results are aggregated and visualized in two plots: one depicting the relationship between the maximum eigenvalue of $C$ and the minimum gradient norm, and the other illustrating the effect of varying off-diagonal variance on algorithm performance.

### I.3. Communication Efficiency of Decoupled SGDA for Non-Convex Functions

In this experiment, we investigate the performance of Decoupled Single Oracle GDA under various settings of $\lambda$ and $K$. We evaluate the gradient norm achieved over $R = 100$ communication rounds. The $\lambda$ values are sampled logarithmically between $10^{-4.5}$ and $10^3$, while $K$ values range from 1 to 5. For each combination of $\lambda$ and $K$, we compute the lowest gradient norm over 5 independent trials. The gradient norms are averaged and plotted, with vertical lines marking the transition to the weakly coupled regime at $\lambda = 50$. The final results show the relationship between $\lambda$ and the minimum gradient norm for different values of $K$, highlighting the weakly coupled regime.

### I.4. Communication Efficiency of Decoupled SGDA in GAN Training

In this experiment, a Generative Adversarial Network (GAN) was trained using the CIFAR-10 and SVHN datasets, both resized to $32 \times 32$ pixels. The GAN was trained with a learning rate of $1 \times 10^{-4}$, a batch size of 256, and 50,000 rounds of updates. The hidden dimension size for the generator was 128. For evaluation, 256 samples were used to compute the Fréchet Inception Distance (FID) every 200 iterations. Both the generator and discriminator were optimized using the Adam optimizer, with a learning rate scheduler that decayed by a factor of 0.95 every 1000 steps. Additionally, a gradient penalty term was applied to stabilize training. The generator's latent space dimension was set to 100, and its Exponential Moving Average (EMA) was maintained with a decay factor of 0.999 for evaluation purposes. Training was conducted using CUDA on an NVIDIA L4 GPU.

The Generator uses a series of transposed convolutions, starting from a 100-dimensional latent vector, to generate a $32 \times 32 \times 3$ image, with BatchNorm and ReLU, ending with a Tanh activation. The Discriminator applies four convolutional layers to downsample the input, using LeakyReLU and BatchNorm, and outputs a real/fake probability through a Sigmoid activation.

