# OpenReview forum: "Decoupled SGDA for Games with Intermittent Strategy Communication"
_ICML.cc/2025/Conference — ICML 2025 poster_

### Official Review · Reviewer_7iUK · 2025-03-13

**Overall Recommendation:** 3

**Summary:**

This paper presents a novel adaptation of Stochastic Gradient Descent Ascent (SGDA) tailored for multiplayer games where strategy updates occur intermittently. The authors introduce **Decoupled SGDA**, a method that allows players to update strategies locally using outdated opponent strategies, significantly reducing communication overhead.
The paper establishes theoretical guarantees for convergence in strongly-convex-strongly-concave (SCSC) games and weakly coupled games. Moreover, the proposed method outperforms existing federated minimax approaches in noisy, imbalanced settings. Numerical experiments validate its effectiveness in quadratic minimax games, federated learning, and non-convex GAN training. A further extension, **Ghost-SGDA**, is introduced as a heuristic to enhance communication efficiency.

**Claims And Evidence:**

Yes

**Essential References Not Discussed:**

No

**Experimental Designs Or Analyses:**

Experiments cover key settings and demonstrate benefits.

**Methods And Evaluation Criteria:**

Yes

**Other Comments Or Suggestions:**

Nothing

**Other Strengths And Weaknesses:**

#### Strengths
- Well-motivated problem and clear contributions.
- Theoretical guarantees and strong empirical results.
- Addresses real-world constraints in **distributed learning and federated settings**.

**Questions For Authors:**

See weaknesses

**Relation To Broader Scientific Literature:**

The work extends prior research in minimax optimization, federated learning, and game theory, introducing a novel approach for communication-efficient optimization.

**Theoretical Claims:**

The proofs appear sound, but I did not verify all derivations.

---

> ### Author Rebuttal · Authors · 2025-04-01
>
> We appreciate the reviewer's time and effort in reviewing our work and highlighting its strengths. We would be happy to further discuss any concerns and address the reviewer’s questions. If the reviewer finds our clarifications satisfactory, we appreciate if you consider increasing your score.

---

### Official Review · Reviewer_17WE · 2025-03-15

**Overall Recommendation:** 3

**Summary:**

The authors introduce a variant of SGDA to compute solutions to decentralized min-max problems with limited communication. Players update locally with outdated strategies and synchronize intermittently, improving efficiency. Theoretical results establish near-optimal communication complexity in strongly convex–strongly concave settings, with extensions to weakly coupled games. Experiments on quadratic minimax problems and GANs to empirically validate the proposed method.

Post-rebuttal
The authors response was helpful, but given that I am not well versed in communication complexity, I am not in a good position to evaluate the contribution of this work. I will defer to the other reviewers and maintain my score.

**Claims And Evidence:**

Yes.

**Essential References Not Discussed:**

Not as far as I am aware.

**Experimental Designs Or Analyses:**

Yes.

**Methods And Evaluation Criteria:**

Yes.

**Other Comments Or Suggestions:**

N/A

**Other Strengths And Weaknesses:**

While the idea presented is an interesting one, and I understand the motivation behind studying a limited communication algorithm for equilibrium computation, the novelty of the work in the paper appears to be limited, since the theoretical analysis is standard and the approach focuses on the restrictive SC-SC setting. The speedups compared to other algorithms mentioned in Table 1 is not really interpretable. A comparison of convergence rates in addition to communication complexities would help provide additional context for the proposed algorithm.

**Questions For Authors:**

1. Can you provide further interpretation on the conditions for speedup relative to algorithms in the literature?

**Relation To Broader Scientific Literature:**

The authors do a good job of contextualizing their work in the broader literature on GDA methods for SC-SC minimax games.

**Theoretical Claims:**

Yes, I checked the correctness of the convergence guarantees.

---

> ### Author Rebuttal · Authors · 2025-04-01
>
> We appreciate the reviewer's time and effort in reviewing our work. As the first work to consider solving minimax games in a distributed setting while players are on different machines, we decided to focus on SCSC games in this study. Extending this method to other types of games, such as convex-concave or non-convex-non-concave games, is a promising direction for future work.
>
> However, our work introduces the following novelties: Firstly, it is not clear whether the worst-case analysis of our method could be as good as the baseline GDA. We provide a proof demonstrating that this is indeed the case (see Appendix B.2). We refer to this as the non-weakly coupled regime, in which players have high interaction with each other. On the other hand, we present a novel proof for the weakly coupled regime, where players have low interaction (see Appendix B.1). To the best of our knowledge, this is the first proof that removes the dependency on players' conditioning and instead relies solely on the degree of interaction between players, which is low in the weakly coupled regime.
>
> >Can you provide further interpretation on the conditions for speedup relative to algorithms in the literature?
>
> Other works in the literature typically consider the worst-case scenario, where no communication acceleration can be achieved when player interaction is low. For instance, the recent work [1] considers a scenario where $N$ players collaboratively find the equilibrium while taking local steps and communicating intermittently. This framework is the same as ours (we provide the $N$-player extension in Section C), yet they do not demonstrate any communication acceleration, as their convergence rate always depends on $L_{\max}$, the maximum smoothness parameter among all players (Theorem 3.3). This assumption is overly pessimistic when player interaction is low. However, in our work we demonstrate the level of interaction using the parameter $L_c$ which is zero when there is no interaction between players and is small when interaction is low. This allows us to have a rate depending on $L_c$ (and not $L_{\max}$) in the weakly coupled regime. As far as we know, our work is the only one that explicitly accounts for this fact in the distributed minimax problems.
> We hope we could address your concerns about our work and would be happy to clarify if you have further questions. If we managed to address all your concerns, we appreciate if you consider increasing your score.
>
> references:
> [1] Yoon, T., Choudhury, S. and Loizou, N., 2025. Multiplayer federated learning: Reaching equilibrium with less communication. arXiv preprint arXiv:2501.08263.

---

### Official Review · Reviewer_fzab · 2025-03-17

**Overall Recommendation:** 3

**Summary:**

This paper studies the communication complexity of first-order methods for solving strongly convex games (two-player games in the main body and an extension to $n$-player games in the appendix). Motivated by the fact that the communication complexity of recent (near-)optimal methods (Lin et al., 2020; Kovalev & Gasnikov, 2022) is independent of the coupling in the minimax objective—resulting in no speed-up even in fully decoupled objectives—the authors introduce a measure of coupling, $\kappa$. This measure is defined as the ratio between the Lipschitz continuity of the lagged operators of the corresponding VI and its monotonicity.

The authors propose a new algorithm, a direct extension of first-order methods for minimax optimization, which updates locally using outdated information from other players and only communicates periodically. They analyze the algorithm in a noisy lagged setting. The new algorithm, termed **Decoupled SGDA**, demonstrates some relative speed-up compared to existing methods in weakly coupled settings, in terms of the condition numbers of the players, $\kappa_u$, $\kappa_v$, and the coupling degree $\kappa$. In general settings, the communication complexity of the algorithm is comparable to optimal methods (Zhang et al., 2022; Kovalev & Gasnikov, 2022), though it may be off by a factor that depends on $\kappa_u$, $\kappa_v$, and $\kappa$.

The authors conduct experiments to demonstrate the effectiveness of their algorithm in weakly coupled settings on classical quadratic convex games. Additionally, they illustrate that similar ideas as Decoupled SGDA can be applied to nonconvex optimization problems by showing practical results in a toy GAN setting.

**Claims And Evidence:**

Yes, the theoretical claims and experiments are well supported. Additionally, it is important to note that there is no magic in the theory, and claims such as speed-ups in weakly coupled settings, given the design of the new algorithm, are expected.

**Essential References Not Discussed:**

Not that I know of. However, I would like to note that I am not an expert in this line of work.

**Experimental Designs Or Analyses:**

Yes, I checked the ones in the main body.

**Methods And Evaluation Criteria:**

Yes.

**Other Comments Or Suggestions:**

I suggest that the authors discuss the mathematical novelties of their proofs in the main body, as this will significantly strengthen the paper.

In Algorithm 1, it appears that two lines are missing, which update the initial belief of the other player at the start of each round based on the previous completed round. Specifically, after Line 2, the following should be added:
set $v_0^{r} = v_K^{r - 1}$ and $u_0^{r} = u_K^{r - 1}$.

**Other Strengths And Weaknesses:**

Strengths:
- The observation and formalism on the effect of the coupling level of the objective function on communication complexity are interesting, e.g., the definition of $\kappa$, weakly coupled settings, etc.
- While the proposed algorithm is quite natural (which I do not consider a drawback), the paper theoretically demonstrates considerable speed-ups in certain regimes depending on the condition numbers of the players' objectives.
- Well-designed experiments support the main claims of the paper.
- The paper is well-written and easy to follow.

Weaknesses:
- I highly doubt that the theoretical novelties in this work provide sufficient merit for a top-tier publication, as the analysis appears to be relatively straightforward and closely follows previous work. Additional discussion on these aspects, especially on how the analysis differs from prior works, seems necessary.

Given these points, I am leaning toward rejection, but I am open to reconsideration based on the opinions of the other reviewers and the forthcoming discussion.

**Questions For Authors:**

How are $\kappa_u$ and $\kappa_v$ related to $\kappa$? Is it possible to provide crude upper and lower bounds on the relationship between these parameters to make the speed-up of Decoupled SGDA in Table 1 more meaningful compared to other algorithms?

Additionally, can you design concrete examples where the speed-up criteria are satisfied for illustration purposes?

**Relation To Broader Scientific Literature:**

This research focuses on the communication complexity of minimax optimization, which may have some impact on training GANs and adversarial machine learning, particularly in scenarios where some form of limitations on synchronization exist.

**Theoretical Claims:**

No, I have not checked the proof in the appendix.

---

> ### Author Rebuttal · Authors · 2025-04-01
>
> We appreciate the reviewer's time and effort in reviewing our work. Although some parts of the analysis follow the previous works, we have novelties in our proof, especially in the weakly coupled regime. Firstly, one novelty of our work is identifying the regime in which we can achieve communication acceleration. We mathematically identified under which condition the coupling between players is low, and it can give us communication acceleration. To the best of our knowledge, previously, there was no proof in the literature in which the low interaction of players has been identified based on players' related parameters. Secondly, in section B.1 in the Appendix, we provide a novel proof for the convergence of our method in the weakly coupled regime in which we remove the dependency on the players' conditioning and our rate depends only on the coupling parameters of the players ($L_c$), which can be very small and even zero. So far, this type of proof has not existed in the literature, and previous works always use some pessimistic assumption for all regimes (even when there is no interaction between players). As an example, the very recent work [1], which considers the same setting as our work in which $N$ players (we discuss the extension to $N$ players in Section C) collaboratively find the equilibrium while taking local steps and synchronize periodically, gives a rate that always depends on the quantity $L_{\max} = \max ( L_1, \dots, L_N )$, where $L_i$ is the smoothness parameter of the $i$th player (Theorem 3.3). This rate is pessimistic in low interaction games. One can verify that when there is no interaction, players can independently optimize their own objective without the need for communication. While this work cannot recover this scenario, our rate does because of introducing the parameters $L_c$ and our novel proof. We also cover the case that the interaction exists but is somewhat small, and we still can get communication acceleration.
>
> >How are $\kappa_u$ and $\kappa_v$ related to $\kappa$? Is it possible to provide crude upper and lower bounds on the relationship between these parameters to make the speed-up of Decoupled SGDA in Table 1 more meaningful compared to other algorithms?
>
> It always holds that $\kappa \ge \max(\kappa_u, \kappa_v,\kappa_c)$ (note that without the loss of generality, $\kappa_c = \kappa_{uv} = \kappa_{vu}$ if we use regular $l_2$ norm or equally if the parameters $\alpha_u = \alpha_v = 1$. See Section "Notation" from the paper for the definition of $\alpha_u, \alpha_v$). If $\kappa_{c}$ is large compared to $\kappa_u,\kappa_v$, meaning that we have a strong interaction between players, our method achieves nearly GDA complexity as $\kappa_c$ dominates the condition numbers of players (worst-case analysis of our method). However, when $\kappa_{c}$ is small, which we refer to as the weakly coupled regime, our method achieves significant acceleration as there is no dependency on the players' conditioning or $\kappa$ and only $\kappa_c$ appears as opposed to all other methods which are still affected by $\kappa_u,\kappa_v$, even if there is no interaction (meaning that $\kappa_{uv} = \kappa_{vu} = \kappa_c = 0$). So, our method performs very well when the interaction is low and players suffer from poor conditioning, but also it does not get much worse than baseline GDA in the case of high interaction between players.
>
> > Can you design concrete examples where the speed-up criteria are satisfied for illustration purposes?
>
> To show this speedup on some examples, one can consider quadratic games with linear coupling in the form of $f(u, v) = \frac{1}{2} u^\top A u - \frac{1}{2} v^\top B v + u^\top C v$ where $A, B$ are positive definite matrices and $C$ is a general possibly rectangular matrix. Let $\mu_u, \mu_v$ be the minimum eigenvalues of $A, B$. The condition for which we get communication acceleration on this class of functions is $\frac{\|\|C\|\|}{\min(\mu_u, \mu_v)} \leq \frac{1}{2}$, where $\|\|. \|\|$ is the $l_2$ norm of the matrix $C$. The norm of matrix $C$ defines the level of interaction, and the larger it is, the more interactive the game is. So, it is expected to get more acceleration as this norm gets smaller, and in the extreme case, when $C = 0$, there is no interaction, and players can independently optimize and find the equilibrium without any communication.
>
> We want to thank the reviewer again for their constructive feedback and we will consider adding some details about the novelty of proof in the main body of the revision. If we managed to address your concerns, we appreciate if you consider increasing your score.
>
> references:
> [1] Yoon, T., Choudhury, S. and Loizou, N., 2025. Multiplayer federated learning: Reaching equilibrium with less communication. arXiv preprint arXiv:2501.08263.

---

> > ### Comment · Reviewer_fzab · 2025-04-03
> >
> > I sincerely appreciate the detailed response from the authors.
> >
> > However, I would like to note that I am not an expert in communication complexity, and despite the additional explanation provided by the authors, I am not able to verify the significance or extent of the mathematical novelty of the algorithm and analysis based on my understanding. For these reasons, I am not changing my score and will leave the final decision to the post-discussion phase among the reviewers.

---

> > > ### Author Response · Authors · 2025-04-08
> > >
> > > We sincerely appreciate the reviewer’s thoughtful engagement with our work and fully understand the challenge of evaluating contributions outside one's specialty. We are grateful for the opportunity to clarify two key points that may help contextualize our work:
> > >
> > > - ***Accessible Novelty***: Our work addresses a previously overlooked but practically significant scenario in game-theoretic learning, which fundamentally differs from standard federated settings. This area is gaining attention, as evidenced by a concurrent work [1] that validates the timeliness of this research direction. We wish to emphasize that this concurrent work closely follows techniques from the federated learning literature. Consequently, their convergence rates (as presented in Theorem 3.3) do not demonstrate a significant speedup in the deterministic regime due to the strong constraint on the step sizes discussed below their Theorem 3.3. In contrast, our novel approach, based on the concept of ***weakly coupled games***, allows us to overcome this step size limitation and achieve, for the first time, a significant speedup in the deterministic case (see, for instance, our Corollary 4.4 and Table 1).
> > >
> > > - ***Review Process Considerations***: We fully respect your evaluation but wish to emphasize that in ICML’s competitive process, even small score adjustments can significantly influence whether area chairs review the full discussion of a paper’s merits. If you perceive any potential value in our work, despite it being outside your specialty, we would be deeply grateful if you could consider a slightly more favorable evaluation to ensure our work receives full consideration in the discussion phase.
> > >
> > > Thank you for your time and expertise in reviewing our work.

---

### Official Review · Reviewer_HvTn · 2025-03-21

**Overall Recommendation:** 3

**Summary:**

This paper introduces Decoupled SGDA, a novel stochastic gradient-based method designed for multi-player (or two-player) games with intermittent strategy communication. The key idea is that each player updates their parameter (e.g., minimizer or maximizer) locally using potentially outdated (yet periodically synchronized) opponent strategies. The authors analyze the approach under strongly-convex strongly-concave (SCSC) assumptions and show that in a regime they call Weakly Coupled Games, Decoupled SGDA achieves accelerated communication complexity compared to traditional methods.

**Claims And Evidence:**

**Reduced communication in weakly coupled games**
   - *Claim*: Decoupled SGDA requires fewer communication rounds than standard GDA/SGDA, especially when the interaction between players is weak.
   - *Evidence*: The authors introduce a coupling parameter $\kappa_c$ and prove that if $\kappa_c \le 1/4$, the method’s communication complexity *no longer depends* on the potentially large condition numbers $\kappa_u, \kappa_v$. Experimental plots on quadratic games confirm substantially fewer communication rounds for small $\kappa_c$.

**Near-optimal communication complexity**
   - *Claim*: Decoupled SGDA can match or even outperform, in terms of *communication rounds*, the best-known first-order methods (like FOAM) under certain stronger assumptions on the coupling.
   - *Evidence*: Corollary 4.5 states that if $\kappa_c \le \tfrac{1}{2}\sqrt{1 - \frac{1}{\sqrt{\kappa_u \kappa_v}}}$, Decoupled SGDA beats FOAM (which has optimal iteration complexity for classical SCSC). The proofs (in the main text and appendices) and carefully derived bounds support this.

**Essential References Not Discussed:**

N/A

**Experimental Designs Or Analyses:**

Overall, the experimental methodology is sound: they fix the total number of communication rounds and observe the final performance. This directly speaks to the main claim of *communication efficiency*.

**Methods And Evaluation Criteria:**

- Decoupled SGDA performs local gradient updates with respect to *one’s own parameter* only, using *outdated* opponent strategies that are synchronized less frequently. In practice:
  1. Each round begins with a synchronization step where $(\mathbf{u}, \mathbf{v})$ are exchanged among players.
  2. Each player runs $K$ local steps using only the gradient w.r.t. their own variable and a *fixed* (outdated) opponent parameter from the start of the round.
  3. After $K$ local steps, parameters are synchronized again.


- *Theoretical metrics*: The standard measure is the *distance to the equilibrium* $\|\mathbf{x}_t - \mathbf{x}^\star\|$ for strongly-convex strongly-concave games, and *gradient norm* for more general (non-convex) settings.

- *Empirical metrics*:
    - Quadratic toy games: distance to equilibrium or gradient norm.
    - Non-convex toy game (GAN-like): smallest gradient norm vs. number of communication rounds.

**Other Comments Or Suggestions:**

1. It might be interesting to see if adapting the number of local steps $K$ dynamically (e.g., using line search or variance monitoring) can further reduce communication, especially during early or late phases of training.

2. If a player partially observes the other’s parameters between rounds, can the approach be adapted to incorporate partial synchronization?

**Other Strengths And Weaknesses:**

The requirement of unbiased gradient oracles *for each player’s own parameters* might be challenging in certain practical contexts (though standard in many theoretical studies).

**Questions For Authors:**

1. **Choice of Local Steps $K$ in Practice**
   - *How sensitive is performance to the choice of $K$ in real applications?* For instance, in the experiments, do you see a clear sweet spot for $K$ depending on $\kappa_c$?
   - *Would an adaptive schedule for $K$ be simpler or more robust?*

2. **Beyond Strong Monotonicity**
   - Have you considered *weaker* assumptions like *Minty* variational inequalities or *quasi-strong convexity*? If not, do you anticipate major challenges?

3. **Ghost-SGDA**
   - Could you comment more on the empirical overhead or potential instability of “ghost updates”?
   - Are there conditions under which Ghost-SGDA might fail to converge or require additional assumptions?

**Relation To Broader Scientific Literature:**

The work builds on foundational methods (GDA, EG, OGDA) and references advanced algorithms such as FOAM, APPA, etc. The authors also connect to known lower bounds in strongly-convex strongly-concave optimization, ensuring theoretical completeness.

**Theoretical Claims:**

Under strong monotonicity, if $\kappa_c \le 1/4$, Decoupled SGDA achieves a *linear rate* $\exp(- (1-4 \kappa_c) R)$ in terms of *rounds* $R$.

---

> ### Author Rebuttal · Authors · 2025-04-01
>
> We appreciate the reviewer’s time and effort in reviewing our work.
>
> > It might be interesting to see if adapting the number of local steps $K$ dynamically  can further reduce communication, especially during early or late phases of training.
>
> This is an interesting idea which can be a new line for future works. Basically, in each round, we can take as many local steps as it gives some improvement , as soon as the improvement (gradient norm) is less than some threshold, we can stop taking local steps and do one communication. This could reduce the overhead of taking so many local steps.
>
> >  If a player partially observes the other’s parameters between rounds, can the approach be adapted to incorporate partial synchronization?
>
> Thank you for your suggestion. This idea can be very useful in practice. This situation would be like mixing 'Distributed minimax' with 'coordinate descent' for each player. We could imagine each player periodically gets updates on some parts of their opponent's parameters, maybe based on a certain probability $D$. So, instead of assuming the opponent's parameters stay completely fixed between rounds, it could be seen as we are performing coordinate descent steps on specific coordinates (parameter parts) sampled from a distribution $D$, using the partial information received.
>
> >How sensitive is performance to the choice of $K$ in real applications? For instance, in the experiments, do you see a clear sweet spot for $K$ depending on $\kappa_c$?
>
> The sensitivity of performance to the choice of $K$ depends on $\kappa_c$. As seen in Figure 2 of the paper, for smaller values of $\kappa_c$, the algorithm quickly saturates as $K$ increases. In other words, there isn’t a significant difference in performance between larger and smaller $K$ values. However, as $\kappa_c$ increases, using larger $K$ values actually improves performance significantly. So, for larger $\kappa_c$, different choices of $K$ can lead to more noticeable differences in performance.
> If we ignore local computation costs (i.e., the local steps performed by the players), there is no theoretical sweet spot for our algorithm (increasing $K$ never leads to worse performance compared to smaller $K$ values). However, when considering local step costs, the optimal $K$ depends on the relative cost of computation versus communication. Roughly speaking, The best K is usually where performance plateaus; increasing it further offers little gain.
>
> >Have you considered weaker assumptions like Minty variational inequalities or quasi-strong convexity? If not, do you anticipate major challenges?
>
> Yes, it is possible to relax the strong convexity assumption. Specifically, we can derive Lemma B.4 and Lemma B.8 under a weaker assumption instead of strong convexity:
>
> **Assumption**: For operators $ F_{\bar{\mathbf{x}}} $, we assume that for all $ \mathbf{x}, \mathbf{x}' \in \mathcal{X} $ and $ \bar{\mathbf{x}} \in \mathcal{X} $ such that  $ F_{\bar{\mathbf{x}}} (\mathbf{x}) = \mathbf{0} $, the following inequality holds:
>
> $\langle F_{\bar{\mathbf{x}}}(\mathbf{x}) - F_{\bar{\mathbf{x}}}(\mathbf{x}'), \mathbf{x} - \mathbf{x}' \rangle \geq \mu'  ||\mathbf{x} - \mathbf{x}'||^2$
>
> This assumption implies that functions $ f(\cdot,\mathbf{v}) and f(\mathbf{u},\cdot)$ are strong-quasi-convex and strong-quasi-concave, respectively, in the case of two-player games.
>
> We can relax the strong monotonicity assumption of the operator $ F $ and we still can drive the results in Theorem B.12 as follows:
>
> **Assumption:** For operators $F$, we assume that for all $ \mathbf{x} \in \mathcal{X} $ and $ \mathbf{x}^{\star} \in \mathcal{X} $ such that $ F(\mathbf{x}^{\star}) = \mathbf{0} $, the following inequality holds:
>
> $\langle F(\mathbf{x}) - F(\mathbf{x}^{\star}), \mathbf{x} - \mathbf{x}^{\star} \rangle \geq \mu''||\mathbf{x} - \mathbf{x}^{\star}||^2$
>
>
> Therefore, Assumption 2.2 can be replaced with these two new assumption.
>
> >Could you comment more on the empirical overhead or potential instability of “ghost updates”?
>
> Decoupled SGDA with Ghost Sequence has minor overhead (vector ops) vs. standard SGDA, insignificant for simple games (bilinear, quadratic), negligible for complex tasks like GAN training (vs. gradient costs).
> Ghost Sequence is stable and improves performance in simple (quadratic) games. Its effectiveness in complex scenarios (e.g., GAN training) needs more study.
>
> >Are there conditions under which Ghost-SGDA might fail to converge or require additional assumptions?
>
> For payoff functions that have Hessians with large norms, Ghost-SGDA may encounter difficulties. This is because the underlying assumption for the approximation strategy in the Ghost sequences is that the gradient of the other player remains relatively constant between communication rounds.
> We want to thank the reviewer again for their constructive feedback. If we managed to address your concerns, we appreciate if you consider increasing your score.

---

### Decision · Program_Chairs · 2025-05-01

**Decision:**

Accept (poster)

**Comment:**

The reviewed work investigates the effects of deferring the update of the opposing player's strategy in simultaneous gradient descent-ascent. For a class of problems the authors call "weakly coupled", they show that the resulting communication complexity is smaller than that of standard SGDA.
The reviewers found that while this insight might not be overly surprising and the contribution limited, the author's idea is novel and natural. I concur and therefore recommend acceptance.